# VisCoder2: Building Multi-Language Visualization Coding Agents

**Yuansheng Ni**[1][*][†], **Songcheng Cai**[1][*], **Xiangchao Chen**[1][*], **Jiarong Liang**[1], **Zhiheng Lyu**[1],
**Jiaqi Deng**[3], **Ping Nie**[5][♣] **Kai Zou**[4], **Fei Yuan**[5], **Xiang Yue**[2], **Wenhu Chen**[1][*][†]

[1]University of Waterloo,  [2]Carnegie Mellon University,
[3]Korea Advanced Institute of Science & Technology,  [4]Netmind.ai,  [5]Independent Researcher

https://tiger-ai-lab.github.io/VisCoder2

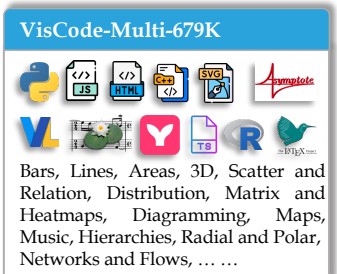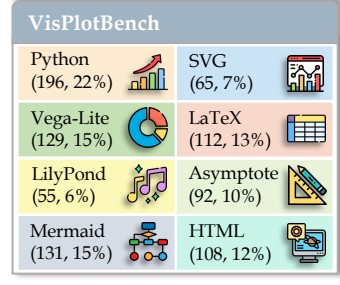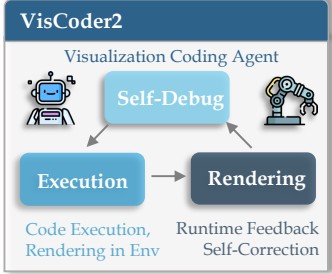

Figure 1: Overview of VisCoder2. We present three components: 1) **VisCode-Multi-679K:** a dataset of 679K executable visualization samples and multi-turn correction dialogues, spanning 12 programming languages; 2)**VisPlotBench**: spanning 8 languages with natural language instructions, executable code, and rendered outputs; 3)**VisCoder2**: a family of visualization coding agents that iteratively execute, render, and self-debug, approaching the performance of proprietary models.

## Abstract

Large language models (LLMs) have recently enabled coding agents capable of generating, executing, and revising visualization code. However, existing models often fail in practical workflows due to limited language coverage, unreliable execution, and lack of iterative correction mechanisms. Progress has been constrained by narrow datasets and benchmarks that emphasize single-round generation and single-language tasks. To address these challenges, we introduce three complementary resources for advancing visualization coding agents. **VisCode-Multi-679K** is a large-scale, supervised dataset comprising 679K validated executable visualization samples and multi-turn correction dialogues, covering 12 programming languages. **VisPlotBench** is a benchmark for systematic evaluation, featuring executable tasks, rendered outputs, and protocols for both initial generation and multi-round self-debug. Finally, we present **VisCoder2**, a family of multi-language visualization models trained on VisCode-Multi-679K. Experiments show that VisCoder2 significantly outperforms strong open-source baselines and approaches the performance of proprietary models like GPT-4.1, with further gains from iterative self-debug, reaching **82.4%** overall execution pass rate at the 32B scale, particularly in symbolic or compiler-dependent languages.

## 1 Introduction

Recent advances in large language models (LLMs) have enabled coding agents Jimenez et al. (2023); Yang et al. (2024b) that can generate visualization code, execute it, and even revise their outputs in response to feedback (Robeyns et al., 2025; Li et al., 2025b). These agents are increasingly

---

[1]Core Contributors; ♣ Project Lead; † Corresponding: {yuansheng.ni, wenhuchen}@uwaterloo.ca

applied to data analysis and reporting workflows, where producing plots and diagrams is a central task (Galimzyanov et al., 2024).

While existing models can attempt these steps, they often fail in practice: generating code that crashes, produces incorrect visuals, or lacks flexibility across programming languages and libraries (Goswami et al., 2025). Building more reliable visualization coding agents requires resources that go beyond single-round generation, supporting multi-language coverage, runtime validation, and iterative correction through execution feedback (Yang et al., 2023). However, current datasets and benchmarks lack these capabilities, limiting progress toward agents that can effectively assist in real-world visualization workflows (Ni et al., 2025).

Visualization presents a uniquely valuable setting for advancing these agents. Unlike general-purpose code generation (Li et al., 2022), visualization tasks produce clear and interpretable outputs: the execution process and rendered figure provide an immediate signal of whether the code executed successfully and whether the output aligns with the intended result (Ni et al., 2025). Moreover, visualization requires cross-domain reasoning, combining knowledge of data handling, plotting syntax, and design conventions (Satyanarayan et al., 2016). Crucially, real-world workflows are inherently iterative — analysts rarely produce perfect visualizations on the first attempt, instead refining their code based on runtime behavior and visual inspection (Goswami et al., 2025). This natural feedback loop makes visualization tasks especially well-suited for developing agents that can generate and self-correct code (Chen et al., 2023).

Despite this potential, existing resources for visualization code generation remain narrow in scope. Most datasets focus on single languages, such as Python or Vega-Lite (Galimzyanov et al., 2024; Luo et al., 2021), and include many snippets that cannot be executed reliably (Ni et al., 2025). They lack validated, executable samples, and they do not provide the multi-turn interactions needed to train models for iterative debugging (Ni et al., 2025). Existing benchmarks also have significant gaps: they emphasize single-round generation and do not support systematic evaluation across languages or multi-round repair scenarios (Yang et al., 2023). As a result, current models are tested in settings that fail to capture the complexity of real-world visualization development (Goswami et al., 2025).

To address these limitations, we introduce two complementary resources. First, we present **VisCode-Multi-679K**, a large-scale supervised instruction-tuning dataset comprising 679K executable visualization and code-correction samples cover twelve programming languages. VisCode-Multi-679K combines validated visualization code extracted from diverse open-source repositories (Lozhkov et al., 2024; Yang et al., 2025a; Rodriguez et al., 2025) and multi-turn dialogues that teach models to revise faulty code based on execution feedback (Zheng et al., 2024). Second, we propose **VisPlotBench**, a benchmark for evaluating visualization coding agents across eight languages. VisPlotBench provides carefully curated, executable tasks with natural language instructions and rendered outputs, along with a standardized evaluation protocol for both initial generation and multi-round self-debug (Galimzyanov et al., 2024).

Finally, we train **VisCoder2**, a family of multi-language visualization models built on VisCode-Multi-679K. VisCoder2 substantially outperforms size-matched open-source baselines (Hui et al., 2024; Guo et al., 2024a; Ni et al., 2025) and closes much of the performance gap with proprietary models such as GPT-4.1 (Fachada et al., 2025). Experiments show that iterative self-debug yields further improvements, reaching **82.4%** at the 32B scale, *on par with GPT-4.1 and surpassing GPT-4.1-mini*, particularly benefiting symbolic or compiler-dependent languages like `LilyPond`, `LaTeX`, and `Asymptote`. Together, **VisCode-Multi-679K**, **VisPlotBench**, and **VisCoder2** establish a foundation for building and evaluating visualization coding agents that can operate reliably across diverse programming languages and real-world visualization tasks.

## 2 RELATED WORK

**LLMs for Visualization Code Generation**  Large language models have shown promising results in generating visualization code from natural language descriptions (Yang et al., 2024c; Chen et al., 2024; Galimzyanov et al., 2024). Most existing approaches focus on single languages, particularly Python with matplotlib or plotly (Wu et al., 2024; Yang et al., 2024a), while some explore specification-based methods using Vega-Lite (Xie et al., 2024) and HTML (Li et al., 2025a). However, these systems face significant limitations: they typically support only one or two programming

languages, lack systematic execution validation, and often generate code that fails to run reliably (Ni et al., 2025; Sun et al., 2025). Multi-language code generation efforts in broader domains (Lozhkov et al., 2024; Muennighoff et al., 2023) provide extensive language coverage but lack the specialized knowledge required for visualization tasks, particularly for domain-specific languages like LaTeX for mathematical plots or LilyPond for musical notation. Our VisCode-Multi-679K dataset addresses these limitations by providing validated, executable visualization samples across twelve programming languages, enabling robust multi-language visualization code generation with systematic quality control and execution verification.

**Self-Debug and Coding Agents**   Recent advances in coding agents have emphasized iterative development capabilities, where models can generate, execute, and refine code through multiple rounds of feedback (Jimenez et al., 2023; Yang et al., 2024b). Self-debug approaches leverage execution traces, error messages, and runtime outcomes to guide automatic code correction (Chen et al., 2023; Madaan et al., 2023; Zheng et al., 2024; Zeng et al., 2025). Agent-based systems further extend these capabilities by incorporating planning, tool use, and collaborative debugging workflows (Grishina et al., 2025; Li et al., 2024; Zhang et al., 2025). While these methods show promise in general programming tasks, their application to visualization remains underexplored. Existing visualization systems like LIDA (Dibia, 2023) incorporate some feedback mechanisms, but lack the systematic multi-turn correction capabilities needed for reliable cross-language deployment. Our work uniquely combines multi-language visualization generation with systematic self-debug, enabling VisCoder2 to iteratively refine code across diverse programming environments, particularly excelling in symbolic languages where execution validation is essential.

**Visualization Benchmark**   Existing visualization benchmarks focus predominantly on Python (Galimzyanov et al., 2024; Chen et al., 2024; Yang et al., 2024c; Rahman et al., 2025) or declarative specifications like Vega-Lite and HTML (Luo et al., 2021; 2025; Li et al., 2025a), limiting their applicability across diverse programming environments used in real-world data analysis. While general code datasets like *the-stack-v2* (Lozhkov et al., 2024) provide broad language coverage, they lack visualization-specific content and execution validation. Most visualization benchmarks evaluate only single-turn generation, failing to capture the iterative debugging workflows that characterize practical visualization development (Ni et al., 2025; Seo et al., 2025). Without multi-language support and multi-round evaluation, existing benchmarks cannot assess whether models can handle the diverse toolchains and iterative workflows essential for real-world visualization tasks. VisCode-Multi-679K addresses these limitations by providing the first large-scale dataset with execution-validated visualization code across twelve programming languages, while VisPlotBench enables a systematic evaluation of both initial generation and multi-turn self-debug capabilities across visualization tasks in multiple programming languages.

## 3   VisCode-Multi-679K: An Instruction Tuning Dataset for Visualization Across Twelve Programming Languages

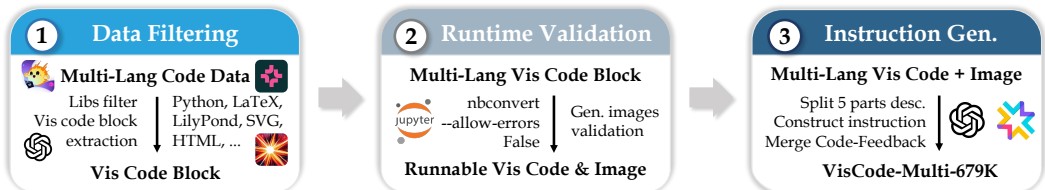

Figure 2: Data construction pipeline for VisCode-Multi-679K. We collect code blocks across twelve programming languages from open-source repositories, including large-scale code corpora, synthetic visualization datasets, and domain-specific diagram collections. We validate executability and render outputs through Jupyter-based runtime checks, yielding instructions paired with images. We integrate multi-turn dialogues from *Code-Feedback* to provide iterative correction supervision.

We present **VisCode-Multi-679K**, a supervised instruction tuning dataset for visualization code generation and feedback-driven correction across twelve programming languages. The dataset supports

robust multi-language code generation and enables iterative refinement through multi-turn supervision, aligning with the needs of interactive visualization workflows.

VisCode-Multi-679K unifies two complementary sources of supervision. The first is a large collection of executable visualization code extracted from open source repositories across twelve programming languages, spanning diverse chart types, libraries, and real-world usage patterns. Each sample is validated for runtime execution and paired with its rendered output, ensuring reliable supervision for multi-language code generation. The second source is 66K multi-turn dialogues from the Code Feedback dataset (Zheng et al., 2024), which provide training signals for revising faulty code based on execution feedback. Although these dialogues are not exclusively visualization-oriented, they are essential for modeling realistic self-correction behaviors in iterative workflows.

Figure 2 summarizes the construction pipeline of VisCode-Multi-679K, forming the raw material for a four-stage process: library-based filtering, code block extraction, runtime validation, and instruction generation. The following subsections detail each stage. Quantitative analysis is provided in Appendix I.4.

## 3.1 Code Extraction from Public Repositories

We construct VisCode-Multi-679K by drawing on three complementary open source corpora: *the-stack-v2*[1] (Lozhkov et al., 2024), *svg-diagrams*[2] (Rodriguez et al., 2025), and *CoSyn-400K*[3] (Yang et al., 2025b; Deitke et al., 2024). These sources are complementary: *the-stack-v2* provides large-scale, diverse code across many languages, capturing realistic visualization embedded in general programs; *svg-diagrams* contributes domain-specific SVG samples focused on diagram rendering; and *CoSyn-400K* offers synthetic but cleanly structured visualization code spanning multiple languages. Together, they cover both natural and synthetic usage across a wide range of languages and visualization styles. From each corpus, we extract code that invokes widely used visualization libraries to capture real-world plotting practices. These sources provide the raw material for a pipeline with four stages: library-based filtering for each language, code block extraction, runtime validation, and instruction generation.

**Filtering and Code Block Extraction.** For *the-stack-v2* (Lozhkov et al., 2024), which contains approximately 900B tokens of code, we restrict our selection to two filtered subsets: *stack-edu*[4] (Allal et al., 2025) and *the-stack-v2-train-smol-ids*[5]. *stack-edu* was curated from *the-stack-v2* using a classifier-based filtering strategy that retains only high-quality educational programming content. *the-stack-v2-train-smol-ids* is a near-deduplicated subset further filtered with heuristics and spanning 17 programming languages. We first apply library-based filters on these subsets to identify approximately 5.3M visualization code candidates in `Python`, `JavaScript`, `C++`, `TypeScript`, `HTML`, and `R`. Because most examples are embedded in broader program contexts rather than self-contained plotting examples, we use GPT-4.1-mini (OpenAI, 2025) to extract standalone plotting blocks for each language. When the original code does not include data, we inject mock inputs so that each block can execute in isolation. This structural cleaning preserves realistic visualization usage while remaining compatible with our runtime pipeline. After filtering and reconstruction, we obtain roughly 900K candidate blocks.

For *svg-diagrams*, which contains 182K domain-specific SVG samples focused on diagrams from *star-vector* (Rodriguez et al., 2025), we apply regular-expression filtering to remove noisy data that lack width, height, or other essential components. This step retains about 79K candidate blocks.

For *CoSyn-400K*, we select 408K visualization snippets across eight languages, including `Python`, `HTML`, `LaTeX`, `SVG`, `Asymptote`, `Mermaid`, `LilyPond`, and `Vega-Lite`. *CoSyn-400K* provides synthetic but cleanly structured code spanning a wide range of styles, with well-rendered outputs and consistent structure. Unlike *the-stack-v2*, its `Python` and `HTML` code store logic and data separately, which requires reconstruction for runtime execution. For languages requiring reconstruction, we rebuild runnable scripts by inserting lightweight annotations such as column headers and

---

[1] hf.co/datasets/bigcode/the-stack-v2
[2] hf.co/datasets/starvector/svg-diagrams
[3] hf.co/datasets/allenai/CoSyn-400K
[4] hf.co/datasets/HuggingFaceTB/stack-edu
[5] hf.co/datasets/bigcode/the-stack-v2-train-smol-ids

a data row to emulate realistic data loading. When necessary, we append missing plotting function calls to ensure that each language can execute within a Jupyter notebook environment.

**Runtime Validation.** To ensure executability, we run each candidate block in isolated Jupyter environments. `C++`, `JavaScript`, and `R` are executed in dedicated kernels, while all other languages share the Python kernel. Each block is run with `nbconvert` using `allow-error=False` to enforce strict filtering. We apply a fixed timeout and terminate runs that hang or enter infinite loops via a simulated keyboard interrupt. Only samples that execute successfully and generate valid image files that are non-monochrome and larger than 10KB are retained. This step produces 245K validated plotting scripts from *the-stack-v2*, 43K from *svg-diagrams*, and 322K from *CoSyn-400K*, each paired with its rendered output. The detailed distribution is shown in Table 5.

**Instruction Generation.** To enable models to learn from both structural code features and rendered visual outputs, we generate natural language instructions for each validated example using GPT-4.1 (OpenAI, 2025). This process ensures that supervision captures not only code syntax but also the semantics of the corresponding visualization.

To capture both data semantics and visual design, each instruction is structured into five components: (1) a brief setup description specifying the programming language and visualization libraries used; (2) a description of either the underlying data (for data-driven code) or the visible elements of the figure (for non-data-driven code); (3) a data block that either contains a copied data-generation line or a two-row preview, left empty for non-data-driven cases; (4) a high-level output description that conveys the intended visualization conceptually; and (5) a style description capturing colors, grid layout, and other visual properties. These components are assembled into a fixed template:

```
[Output Description]
[Setup]
[Data/Visual Description]
"The data is shown below:" or None
[Data] or None
[Style Description]
```

This format enforces a consistent prompt structure across sources and languages, ensuring that models receive a unified description of the visualization target, its data, and its stylistic attributes.

## 3.2 MULTI-TURN INSTRUCTION-FOLLOWING DIALOGUES WITH EXECUTION FEEDBACK

VisCode-Multi-679K further includes over 66K multi-turn dialogues from the *Code-Feedback*[6] dataset (Zheng et al., 2024). These dialogues cover programming tasks in `Python`, `HTML`, `JavaScript`,`R`, and other languages, with user instructions, model-generated code, and follow-up turns carrying execution feedback or revision prompts.

Although not tailored to visualization, they provide essential supervision for teaching models to revise faulty code based on runtime signals and to reason over iterative interactions. We incorporate these dialogues into the instruction tuning corpus alongside single-turn samples from *stack-edu*, *the-stack-v2*, *svg-diagrams*, and *CoSyn-400K*. This integration allows models to practice both initial code generation and multi-turn refinement strategies.

## 4 VISPLOTBENCH: MULTI-LANGUAGE BENCHMARK FOR VISUALIZATION CODING AGENTS

**VisPlotBench** is a benchmark for evaluating visualization coding agents across eight languages. Unlike prior efforts that focus on a single language or specification style, VisPlotBench spans imperative libraries, declarative grammars, markup-based formats, and symbolic notations, providing a standardized protocol for assessing both initial code generation and multi-round self-debug.

---

[6]hf.co/datasets/m-a-p/Code-Feedback

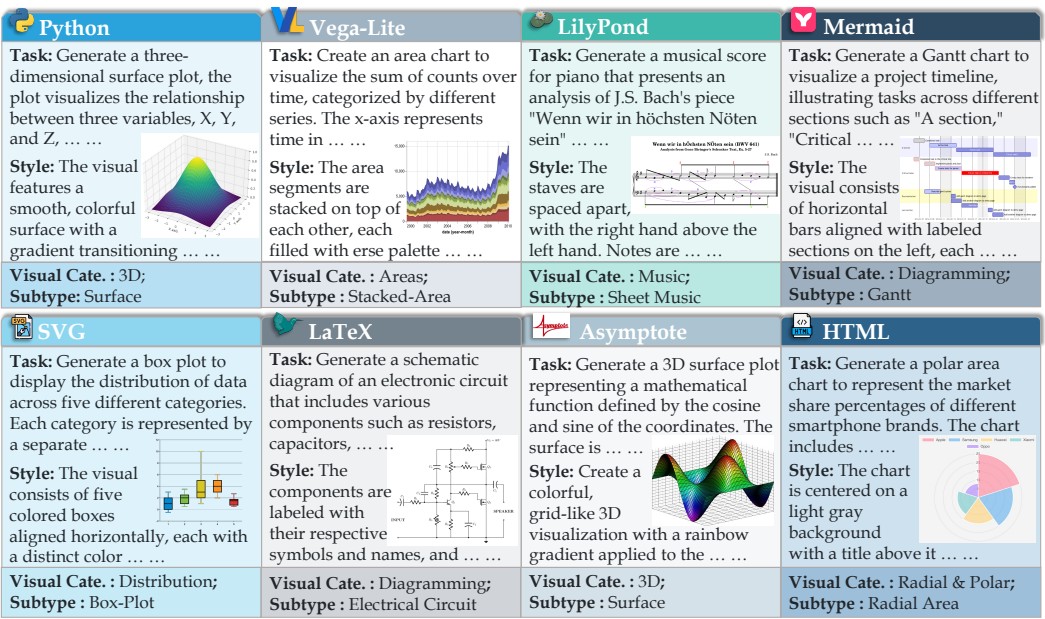

Figure 3: Overview of VisPlotBench. The benchmark covers eight visualization languages and contains 888 diverse visualization tasks, each combining a natural language instruction and a rendered visual. Tasks are annotated with a Visual category and a Subtype, spanning 13 categories in total.

## 4.1 OVERVIEW

Existing visualization benchmarks are narrow in scope: most cover a single language, few chart families, and no iterative debugging. **VisPlotBench** fills these gaps with 888 tasks across eight languages and 13 Visual categories (Figure 4). The taxonomy spans common families such as Bars, Lines, and Scatter, while adding rarely represented ones like Hierarchies, Music, and Networks & Flows. Each task combines a natural language instruction, executable code, and a rendered output, enabling execution-grounded evaluation. With its execute–render–score protocol and multi-round self-debug loop, VisPlotBench provides the first systematic benchmark for assessing visualization coding agents across languages and task types.

Table 1: Comparison with existing benchmarks. VisPlotBench provides executable, multi-language tasks with natural language instructions, rendered outputs, and a standardized protocol for both initial code generation and multi-round self-debugging.

| Benchmark | Coverage | Self-debug | Visual Category | Num |
|---|---|---|---|---|
| VisEval (Chen et al., 2024) | Python | ✗ | 4 | 2,524 |
| MatPlotBench (Yang et al., 2024c) | Python | ✗ | 11 | 100 |
| nvBench (Luo et al., 2021) | Vega–Lite | ✗ | 4 | 25,750 |
| nvBench 2.0 (Luo et al., 2025) | Vega–Lite | ✗ | 5 | 7,878 |
| Text2Vis (Rahman et al., 2025) | Python | ✗ | 10 | 1,985 |
| PandasPlotBench (Galimzyanov et al., 2024) | Python | ✗ | 10 | 175 |
| PandasPlotBench-Enhanced (Ni et al., 2025) | Python | ✔ | 10 | 175 |
| **VisPlotBench (ours)** | 8 languages | ✔ | 13 | 888 |

Table 1 positions VisPlotBench among representative benchmarks across four dimensions: language coverage, visual categories, self-debug support, and dataset size. Earlier resources remain narrow—focusing on `Python` or `Vega-Lite`, with limited chart types and no iterative debugging. VisCoder introduced self-debugging for PandasPlotBench, while VisPlotBench generalizes this to eight languages, expands coverage to 13 categories, including Hierarchies, Music, and Networks & Flows, and standardizes evaluation for systematic cross-language assessment.

## 4.2 DATA COLLECTION AND CURATION

We assemble 888 executable tasks from publicly available examples, library documentation, and high-quality code snippets across eight programming languages. The tasks span 13 Visual categories and 116 Subtypes, covering common families such as Bars, Lines, and Scatter, as well as underrepresented ones including Hierarchies, Music, and Networks & Flows.

Each candidate script is executed in an isolated runtime with language-specific kernels or headless renderers. Tasks are retained only if execution succeeds and a valid image is produced. We discard visually trivial outputs (e.g., near-monochrome images) and remove duplicates by hashing rendered outputs and normalizing code. This process yields a pool of verified code–image pairs compatible with our evaluation pipeline.

Annotators then review verified pairs, removing low-quality items such as unreadable or degenerate plots. Each remaining task is annotated with a Visual category and Subtype from the shared taxonomy shown in Table 7, with library-specific idioms added when appropriate. A double-pass review with conflict resolution ensures consistency across languages.

## 4.3 TASK CONSTRUCTION

Each VisPlotBench task extends the verified code–image pair with a structured natural language instruction. To ensure consistency across languages, we adopt a five-part schema: *Setup → Plot Instruct → Data Instruct → Task Description → Style Description*. This schema provides a unified template that reflects both the semantic intent and the stylistic requirements of each visualization.

*Setup*, *Plot Instruct*, and *Data Instruct* are authored separately for each language so that tasks capture real usage, including syntax constraints, runtime notes, and data access conventions. *Task Description* and *Style Description* are generated with GPT-4.1 conditioned on the verified code and its rendered visual. The *Task Description* specifies the semantic intent and structural elements required for correctness, while the *Style Description* summarizes perceptual attributes such as layout, annotations, label formatting, and color usage. Detailed authoring templates and generation prompts are provided in Appendix D.2 and D.3.

The final instruction is the concatenation of the five components, producing a unified input format across languages. This design enables coding agents to condition on natural language instructions paired with minimal data previews and generate executable code that satisfies both the semantic and stylistic requirements of the task.

## 4.4 EVALUATION PROTOCOL

VisPlotBench adopts a standardized **execute–render–score** pipeline. Each submission is executed in an isolated runtime with language-specific kernels or headless renderers, subject to strict timeouts and log capture. The process outputs three artifacts: rendered image, execution log and metadata record, supporting execution-grounded and judgment-based evaluation.

Evaluation metrics extend those of PandasPlotBench and VisCoder. **Execution Pass Rate** checks whether the code runs without error and produces a valid visualization. **Task Score** measures instruction compliance using an LLM judge guided by semantic and structural rubrics, and **Visual Score** assesses perceptual similarity between generated and reference outputs. Both follow the GPT-based judging protocol of PandasPlotBench.

To assess iterative refinement, VisPlotBench includes a multi-round self-debug protocol. Unresolved tasks are revisited for up to three rounds, where the model receives the instruction, its prior code, and an excerpt of the execution log before producing a revision. The final score reflects the best attempt, mirroring real-world correction loops and enabling systematic evaluation of both baseline generation and feedback-driven recovery.

## 5 MAIN RESULTS

We evaluate both proprietary and open-source models on VisPlotBench to compare execution reliability across parameter scales, programming languages, and evaluation modes. Proprietary refer-

ences include GPT-4.1 (OpenAI, 2025) and its lighter variant GPT-4.1-mini (OpenAI, 2025), while open-source baselines include DeepSeek-Coder (Guo et al., 2024b), DeepSeek-CoderV2 (Zhu et al., 2024), Qwen2.5-Coder (Hui et al., 2024), and VisCoder (Ni et al., 2025). Our VisCoder2 models are trained on VisCode-Multi-679K using Qwen2.5-Coder backbones at 3B, 7B, 14B, and 32B scales. Additional evaluation results on PandasPlotBench (Galimzyanov et al., 2024) and Human-Eval (Chen et al., 2021) are provided in Appendix I.2.

Table 2: Overall execution pass rate (%) of selected models on the VisPlotBench benchmark. The best-performing model in each scale is shown in **bold**, and the second best is underlined.

| Model | Exec Pass Overall | Python (196) | Vega-Lite (129) | LilyPond (55) | Mermaid (131) | SVG (65) | LaTeX (112) | Asymptote (92) | HTML (108) |
|---|---|---|---|---|---|---|---|---|---|
| GPT-4.1 | 63.4 | 64.3 | 84.5 | 43.6 | 68.7 | 95.4 | 31.3 | 21.7 | 89.8 |
| GPT-4.1 + Self Debug | **82.4** | **84.2** | 96.1 | **63.6** | 93.9 | **96.9** | **66.1** | 46.7 | 97.2 |
| GPT-4.1-mini | 58.9 | 64.8 | 84.5 | 16.4 | 51.9 | 95.4 | 29.5 | 23.9 | 86.1 |
| GPT-4.1-mini + Self Debug | 81.1 | 80.6 | **96.9** | 56.4 | **94.7** | **96.9** | 58.9 | **48.9** | **100.0** |
| ~ 3B Scale | | | | | | | | | |
| DeepSeek-Coder-1.3B-Ins. | 32.3 | 29.1 | 53.5 | 30.9 | 63.4 | 7.7 | 4.5 | 13.0 | 36.1 |
| Qwen2.5-Coder-3B-Ins. | 45.8 | 34.2 | 68.2 | 3.6 | 74.1 | 75.4 | 17.9 | 18.5 | 62.0 |
| VisCoder-3B | 56.1 | 45.4 | 83.7 | 21.8 | 75.6 | 76.9 | 23.2 | 30.4 | 79.6 |
| **VisCoder2-3B** | 67.7 | 56.1 | 83.0 | 50.9 | **76.3** | **87.7** | 36.6 | 62.0 | 93.5 |
| **VisCoder2-3B + Self Debug** | **70.0** | **63.3** | **84.5** | **52.7** | **76.3** | **87.7** | **38.4** | **63.0** | **94.4** |
| ~ 7B Scale | | | | | | | | | |
| DeepSeek-Coder-6.7B-Ins. | 46.4 | 39.3 | 79.8 | 7.3 | **91.6** | **96.9** | 18.8 | 0.0 | 22.2 |
| Qwen2.5-Coder-7B-Ins. | 51.2 | 41.3 | 76.0 | 5.5 | 77.9 | 92.3 | 25.9 | 13.0 | 64.8 |
| VisCoder-7B | 57.2 | 58.2 | 71.3 | 23.6 | 77.1 | 93.9 | 25.9 | 17.4 | 75.9 |
| **VisCoder2-7B** | 70.9 | 64.8 | 83.0 | 69.1 | 78.6 | **96.9** | 39.3 | 64.1 | 82.4 |
| **VisCoder2-7B + Self Debug** | **76.4** | **77.0** | **84.5** | **72.7** | 84.7 | **96.9** | **42.9** | **70.7** | **84.3** |
| ~ 14B Scale | | | | | | | | | |
| DeepSeek-Coder-V2-Lite-Ins. | 55.3 | 47.5 | 75.2 | 49.1 | 69.5 | 93.9 | 29.5 | 20.7 | 64.8 |
| Qwen2.5-Coder-14B-Ins. | 59.5 | 50.0 | 83.0 | 25.5 | 74.8 | **98.5** | 30.4 | 25.0 | 83.3 |
| **VisCoder2-14B** | 72.1 | 65.3 | 93.0 | 54.6 | 81.7 | 89.2 | 42.0 | 56.5 | 90.7 |
| **VisCoder2-14B + Self Debug** | **78.4** | **78.1** | **94.6** | **63.6** | **86.3** | 90.8 | **45.5** | **66.3** | **94.4** |
| ~ 32B Scale | | | | | | | | | |
| DeepSeek-Coder-33B-Ins. | 54.3 | 58.2 | 90.7 | 30.9 | 87.0 | 92.3 | 24.1 | 21.7 | 12.0 |
| Qwen2.5-Coder-32B-Ins. | 57.5 | 50.5 | 83.0 | 30.9 | 71.0 | 93.9 | 29.5 | 17.4 | 78.7 |
| **VisCoder2-32B** | 73.1 | 65.3 | 94.6 | 56.4 | 87.0 | 81.5 | 42.9 | 58.7 | 91.7 |
| **VisCoder2-32B + Self Debug** | **82.4** | **81.6** | **96.1** | **69.1** | **90.1** | 86.2 | **61.6** | **71.7** | **93.5** |

## 5.1 OVERALL COMPARISON

Table 2 summarizes execution pass rates for all models across eight visualization languages and overall averages. The following analysis examines differences between proprietary and open-source models, variation across languages, and the relative advantages of VisCoder2 under both default and self-debug evaluation modes.

**Proprietary Models Remain Stronger.** GPT-4.1 achieves 63.4% overall, the highest among reference models, and GPT-4.1-mini follows closely. Both perform strongly on standardized declarative or markup languages such as `Vega-Lite`, `SVG`, and `HTML`, all above 84%. In contrast, instruction-tuned open-source models remain far behind. At the 7B scale, Qwen2.5-Coder reaches only 51.2% overall, with fewer than 30% on `LaTeX` and just 5.5% on `LilyPond`. Previous VisCoder variants improve `Python` performance but fail to generalize across languages. These results underline the substantial gap between proprietary and open-source models.

**Cross-Language Variation.** Performance differs sharply across visualization languages. `Vega-Lite` and `HTML` are close to saturation for most models, while `Python` shows steady gains with scale. By contrast, symbolic and compiler-dependent languages remain the most difficult. Even GPT-4.1 achieves less than 45% on `LilyPond` and under 25% on `Asymptote`, and open-source baselines fall much lower. This uneven landscape highlights that progress on symbolic grammars is the key bottleneck for reliable multi-language visualization.

**VisCoder2 Advantage.** Across all scales, VisCoder2 consistently outperforms size-matched open-source baselines. At 32B, it improves overall execution pass rate by approximately 15 points compared with Qwen2.5-Coder and reaches parity with GPT-4.1. The only consistent shortfall is on `SVG`, where VisCoder2 trails the strongest baseline by over 10 points. Overall, VisCoder2 is the first open-source model to match proprietary reliability on executable visualization tasks.

**Effect of Self-Debug.** Iterative correction consistently improves execution reliability across model families and scales. Proprietary models benefit strongly, and VisCoder2 follows the same trend: at larger scales, overall execution rises by nearly ten points when self-debugging is enabled. The effect is especially pronounced for symbolic and compiler-dependent languages such as `LilyPond`, `LaTeX`, and `Asymptote`, where fragile syntax or compilation errors dominate. Self-debugging enables the model to repair these shallow but frequent failures, allowing models to resolve previously intractable failures into valid outputs. This demonstrates that feedback-driven refinement is not just a marginal improvement but a critical mechanism for tackling the hardest visualization languages.

## 5.2 TASK AND VISUAL SCORE ANALYSIS

We analyze Task Score and Visual Score on three representative languages that highlight different behaviors, as shown in Table 3: `LaTeX` illustrates execution–semantics mismatch, `LilyPond` shows the largest gains on symbolic grammars, and `SVG` exposes model–library sensitivity where semantic and perceptual signals diverge. Results for all languages and scales are provided in Appendix F. A detailed analysis of self-debug behavior and Task/Visual scores is presented in Appendix I.3.

Table 3: Performance of selected languages on the VisPlotbench benchmark. For each model, we report (1) execution pass rate (**Exec Pass**), (2) mean visual and task scores (**Mean**), and (3) the proportion of samples scoring at least 75 (**Good**). The best-performing model in each scale is shown in **bold**, and the second best is underlined.

| Model | LaTeX (112) | | | | | | LilyPond (55) | | | | | | SVG (65) | | | | | |
| | Exec Pass | Mean | | Good(≥75) | | | Exec Pass | Mean | | Good(≥75) | | | Exec Pass | Mean | | Good(≥75) | | |
| | | vis | task | vis | task | | | vis | task | vis | task | | | vis | task | vis | task | |
| GPT-4.1 | 31.3 | 18 | 26 | 13% | 25% | | 43.6 | 14 | 38 | 5% | 36% | | 95.4 | 45 | 92 | 14% | 94% | |
| GPT-4.1 + Self Debug | **66.1** | 38 | 56 | 25% | 51% | | **63.6** | 17 | 54 | 5% | 53% | | **96.9** | 45 | 93 | 14% | 95% | |
| GPT-4.1-mini | 29.5 | 21 | 25 | 18% | 25% | | 16.4 | 2 | 12 | 0% | 11% | | 95.4 | 41 | 86 | 11% | 86% | |
| GPT-4.1-mini + Self Debug | 58.9 | 35 | 50 | 23% | 49% | | 56.4 | 14 | 42 | 0% | 35% | | **96.9** | 42 | 88 | 11% | 88% | |
| Qwen2.5-Coder-7B-Instruct | 25.9 | 11 | 15 | 6% | 8% | | 5.5 | 0 | 3 | 0% | 4% | | 92.3 | 23 | 58 | 0% | 40% | |
| **VisCoder2-7B** | 39.3 | 15 | 23 | 6% | 15% | | 69.1 | 16 | 52 | 2% | 45% | | **96.9** | 34 | 73 | 3% | 62% | |
| **VisCoder2-7B + Self Debug** | 42.9 | 16 | 24 | 6% | 15% | | **72.7** | 17 | 55 | 2% | 45% | | **96.9** | 34 | 73 | 3% | 62% | |
| Qwen2.5-Coder-32B-Instruct | 29.5 | 14 | 25 | 9% | 27% | | 30.9 | 5 | 22 | 2% | 18% | | **93.9** | 34 | 81 | 3% | 75% | |
| **VisCoder2-32B** | 42.9 | 20 | 35 | 11% | 34% | | 56.4 | 14 | 39 | 2% | 27% | | 81.5 | 33 | 68 | 11% | 63% | |
| **VisCoder2-32B + Self Debug** | **61.6** | 28 | 45 | 14% | 42% | | **69.1** | 16 | 48 | 2% | 35% | | 86.2 | 34 | 71 | 11% | 66% | |

**LaTeX: Execution–Semantics Mismatch.** Models often capture the intended structure of a figure but fail to compile reliably. For example, GPT-4.1 improves from 31.3% to 66.1% execution pass rate with Self-Debug, while task scores remain around 50 even when execution fails. VisCoder2 raises execution and task scores compared with baselines, but compilation errors remain frequent. This pattern indicates that semantic alignment does not always translate into successful rendering.

**LilyPond: Symbolic Grammar Gains.** VisCoder2 delivers the clearest advantage on symbolic languages. At 7B, Qwen2.5-Coder executes only 5.5% of tasks, while VisCoder2 reaches 69.1% and further improves with Self-Debug. The proportion of examples with task scores above 75 also increases by more than tenfold. These results show that targeted coverage of symbolic grammars in VisCode-Multi-679K translates directly into reliable generation and semantic adherence.

**SVG: Sensitivity to Rendering Libraries.** Execution success is high across most models, yet visual scores lag behind task scores. For instance, GPT-4.1 with Self-Debug achieves 95.4% execution and a task score near 90, but the average visual score is below 50. VisCoder2 performs competitively but trails Qwen2.5 on execution at larger scales (81.5% versus 93.9% at 32B). These discrepancies suggest that evaluation on `SVG` is strongly influenced by library-specific rendering details rather than semantic understanding alone.

## 5.3 ERROR ANALYSIS

To better understand failure modes across languages, we analyze execution errors before and after self-debug. Many language-specific exceptions, such as FunctionSignatureError in `Asymptote` or MarkupError in `LilyPond`, were merged into four broader categories for clarity: Structural Errors (syntax or parsing), Type&Interface Errors (invalid calls or arguments), Semantic / Data Errors (mismatched variables or values), and Runtime / Environment Errors (renderer or package issues). Representative results for VisCoder2-32B are shown in Table 4, with full breakdowns in Appendix H.

Table 4: Representative error transitions for VisCoder2-32B across eight visualization languages. Each cell shows error counts from initial failure to the final self-debug round (X → Y). A dash indicates the error type does not occur for that language.

| Error Category | Python | Vega-Lite | LilyPond | Mermaid | SVG | LaTeX | Asymptote | HTML |
|---|---|---|---|---|---|---|---|---|
| Structural Errors | 1 → 1 | 2 → 1 | 14 → 10 | 12 → 9 | 8 → 7 | 10 → 4 | 9 → 3 | - |
| Type & Interface | 13 → 3 | 2 → 1 | 5 → 2 | - | - | - | - | - |
| Semantic / Data | 19 → 8 | - | - | - | - | 28 → 23 | 15 → 11 | - |
| Runtime / Env. | - | 2 → 2 | - | - | - | 27 → 6 | 8 → 6 | 3 → 2 |

**Effective recovery on structural and interface errors.** Self-debug reduces shallow errors such as missing tokens or invalid arguments across multiple languages. For example, `Python` interface errors fall from 13 to 3 (Figure 6), and structural errors in `LilyPond` decrease from 14 to 10 (Figure 12). `Mermaid` and `Asymptote` show the same trend, with syntax and function signature errors shrinking after correction (Figure 15). These cases benefit from explicit diagnostic traces, making them relatively easy to fix through iterative feedback.

**Persistent failures in semantic and runtime errors.** Errors involving semantics or execution environments remain difficult to resolve. In `LaTeX`, undefined variables decrease only slightly (28 to 23), and `Asymptote` variable mismatches improve only marginally (15 to 11) (Figure 24). Renderer failures such as `Vega-Lite` rendering errors (2 to 2) and `HTML` request failures (3 to 2) often persist across all rounds (Figure 28). These errors require deeper reasoning over symbolic grammars and runtime contexts, which current self-debug protocols cannot fully capture. Symbolic languages and renderer-sensitive environments therefore remain the dominant bottlenecks, pointing to the need for grammar-aware training objectives and more robust runtime integration.

## 6 CONCLUSION

Reliable visualization coding goes beyond single-pass generation: it requires competence across diverse languages and the ability to refine outputs iteratively in response to execution feedback. Existing datasets and benchmarks lack these capabilities, limiting progress toward practical agents for real-world workflows.

We addressed these gaps through three contributions. First, we introduced VisCode-Multi-679K, a large-scale instruction tuning dataset that unifies executable visualization code across twelve languages with multi-turn feedback dialogues. Second, we built VisPlotBench, a benchmark covering eight visualization languages under a standardized execute–render–score protocol, with tasks spanning 13 categories and 116 subtypes. Third, we trained the VisCoder2 model family on these resources, showing that it consistently outperforms open-source baselines and approaches proprietary models in execution reliability.

Our experiments highlight two insights. Broad multi-language coverage is essential: symbolic and compiler-dependent languages such as `LaTeX`, `LilyPond`, and `Asymptote` remain challenging, yet progress on them is decisive for true generalization. Iterative refinement further proves indispensable: self-debug delivers large gains across models, especially on languages where structural and semantic errors are common.

Taken together, VisCode-Multi-679K, VisPlotBench, and VisCoder2 establish the first systematic framework for building and evaluating visualization coding agents. We believe these resources can accelerate the development of agents that are not only multi-language but also capable of realistic correction loops, pushing toward reliable coding assistants for data analysis, reporting, and beyond.

## LIMITATIONS

VisCode-Multi-679K and VisPlotBench take a step toward more reliable multi-language visualization coding agents, but several limitations remain. First, the training corpus is imbalanced across languages: high-resource ecosystems such as `Python` and `Vega-Lite` are well represented, whereas symbolic and domain-specific languages have far fewer samples, which may bias models toward dominant languages. Second, VisPlotBench currently covers eight visualization languages; extending it to additional frameworks and languages would provide broader coverage and enable more comprehensive evaluation.

## ETHICS STATEMENT

This work complies with the ICLR Code of Ethics. In this study, no human subjects or animal experimentation were involved. All datasets used were accessed and used under their licenses/terms of use, in line with applicable usage guidelines and with no infringement of privacy. We took steps to minimize unfair bias or discriminatory effects throughout the research process. No personally identifiable information was processed, and no experiments were conducted that could raise privacy or security risks. We remain committed to transparency and research integrity across the entire project lifecycle.

## REPRODUCIBILITY STATEMENT

We have made every effort to ensure that the results presented in this paper are reproducible. We release all prompts used to construct the training datasets in Appendix D.1, the full processing pipeline applied to raw sources, and the prompts and instructions used in building the *VisPlotBench* benchmark in Appendix D.2 and subsection D.3. We also provide a detailed description of the experimental setup in Appendix A.

In addition, all datasets used (`stack-edu`, `the-stack-v2-train-smol-ids`, `svg-diagrams`, and `Code-Feedback`) in constructing *VisCode-Multi-679K* are publicly available, as described in subsection 3.1. We will open-source our complete training and evaluation codebase, along with all trained model weights after publication. This release will include comprehensive documentation and usage examples to support future research and enable direct comparison with our results.

We believe these measures will enable other researchers to reproduce our findings, trace our dataset construction pipeline, and extend our work in future studies.

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

# Table of Contents in Appendix

## A    EXPERIMENT SETUP

**Training Setup.**    We fine-tune Qwen2.5-Coder-Instruct (Hui et al., 2024) at four parameter scales: 3B, 7B, 14B, and 32B. This setup allows us to assess the generalizability of VisCode-Multi-679K across capacities. All models are trained for 3 epochs with a learning rate of $5 \times 10^{-6}$, a warm-up ratio of 0.05, and a cosine scheduler. We perform full-parameter tuning in `bfloat16` precision on 8×H100 GPUs with a total batch size of 64, using the SWIFT infrastructure (Zhao et al., 2024).

**Evaluation Setup.**    All evaluations are conducted on VisPlotBench using the standardized protocol in Section 4.4. We report three metrics: Execution Pass Rate, Task Score, and Visual Score (detailed in Appendix I.1), capturing executability, semantic alignment, and perceptual similarity. Models are also tested under the self-debug protocol with up to three rounds of correction based on execution feedback, assessing both baseline generation and recovery through iterative refinement.

## B    TRAINING DATA DISTRIBUTION

Table 5: Distribution of visualization code samples across languages and sources. The final column reports per-language totals, and the final row reports per-source totals.
[Back to Appendix Contents]

| Language | CoSyn-400K | the-stack-v2 | svg-diagrams | Total |
|---|---|---|---|---|
| Python | 66,052 | 120,902 | - | 186,954 |
| HTML | 75,315 | 59,915 | - | 135,230 |
| LaTeX | 124,039 | - | - | 124,039 |
| SVG | 2,693 | - | 43,928 | 46,621 |
| JavaScript | - | 28,807 | - | 28,807 |
| Asymptote | 22,539 | - | - | 22,539 |
| C++ | - | 16,776 | - | 16,776 |
| R | - | 13,437 | - | 13,437 |
| Mermaid | 13,381 | - | - | 13,381 |
| LilyPond | 12,093 | - | - | 12,093 |
| Vega-Lite | 6,790 | - | - | 6,790 |
| TypeScript | | 6,315 | - | 6,315 |
| **Total** | **322,902** | **246,152** | **43,928** | **612,982** |

# C ABLATION STUDY

To disentangle the contribution of each data source, we conduct a controlled ablation study using Qwen2.5-Coder-7B as the base model. Separate models are fine-tuned on individual subsets of `The-Stack-V2`, `CoSyn`, `StarVector`, and `Code-Feedback`, under the same instruction-tuning setup as the full configuration. We report execution pass rates on VisPlotBench in both default and self-debug modes, with comparisons to the untuned Qwen2.5-Coder-7B baseline and the full VisCode-Multi-679K model (Table 6).

Table 6: Execution pass rates of Qwen2.5-Coder-7B models trained on individual subsets of VisCode-Multi-679K. Each model is evaluated under both default (✗) and self-debug (✔) modes.

| Model | Self-Debug | Overall | Python | Vega-Lite | LilyPond | Mermaid | SVG | LaTeX | Asymptote | HTML |
|---|---|---|---|---|---|---|---|---|---|---|
| Qwen2.5-Coder-7B-Ins. | ✗ | 51.2 | 41.3 | 76.0 | 5.5 | 77.9 | 92.3 | 25.9 | 13.0 | 64.8 |
|  | ✔ | 59.0 | 61.7 | 77.5 | 5.5 | 79.4 | 92.3 | 30.4 | 20.7 | 76.9 |
| + The-Stack-V2-246K | ✗ | 49.0 | 47.5 | 81.4 | 7.3 | 69.5 | 84.6 | 0.9 | 17.4 | 64.8 |
|  | ✔ | 56.5 | 58.2 | 83.7 | 10.9 | 73.3 | 84.6 | 31.3 | 18.5 | 65.7 |
| + CoSyn-323K | ✗ | 59.2 | 25.5 | 83.7 | 65.5 | 57.3 | 100.0 | 36.6 | 56.5 | 91.7 |
|  | ✔ | 62.2 | 31.1 | 84.5 | 69.1 | 61.1 | 100.0 | 38.4 | 62.0 | 91.7 |
| + StarVector-44K | ✗ | 40.1 | 43.4 | 72.1 | 5.5 | 67.9 | 16.9 | 10.7 | 13.0 | 47.2 |
|  | ✔ | 44.5 | 53.6 | 73.6 | 7.3 | 70.2 | 18.5 | 13.4 | 19.6 | 50.0 |
| + Code-Feedback-66K | ✗ | 55.2 | 47.5 | 78.3 | 20.0 | 81.7 | 92.3 | 27.7 | 17.4 | 65.7 |
|  | ✔ | 63.1 | 62.2 | 80.6 | 21.8 | 81.7 | 92.3 | 38.4 | 23.9 | 83.3 |
| + Full VisCode-Multi-679K | ✗ | 70.9 | 64.8 | 83.0 | 69.1 | 78.6 | 96.9 | 39.3 | 64.1 | 82.4 |
|  | ✔ | 76.4 | 77.0 | 84.5 | 72.7 | 84.7 | 96.9 | 42.9 | 70.7 | 84.3 |

**Natural vs. Synthetic.** Training on `The-Stack-V2` alone yields limited improvements and even degrades symbolic languages such as `LaTeX`, reflecting the sparsity of clean visualization signals in general-purpose code. By contrast, `CoSyn` delivers large gains on symbolic and grammar-sensitive languages, with execution rates on `LilyPond` and `Asymptote` rising by over 60 points compared to the baseline. This contrast shows that large-scale synthetic data provides valuable structural coverage that complements natural code.

**Domain vs. Multi-turn.** The `StarVector` subset contributes primarily to `SVG` but is too small to improve overall performance. In contrast, `Code-Feedback` does not drastically shift baseline pass rates but produces consistent gains under self-debug, lifting overall execution from 55.2% to 63.1%. This demonstrates that multi-turn dialogue data provides critical supervision for recovery through iterative correction, rather than improving one-shot generation.

**Full Dataset Synergy.** Combining all subsets yields the strongest model. With VisCode-Multi-679K, the overall pass rate reaches 70.9% in default mode and 76.4% with self-debug, substantially surpassing both the untuned baseline and any single-source variant. These results confirm that the dataset's diverse composition—balancing natural, synthetic, domain-specific, and iterative data—is essential for building robust multi-language visualization coding agents.

# D PROMPT USED AND INSTRUCT DESIGN

In this section, we present the prompts used during the construction of VisCode-Multi-679K and VisPlotBench.

## D.1 PROMPT USED IN VISCODE-MULTI-679K

---

**Code Extraction Prompt**

**Model: GPT-4.1-mini**

```
# LANGUAGE= [Python, JavaScript, TypeScript, C++, R, HTML]
# LANG_BULLET = {
```
`Python`: Write a single `.py` file. No external files or internet access. Use helper libraries only if truly required for the chart.
`JavaScript`: Write a single `.js` file. Add necessary imports for any required libraries.
`TypeScript`: Write a single `.ts` file. Use ES6 module syntax (import/export). The code should not require a module bundler.
`C++`: Write a single `.cpp` file with `main()`. Program must exit automatically. Do not load external assets. If linking is needed, add one build command as a comment.
`R`: Write a single `.R` file. Use `library(Library)`. Create mock data if needed. No external files.
`HTML`: Write a single HTML file. Include a `<script>` tag and the needed DOM element for the chart. All code runs in the browser.
```
}
```

You are a {`LANGUAGE`} visualization code extraction agent.

Given a {`LANGUAGE`} code snippet and the used library, your task is to extract a **minimal yet runnable** {`LANGUAGE`} snippet that reflects how the library is actually used **for visual output**.
Guidelines: - Keep only the logic needed for the **visual output**; remove unrelated code.
- If the library is **not used for rendering/drawing/plotting** in Code, return "null".
- If inputs/assets are missing, create **semantically relevant** mock data that makes the output meaningful.
- Preserve the **main intent**, API pattern, key parameters, and style** from the original code; simplify when it improves clarity, and avoid adding new wrappers or layers that are not essential.
- Make the **visual output** clear and professional: use appropriate visual cues (titles/labels/legends) when applicable); keep layout readable.
- Ensure the snippet runs standalone in a minimal environment and **terminates automatically** (no user input required).
- {`LANG_BULLET`}
- If the library is unused or information is insufficient, return "null".

Used Library: {`used_libs`}
Code: {`code`}

---

---

**Instruct Generation Prompt: `the-stack-v2` & `svg-diagrams`**

**Model: GPT-4.1**

`# LANGUAGE=[Python, JavaScript, TypeScript, C++, R, HTML, SVG]`

You are given a {`LANGUAGE`} code snippet that renders an image. A rendered image of the resulting output is provided at the end. Your task is to infer and clearly describe the purpose, structure, and style of this image.

Break your response into the following five parts:
1. Setup (state the language and rendering context, including any tools or libs implied).
2. Data/Visual Description
- If the code is data-driven: summarize the inputs the code relies on and any shaping operations.
- If the code is not data-driven: summarize the visible content of the image.
3. Data Generation (the data-generation lines copied verbatim, or "None" if not applicable).
4. Output Description (omit language constructs; start with "Generate..." or "Create...", and describe the final image conceptually).
5. Style Description (describe appearance and layout without naming language constructs).

Each part must start on a new line, numbered 1 through 5.
Use plain text only; no markdown.

Code:
{`code`}

Image:

---

**Instruct Generation Prompt: `CoSyn-400K`**

**Model: GPT-4.1**

`# FOR DATA-DRIVEN LANGUAGES`
`# LANGUAGE=[Python, Vega-Lite, HTML, LilyPond, Mermaid]`

You are given a {`LANGUAGE`} code snippet that produces a rendered visual. A rendered image of the resulting output is provided at the end. Your task is to infer and clearly describe the purpose and structure of this visual.

Break your response into the following four parts:
1. Setup (state the {`LANGUAGE`} and its rendering context, including any tools or specification frameworks implied).
2. Data/Content Description (summarize the input fields, entities, or content the code relies on, including any shaping or transformation operations).
3. Output Description (omit library, directive, or element names; start with "Generate..." or "Create...", and describe the visual conceptually).
4. Style Description (describe appearance and layout without naming language constructs).

Each part must start on a new line, numbered 1 through 4.
Use plain text only; no markdown.

Code:
{`code`}

Image:

---

**Instruct Generation Prompt: CoSyn-400K**

**Model: GPT-4.1**

```
# FOR NONE DATA-DRIVEN LANGUAGES
# LANGUAGE=[Asymptote, SVG]
```

You are given a {LANGUAGE} code snippet that renders an image. A rendered image of the resulting output is provided at the end. Your task is to infer and clearly describe the purpose and structure of this image.

Break your response into the following four parts:
1. Setup (state the {LANGUAGE} and its rendering context).
2. Visual Elements (summarize the visible components of the image).
3. Output Description (omit language constructs; start with "Generate..." or "Create...", and describe the image conceptually).
4. Style Description (describe appearance and layout without naming language constructs).

Each part must start on a new line, numbered 1 through 4.
Use plain text only; no markdown.

Code:
{code}

Image:

---

### D.2 PROMPT USED IN VISPLOTBENCH

---

**Task & Style Description Generation Prompt:**

**Model: GPT-4.1**
`# LANGUAGE=` `[Python, Vega-Lite, HTML, LilyPond, Mermaid, Asymptote, HTML, SVG]`

You are given a `{LANGUAGE}` code that produces a rendered visual. A rendered image of the resulting output is provided at the end. Your task is to infer and clearly describe the purpose and structure of this visual.

Break your response into the following two parts: 1. Task Description (omit libraries and specific function names at this part, start with "Generate..." or "Create..."). 2. Style Description (describe appearance and layout without using specification keywords).

Each part must start on a new line, numbered 1 through 2. Use plain text only; no markdown.

CODE: code
IMAGE:

---

**Vis & Task Judge Prompt**

`# Visual Judge`
You are an excellent judge at evaluating visualization plots between a model generated plot and the ground truth.

You will be giving scores on how well it matches the ground truth plot.

The generated plot will be given to you as the first figure.

Another plot will be given to you as the second figure, which is the desired outcome of the user query, meaning it is the ground truth for you to reference.

Please compare the two figures head to head and rate them.

Suppose the second figure has a score of 100, rate the first figure on a scale from 0 to 100.

Scoring should be carried out in the following aspect:

Plot correctness: compare closely between the generated plot and the ground truth, the more resemblance the generated plot has compared to the ground truth, the higher the score. The score should be proportionate to the resemblance between the two plots.

Ignore color matching. If the plots present the same information but are made in different colors, consider them matching. Capture the resemblance of the main idea of the plot.

Only rate the first figure, the second figure is only for reference.

After scoring from the above aspect, please give a final score. Do not write anything else. The final score is preceded by the [FINAL SCORE] token.

For example [FINAL SCORE]: 40

`# Task Judge`
You are an excellent judge at evaluating visualization plot according to the given task.

You will be giving scores on how well plot image matches the task.

The generated plot will be given to you as an image.

Please score how well plot matches the task. Score it on a scale from 0 to 100.

Scoring should be carried out in the following aspect:

Task adherence: how the plot corresponds to the task given below (begins from [PLOT TASK] token).

After scoring from the above aspect, please give a final score. Do not write anything else. The final score is preceded by the [FINAL SCORE] token.

For example [FINAL SCORE]: 40

---

## D.3 INSTRUCT DESIGN IN VISPLOTBENCH EVALUATION

---

**Python Instruct**

```
# SYSTEM PROMPT:
```
You are a helpful programming assistant proficient in Python. All answers must be enclosed in a block
"'python SOME CODE"' containing one complete Python code. Example minimal spec: "'python

```
 print('hello world')
```
"'

```
# SETUP INSTRUCT:
```
Use Python programming language. Import essential libraries. The essential libraries needed are
pandas for managing dataframes and [USED_LIB] with its subsidiary libraries for plotting. Ensure
importing numpy as np and scipy if they are used in program. DO NOT use or import other
visualization libraries.

```
# PLOT INSTRUCT:
```
Write a code to build a plot of dataframe according to following instructions. Write a code that returns
plot, not just function declaration. Do not write explanations, just a code enclosed in codeblock.
Important reasoning write in comments to the code. Make sure that all used libraries and functions are
imported.

```
# DATA INSTRUCT:
```
Load df dataframe by single line df = pd.read_csv("data.csv"). DO NOT alter df dataframe columns or
add columns. This df dataframe should remain intact. The metadata of the dataframe is following:

---

**Vega-Lite Instruct**

```
# SYSTEM PROMPT:
```
You are a helpful programming assistant proficient in Vega-Lite. All answers must be enclosed in a
block "'vegalite SOME CODE"' containing one complete Vega-Lite specification. Example minimal
spec: "'vegalite

```
{
    "$schema":"https://vega.github.io/schema/vega-lite/v6.json",
    "data":{"values":[{"hello":"world"}]},
    "mark":"text",
    "encoding":{"text":{"field":"hello","type":"nominal"}}
}
```
"'

```
# SETUP INSTRUCT:
```
Setup. Use the Vega-Lite v6 JSON schema and produce exactly one valid Vega-Lite specification as
a single top-level JSON object that MUST include the $schema property. Do not output raw Vega
specifications, imperative code, language-specific wrappers, or references to other plotting libraries;
use only Vega-Lite encodings, transforms, and configuration required by the plot.

```
# PLOT INSTRUCT:
```
Write a code to build a plot of the dataset according to the following instructions. Return one complete
Vega-Lite JSON specification enclosed in a single code block, and do not include explanations or
comments. Use only the constructs permitted by the setup, ensure that all referenced field names
exactly match the dataset metadata, and do not rename or drop columns; only use non-destructive
Vega-Lite transforms if required.

```
# DATA INSTRUCT:
```
Data description. Load the dataset by setting "data": "url": "data.csv". Do not create synthetic data or
load inline data values. The metadata of the dataset is following:

---

**Mermaid Instruct**

```
# SYSTEM PROMPT:
```
You are a helpful programming assistant proficient in Mermaid diagrams. All answers must be enclosed in a block "'mermaid SOME CODE"' containing one complete Mermaid specification. Example minimal spec: "'mermaid

```
graph TD
    A[Start] -\rightarrow B{Condition}
    B -\rightarrow|Yes| C[Do something]
    B -\rightarrow|No| D[Stop]
"'
```

```
# SETUP INSTRUCT:
```
Setup. Use Mermaid syntax only and produce exactly one valid Mermaid diagram definition. Do not output explanations, comments outside the code block, or code in other languages or formats. Do not split the diagram into multiple blocks. Ensure the code can be rendered directly by mermaid-cli (mmdc).

```
# PLOT INSTRUCT:
```
Write a diagram according to the following instructions. Do not include explanations or natural language outside the code block. Ensure that the diagram is self-contained, syntactically correct Mermaid code, and does not rely on external data or libraries. Use node and edge labels exactly as provided in the instructions.

```
# DATA INSTRUCT:
```
Data description. The diagram is constructed only from the provided instructions. Do not load external files or datasets.

**LilyPond Instruct**

```
# SYSTEM PROMPT:
```
You are a helpful programming assistant proficient in LilyPond. Always use the version statement `\version "2.22.1"`. All answers must be enclosed in a block "'lilypond SOME CODE"' containing one complete LilyPond score. Example minimal spec: "'lilypond

```
\version "2.22.1"
\score {
    \new Staff { c' d' e' f' }
    \layout { }
}
"'
```

```
# SETUP INSTRUCT:
```
Setup. Use LilyPond syntax only and produce exactly one valid LilyPond music notation definition. Do not output explanations, comments outside the code block, or code in other languages or formats. Do not split the notation into multiple blocks. Ensure the code can be rendered directly by LilyPond.

```
# PLOT INSTRUCT:
```
Write a music notation according to the following instructions. Do not include explanations or natural language outside the code block. Ensure that the notation is self-contained, syntactically correct LilyPond code, and does not rely on external data or libraries. Use note names and other musical symbols exactly as provided in the instructions.

```
# DATA INSTRUCT:
```
Data description. The music notation is constructed only from the provided instructions. Do not load external files or datasets.

**SVG Instruct**

```
# SYSTEM PROMPT:
```
You are a helpful programming assistant proficient in SVG. All answers must be enclosed in a block "`svg SOME CODE`" containing one complete SVG specification. Example minimal spec: "`svg

```
<svg width="100" height="100" xmlns="http://www.w3.org/2000/svg">
  <circle cx="50" cy="50" r="40" stroke="black"
    stroke-width="2" fill="red" />
</svg>
```
"`

```
# SETUP INSTRUCT:
```
Setup. Use SVG syntax only and produce exactly one valid SVG definition. Do not output explanations, comments outside the code block, or code in other languages or formats. Do not split the diagram into multiple blocks. Ensure the code can be rendered directly by SVG viewers.

```
# PLOT INSTRUCT:
```
Write an SVG according to the following instructions. Do not include explanations or natural language outside the code block. Ensure that the SVG is self-contained, syntactically correct SVG code, and does not rely on external data or libraries. Use shapes and attributes exactly as provided in the instructions.

```
# DATA INSTRUCT:
```
Data description. The SVG is constructed only from the provided instructions. Do not load external files or datasets.

**Asymptote Instruct**

```
# SYSTEM PROMPT:
```
You are a helpful programming assistant proficient in Asymptote. All answers must be enclosed in a block "`asymptote SOME CODE`" containing one complete Asymptote specification. Example minimal spec: "`asymptote

```
import graph;
size(100);
draw((0,0)--(1,1));
```
"`

```
# SETUP INSTRUCT:
```
Setup. Use Asymptote syntax only and produce exactly one valid Asymptote definition. Do not output explanations, comments outside the code block, or code in other languages or formats. Do not split the diagram into multiple blocks. Ensure the code can be rendered directly by Asymptote.

```
# PLOT INSTRUCT:
```
Write an Asymptote according to the following instructions. Do not include explanations or natural language outside the code block. Ensure that the Asymptote is self-contained, syntactically correct Asymptote code, and does not rely on external data or libraries. Use shapes and attributes exactly as provided in the instructions.

```
# DATA INSTRUCT:
```
Data description. Please use the provided data definition code to construct the plot. Do not modify this code or create data by yourself. Use the variables defined in this code directly when building the plot. The data definition code is as follows:

## LaTeX Instruct

# SYSTEM PROMPT:
You are a helpful programming assistant proficient in LaTeX. All answers must be enclosed in a block "'latex SOME CODE"' containing one complete LaTeX document. Example minimal spec: "'latex

```
\documentclass{standalone}
\begin{document}
Hello
\end{document}
```
"'

# SETUP INSTRUCT:
Setup. Use LaTeX syntax only and produce exactly one valid LaTeX document as a single code block. Do not output explanations, comments outside the code block, or code in other languages or formats. Ensure the document can be rendered directly by LaTeX compilers.

# PLOT INSTRUCT:
Write a LaTeX code to build a plot of the dataset according to the following instructions. Return exactly one complete LaTeX document enclosed in a single code block. Include all required packages. Do not include explanations or comments. Do not create synthetic data or modify the dataset.

# DATA INSTRUCT:
Data description. Load the dataset by adding \pgfplotstablereadlatex.csv\datatable. Do not create synthetic data or modify the dataset. The metadata of the dataset is following:

## HTML Instruct

# SYSTEM PROMPT:
You are a helpful programming assistant proficient in HTML. All answers must be enclosed in a block "'html SOME CODE"' containing one complete HTML document Example minimal spec: "'html

```
<!DOCTYPE html>
<html>
<body>Hello</body>
</html>
```
"'

# SETUP INSTRUCT:
Setup. Use HTML syntax only and produce exactly one valid HTML document as a single code block. Do not output explanations, comments outside the code block, or code in other languages or formats. Ensure the document can be rendered directly by web browsers.

# PLOT INSTRUCT:
Write an HTML code to build a plot of the dataset according to the following instructions. Return exactly one complete HTML document enclosed in a single code block. Include all required libraries and scripts. Do not include explanations or comments. Do not create synthetic data or modify the dataset.

# DATA INSTRUCT:
Data description. Load the dataset by defining: const data = [html.csv]; [html.csv] is a placeholder for the parsed CSV rows. Assume that the placeholder will be replaced at runtime by the CSV content converted into a JavaScript array of objects (i.e., a list of dicts), where each object represents one row with column names as keys and cell values as values. Write the code as if "data" is already such a valid JavaScript array of objects. Do not create synthetic data or modify the dataset. The metadata of the dataset is following:

# E    TAXONOMY OF VISUAL CATEGORIES AND SUBTYPES

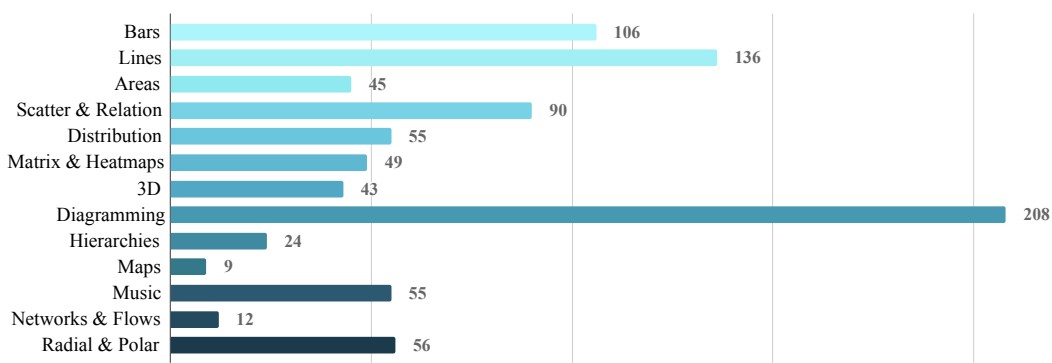

Figure 4: Distribution of fine-grained visualization types in VisPlotBench. Tasks are organized into 13 Visual categories and 116 Subtypes, ensuring broad coverage of both common and underexplored visualization families.
[Back to Appendix Contents]

Table 7: Taxonomy of Visual Categories and Subtypes
[Back to Appendix Contents]

| Visual Category | Subtype | Count | Visual Category | Subtype | Count |
|---|---|---|---|---|---|
| Bars | vertical-bar | 31 | Areas | area | 17 |
| | horizontal-bar | 23 | | stacked-area | 14 |
| | grouped-bar | 15 | | normalized-stacked-area | 4 |
| | normalized-stacked-bar | 8 | | difference-area | 4 |
| | stacked-bar | 7 | | missing-data-matrix | 3 |
| | diverging-bar | 5 | | ternary-area | 1 |
| | dot-plot | 5 | | streamgraph | 1 |
| | lollipop | 3 | | ridgeline | 1 |
| | sorted-bar | 2 | | | |
| | waterfall | 1 | Scatter & Relation | bubble | 25 |
| | polar-bar | 1 | | scatter | 24 |
| | bullet | 1 | | color-scatter | 20 |
| | funnel | 1 | | regression-ci | 4 |
| | combo-chart | 1 | | ternary-line | 4 |
| | missing-bar | 1 | | quadrant-chart | 3 |
| | marimekko | 1 | | ellipse-scatter | 3 |
| | | | | polar-line-scatter | 3 |
| Lines | single-line | 45 | | splom | 2 |
| | multi-line | 39 | | connected-scatter | 1 |
| | function-line | 26 | | dumbbell chart | 1 |
| | step-line | 10 | | | |
| | gapped-line | 6 | | box-plot | 17 |
| | band-line | 4 | | histogram | 13 |
| | slope-chart | 3 | | density-contours | 5 |
| | candlestick | 3 | | violin | 5 |
| | | | | kde-1d | 6 |
| 3D | surface | 21 | Distribution | hexbin-2d | 2 |
| | multi-line | 3 | | qq-plot | 2 |
| | scatter | 4 | | rug-plot | 2 |
| | point-cloud | 3 | | ridgeline | 1 |
| | solid | 3 | | prediction-interval | 1 |
| | single-line | 2 | | spectrum | 1 |
| | vector-field-map | 2 | | | |
| | 3d-density-contours | 2 | | heatmap | 40 |
| | connected-scatter | 1 | | calendar-heatmap | 5 |
| | isosurface | 1 | Matrix & Heatmaps | missing-corr-heatmap | 2 |
| | slices | 1 | | adjacency-matrix | 1 |
| | | | | correlation-heatmap | 1 |
| Diagramming | sequence-diagram | 37 | | | |
| | flowchart | 25 | | treemap | 10 |
| | geometric-figure | 20 | | sunburst | 4 |
| | electrical-circuit-diagram | 16 | Hierarchies | circle-packing | 3 |
| | state-machine | 16 | | missing-dendrogram | 3 |
| | table | 15 | | tidy-tree | 3 |
| | uml-class-diagram | 12 | | indented-tree | 1 |
| | gantt | 11 | | | |
| | timeline | 11 | | choropleth | 4 |
| | simple-figure | 10 | Maps | vector-field-map | 2 |
| | concept-illustration | 10 | | dot-map | 2 |
| | icon | 10 | | proportional-symbol-map | 1 |
| | block-diagram | 3 | | | |
| | physics-diagram | 2 | | sankey | 5 |
| | venn | 2 | | chord | 2 |
| | word-cloud | 2 | Networks & Flows | dependency-graph | 2 |
| | mind-map | 2 | | arc-diagram | 1 |
| | color-palette | 1 | | dag-layered | 1 |
| | arrow-annotations | 1 | | force-directed | 1 |
| | Chemical graph | 1 | | | |
| | sankey | 1 | | pie | 17 |
| | | | | radar | 10 |
| Music | sheet-music | 55 | | polar-line-scatter | 10 |
| | | | Radial & Polar | donut | 7 |
| | | | | radial-bar | 7 |
| | | | | radial-area | 3 |
| | | | | wind-rose | 2 |

# F    BREAKDOWN MAIN RESULTS

In this section, we provide a breakdown of model performance in VisPlotBench. For each visualization language,e report (1) execution pass rate (**Exec Pass**), (2) mean visual and task scores (**Mean**), and (3) the proportion of samples scoring at least 75 (**Good**).

## F.1    PYTHON, VEGA-LITE & LILYPOND

Table 8: Performance of selected languages on the VisPlotbench benchmark. For each model, we report (1) execution pass rate (**Exec Pass**), (2) mean visual and task scores (**Mean**), and (3) the proportion of samples scoring at least 75 (**Good**). The best-performing model in each scale is shown in **bold**, and the second best is underlined.
[Back to Appendix Contents]

| Model | Python (196) | | | | | Vega-Lite (129) | | | | | LilyPond (55) | | | | |
| | Exec Pass | Mean vis | Mean task | Good(≥75) vis | Good(≥75) task | Exec Pass | Mean vis | Mean task | Good(≥75) vis | Good(≥75) task | Exec Pass | Mean vis | Mean task | Good(≥75) vis | Good(≥75) task |
|---|---|---|---|---|---|---|---|---|---|---|---|---|---|---|---|
| GPT-4.1 | 64.3 | 53 | 61 | 51% | 61% | 84.5 | 60 | 68 | 56% | 66% | 43.6 | 14 | 38 | 5% | 36% |
| GPT-4.1 + Self Debug | **84.2** | 66 | 76 | 64% | 76% | 96.1 | 64 | 74 | 60% | 72% | **63.6** | 17 | 54 | 5% | 53% |
| GPT-4.1-mini | 64.8 | 53 | 61 | 47% | 59% | 84.5 | 53 | 63 | 45% | 60% | 16.4 | 2 | 12 | 0% | 11% |
| GPT-4.1-mini + Self Debug | 80.6 | 61 | 71 | 56% | 67% | **96.9** | 60 | 71 | 51% | 68% | 56.4 | 14 | 42 | 0% | 35% |
| ~ 3B Scale | | | | | | | | | | | | | | | |
| DeepSeek-Coder-1.3B-Instruct | 29.1 | 16 | 19 | 10% | 11% | 53.5 | 1 | 2 | 0% | 0% | 30.9 | 2 | 1 | 0% | 0% |
| Qwen2.5-Coder-3B-Instruct | 34.2 | 23 | 28 | 17% | 24% | 68.2 | 25 | 34 | 13% | 22% | 3.6 | 1 | 2 | 0% | 0% |
| VisCoder-3B | 45.4 | 32 | 39 | 26% | 35% | 83.7 | 31 | 37 | 20% | 26% | 21.8 | 3 | 7 | 0% | 2% |
| **VisCoder2-3B** | 56.1 | 39 | 45 | 33% | 38% | 83.0 | 41 | 49 | 33% | 40% | 50.9 | 10 | 31 | 2% | 15% |
| **VisCoder2-3B + Self Debug** | **63.3** | 42 | 49 | 35% | **40%** | 84.5 | 43 | 50 | 34% | **41%** | 52.7 | 10 | 32 | 2% | 15% |
| ~ 7B Scale | | | | | | | | | | | | | | | |
| DeepSeek-Coder-6.7B-Instruct | 39.3 | 25 | 29 | 19% | 23% | 79.8 | 37 | 47 | 24% | 37% | 7.3 | 0 | 3 | 0% | 4% |
| Qwen2.5-Coder-7B-Instruct | 41.3 | 29 | 37 | 24% | 32% | 76.0 | 40 | 50 | 29% | 40% | 5.5 | 0 | 3 | 0% | 4% |
| VisCoder-7B | 58.2 | 40 | 48 | 33% | 42% | 71.3 | 39 | 49 | 31% | 43% | 23.6 | 4 | 11 | 2% | 4% |
| **VisCoder2-7B** | 64.8 | 44 | 54 | 37% | 49% | 83.0 | 49 | 58 | 43% | 51% | 69.1 | 16 | 52 | 2% | 45% |
| **VisCoder2-7B + Self Debug** | **77.0** | 50 | 61 | 41% | 54% | **84.5** | 49 | 59 | 43% | 52% | **72.7** | 17 | 55 | 2% | 45% |
| ~ 14B Scale | | | | | | | | | | | | | | | |
| DeepSeek-Coder-V2-Lite-Instruct | 47.5 | 32 | 40 | 28% | 36% | 75.2 | 36 | 43 | 27% | 33% | 49.1 | 9 | 28 | 0% | 13% |
| Qwen2.5-Coder-14B-Instruct | 50.0 | 35 | 43 | 28% | 39% | 83.0 | 52 | 61 | 42% | 53% | 25.5 | 5 | 12 | 2% | 4% |
| **VisCoder2-14B** | 65.3 | 47 | 56 | 39% | 52% | 93.0 | 55 | 63 | 47% | 58% | 54.6 | 11 | 44 | 0% | 40% |
| **VisCoder2-14B + Self Debug** | **78.1** | 55 | 64 | 46% | 58% | **94.6** | 56 | 64 | 47% | 60% | **63.6** | 12 | 47 | 0% | 40% |
| ~ 32B Scale | | | | | | | | | | | | | | | |
| DeepSeek-Coder-33B-Instruct | 58.2 | 40 | 48 | 34% | 41% | 90.7 | 52 | 61 | 40% | 51% | 30.9 | 3 | 11 | 0% | 4% |
| Qwen2.5-Coder-32B-Instruct | 50.5 | 36 | 43 | 30% | 41% | 83.0 | 48 | 57 | 39% | 49% | 30.9 | 5 | 22 | 2% | 18% |
| **VisCoder2-32B** | 65.3 | 49 | 56 | 42% | 54% | 94.6 | 60 | 70 | 53% | 65% | 56.4 | 14 | 39 | 2% | 27% |
| **VisCoder2-32B + Self Debug** | **81.6** | 58 | 68 | 46% | 62% | **96.1** | 62 | 72 | 54% | 67% | **69.1** | 16 | 48 | 2% | 35% |

## F.2 MERMAID, SVG & LATEX

Table 9: Performance of selected languages on the VisPlotbench benchmark. For each model, we report (1) execution pass rate (**Exec Pass**), (2) mean visual and task scores (**Mean**), and (3) the proportion of samples scoring at least 75 (**Good**). The best-performing model in each scale is shown in **bold**, and the second best is underlined.
[Back to Appendix Contents]

| Model | Mermaid (131) Exec Pass | Mean vis | task | Good(≥75) vis | task | SVG (65) Exec Pass | Mean vis | task | Good(≥75) vis | task | LaTeX (112) Exec Pass | Mean vis | task | Good(≥75) vis | task |
|---|---|---|---|---|---|---|---|---|---|---|---|---|---|---|---|
| GPT-4.1 | 68.7 | 41 | 57 | 22% | 56% | 95.4 | 45 | 92 | 14% | 94% | 31.3 | 18 | 26 | 13% | 25% |
| GPT-4.1 + Self Debug | 93.9 | 56 | 77 | 32% | 73% | 96.9 | 45 | 93 | 14% | 95% | 66.1 | 38 | 56 | 25% | 51% |
| GPT-4.1-mini | 51.9 | 33 | 45 | 18% | 43% | 95.4 | 41 | 86 | 11% | 86% | 29.5 | 21 | 25 | 18% | 25% |
| GPT-4.1-mini + Self Debug | 94.7 | 58 | 79 | 26% | 74% | 96.9 | 42 | 88 | 11% | 88% | 58.9 | 35 | 50 | 23% | 49% |
| ~ 3B Scale | | | | | | | | | | | | | | | |
| DeepSeek-Coder-1.3B-Instruct | 63.4 | 19 | 25 | 2% | 8% | 7.7 | 1 | 1 | 0% | 0% | 4.5 | 2 | 1 | 2% | 1% |
| Qwen2.5-Coder-3B-Instruct | 74.1 | 30 | 38 | 9% | 21% | 75.4 | 18 | 39 | 2% | 28% | 17.9 | 6 | 9 | 3% | 5% |
| VisCoder-3B | 75.6 | 32 | 40 | 12% | 21% | 76.9 | 13 | 31 | 0% | 12% | 23.2 | 9 | 12 | 7% | 9% |
| **VisCoder2-3B** | 76.3 | 43 | 59 | 23% | 50% | 87.7 | 25 | 59 | 3% | 48% | 36.6 | 14 | 21 | 3% | 12% |
| **VisCoder2-3B + Self Debug** | 76.3 | 43 | 59 | 23% | 50% | 87.7 | 25 | 59 | 3% | 48% | 38.4 | 14 | 23 | 3% | 13% |
| ~ 7B Scale | | | | | | | | | | | | | | | |
| DeepSeek-Coder-6.7B-Instruct | 91.6 | 40 | 50 | 11% | 28% | 96.9 | 19 | 46 | 0% | 22% | 18.8 | 6 | 11 | 3% | 8% |
| Qwen2.5-Coder-7B-Instruct | 77.9 | 39 | 53 | 13% | 38% | 92.3 | 23 | 58 | 0% | 40% | 25.9 | 11 | 15 | 6% | 8% |
| VisCoder-7B | 77.1 | 41 | 54 | 17% | 43% | 93.9 | 23 | 53 | 2% | 32% | 25.9 | 10 | 15 | 6% | 12% |
| **VisCoder2-7B** | 78.6 | 43 | 59 | 20% | 53% | 96.9 | 34 | 73 | 3% | 62% | 39.3 | 15 | 23 | 6% | 15% |
| **VisCoder2-7B + Self Debug** | 84.7 | 45 | 62 | 21% | 54% | 96.9 | 34 | 73 | 3% | 62% | 42.9 | 16 | 24 | 6% | 15% |
| ~ 14B Scale | | | | | | | | | | | | | | | |
| DeepSeek-Coder-V2-Lite-Instruct | 69.5 | 34 | 46 | 12% | 34% | 93.9 | 23 | 55 | 2% | 34% | 29.5 | 10 | 16 | 4% | 10% |
| Qwen2.5-Coder-14B-Instruct | 74.8 | 39 | 56 | 15% | 48% | 98.5 | 33 | 80 | 5% | 77% | 30.4 | 15 | 22 | 6% | 15% |
| **VisCoder2-14B** | 81.7 | 53 | 67 | 32% | 62% | 89.2 | 34 | 72 | 8% | 65% | 42.0 | 22 | 33 | 12% | 27% |
| **VisCoder2-14B + Self Debug** | 86.3 | 55 | 70 | 33% | 64% | 90.8 | 34 | 72 | 8% | 65% | 45.5 | 24 | 35 | 12% | 28% |
| ~ 32B Scale | | | | | | | | | | | | | | | |
| DeepSeek-Coder-33B-Instruct | 87.0 | 44 | 57 | 15% | 40% | 92.3 | 23 | 58 | 0% | 43% | 24.1 | 8 | 14 | 4% | 11% |
| Qwen2.5-Coder-32B-Instruct | 71.0 | 41 | 56 | 21% | 53% | 93.9 | 34 | 81 | 3% | 75% | 29.5 | 14 | 25 | 9% | 27% |
| **VisCoder2-32B** | 87.0 | 51 | 67 | 31% | 62% | 81.5 | 33 | 68 | 11% | 63% | 42.9 | 20 | 35 | 11% | 34% |
| **VisCoder2-32B + Self Debug** | 90.1 | 54 | 69 | 34% | 63% | 86.2 | 34 | 71 | 11% | 66% | 61.6 | 28 | 45 | 14% | 42% |

## F.3 ASYMPTOTE & HTML

Table 10: Performance of selected languages on the VisPlotbench benchmark. For each model, we report (1) execution pass rate (**Exec Pass**), (2) mean visual and task scores (**Mean**), and (3) the proportion of samples scoring at least 75 (**Good**). The best-performing model in each scale is shown in **bold**, and the second best is underlined.

| Model | Asymptote (92) | | | | | HTML (108) | | | | |
| | Exec Pass | Mean | | Good(≥75) | | Exec Pass | Mean | | Good(≥75) | |
| | | vis | task | vis | task | | vis | task | vis | task |
|---|---|---|---|---|---|---|---|---|---|---|
| GPT-4.1 | 21.7 | 12 | 20 | 7% | 20% | 89.8 | 48 | 64 | 21% | 50% |
| GPT-4.1 + Self Debug | 46.7 | 22 | 41 | 9% | 39% | 97.2 | 51 | 68 | 22% | 52% |
| GPT-4.1-mini | 23.9 | 13 | 22 | 7% | 21% | 86.1 | 36 | 53 | 11% | 34% |
| GPT-4.1-mini + Self Debug | **48.9** | 21 | 40 | 9% | 36% | **100** | 42 | 62 | 12% | 42% |
| ∼ 3B Scale | | | | | | | | | | |
| DeepSeek-Coder-1.3B-Instruct | 13.0 | 0 | 0 | 0% | 0% | 36.1 | 2 | 3 | 1% | 0% |
| Qwen2.5-Coder-3B-Instruct | 18.5 | 8 | 11 | 4% | 9% | 62.0 | 16 | 19 | 6% | 7% |
| VisCoder-3B | 30.4 | 7 | 12 | 3% | 8% | 79.6 | 21 | 29 | 9% | 17% |
| **VisCoder2-3B** | 62.0 | 23 | 36 | 7% | 26% | 93.5 | 34 | 47 | 8% | 23% |
| **VisCoder2-3B + Self Debug** | **63.0** | 23 | 37 | 7% | 27% | **94.4** | 34 | 47 | 8% | 23% |
| ∼ 7B Scale | | | | | | | | | | |
| DeepSeek-Coder-6.7B-Instruct | 0 | 0 | 0 | 0% | 0% | 22.2 | 5 | 8 | 1% | 3% |
| Qwen2.5-Coder-7B-Instruct | 13.0 | 7 | 10 | 5% | 9% | 64.8 | 20 | 31 | 6% | 13% |
| VisCoder-7B | 17.4 | 7 | 11 | 3% | 9% | 75.9 | 20 | 32 | 5% | 16% |
| **VisCoder2-7B** | 64.1 | 27 | 43 | 11% | 33% | 82.4 | 30 | 46 | 7% | 19% |
| **VisCoder2-7B + Self Debug** | **70.7** | 29 | 47 | 11% | 35% | **84.3** | 31 | 47 | 7% | 21% |
| ∼ 14B Scale | | | | | | | | | | |
| DeepSeek-Coder-V2-Lite-Instruct | 20.7 | 5 | 10 | 1% | 9% | 64.8 | 21 | 32 | 4% | 18% |
| Qwen2.5-Coder-14B-Instruct | 25.0 | 12 | 17 | 9% | 16% | 83.3 | 34 | 50 | 9% | 31% |
| **VisCoder2-14B** | 56.5 | 27 | 45 | 15% | 41% | 90.7 | 41 | 58 | 12% | 36% |
| **VisCoder2-14B + Self Debug** | **66.3** | 31 | 50 | 16% | 45% | **94.4** | 42 | 60 | 13% | 37% |
| ∼ 32B Scale | | | | | | | | | | |
| DeepSeek-Coder-33B-Instruct | 21.7 | 8 | 14 | 2% | 9% | 12.0 | 4 | 6 | 0% | 0% |
| Qwen2.5-Coder-32B-Instruct | 17.4 | 9 | 13 | 5% | 12% | 78.7 | 33 | 49 | 11% | 32% |
| **VisCoder2-32B** | 58.7 | 27 | 46 | 10% | 39% | 91.7 | 43 | 61 | 18% | 48% |
| **VisCoder2-32B + Self Debug** | **71.7** | 31 | 53 | 10% | 41% | **93.5** | 44 | 62 | 18% | 49% |

## G BREAKDOWN SELF-DEBUG RESULTS

In this section, we provide a breakdown of model performance under the self-debug setting. For each language, we report execution pass rates across up to three rounds of automatic correction, grouped by model series.

### G.1 PYTHON & VEGA-LITE

Table 11: Execution pass rates (%) in Python and Vega-Lite under the normal and self-debug settings. Models that fail initially are allowed up to three rounds of automatic correction. Left columns show Python results, right columns show Vega-Lite results.
[Back to Appendix Contents]

| Model | Normal | Python Self Debug | | | Normal | Vega-Lite Self Debug | | |
|---|---|---|---|---|---|---|---|---|
| | | Round 1 | Round 2 | Round 3 | | Round 1 | Round 2 | Round 3 |
| GPT-4.1 | 64.3 | 75.0 | 81.6 | 84.2 | 84.5 | 95.3 | 96.1 | 96.1 |
| GPT-4.1-mini | 64.8 | 73.5 | 79.1 | 80.6 | 84.5 | 95.3 | 96.9 | 96.9 |
| ∼ 3B Scale | | | | | | | | |
| DeepSeek-Coder-1.3B-Instruct | 29.1 | 35.7 | 35.7 | 35.7 | 53.5 | 53.5 | 53.5 | 53.5 |
| Qwen2.5-Coder-3B-Instruct | 34.2 | 39.8 | 41.8 | 42.9 | 68.2 | 68.2 | 69.0 | 69.0 |
| VisCoder-3B | 45.4 | 51.0 | 52.6 | 52.6 | 83.7 | 83.7 | 83.7 | 83.7 |
| **VisCoder2-3B** | 56.1 | 61.7 | 62.8 | 63.3 | 83.0 | 84.5 | 84.5 | 84.5 |
| ∼ 7B Scale | | | | | | | | |
| DeepSeek-Coder-6.7B-Instruct | 39.3 | 46.9 | 49.5 | 53.1 | 79.8 | 81.4 | 81.4 | 81.4 |
| Qwen2.5-Coder-7B-Instruct | 41.3 | 53.6 | 60.2 | 61.7 | 76.0 | 77.5 | 77.5 | 77.5 |
| VisCoder-7B | 58.2 | 66.8 | 68.9 | 71.9 | 71.3 | 76.0 | 77.5 | 77.5 |
| **VisCoder2-7B** | 64.8 | 72.5 | 76.0 | 77.0 | 83.0 | 84.5 | 84.5 | 84.5 |
| ∼ 14B Scale | | | | | | | | |
| DeepSeek-Coder-V2-Lite-Instruct | 47.5 | 54.6 | 55.6 | 58.7 | 75.2 | 78.3 | 79.8 | 79.8 |
| Qwen2.5-Coder-14B-Instruct | 50.0 | 65.3 | 72.5 | 76.0 | 83.0 | 86.8 | 86.8 | 86.8 |
| **VisCoder2-14B** | 65.3 | 76.5 | 78.1 | 78.1 | 93.0 | 93.8 | 94.6 | 94.6 |
| ∼ 32B Scale | | | | | | | | |
| DeepSeek-Coder-33B-Instruct | 58.2 | 67.9 | 71.4 | 73.0 | 90.7 | 92.3 | 92.3 | 92.3 |
| Qwen2.5-Coder-32B-Instruct | 50.5 | 70.9 | 78.1 | 79.1 | 83.0 | 87.6 | 89.9 | 89.9 |
| **VisCoder2-32B** | 65.3 | 76.0 | 80.1 | 81.6 | 94.6 | 96.1 | 96.1 | 96.1 |

## G.2 LILYPOND & MERMAID

Table 12: Execution pass rates (%) in LilyPond and Mermaid under the normal and self-debug settings. Models that fail initially are allowed up to three rounds of automatic correction. Left columns show LilyPond results, right columns show Mermaid results.
[Back to Appendix Contents]

| Model | Normal | LilyPond Self Debug | | | Normal | Mermaid Self Debug | | |
|---|---|---|---|---|---|---|---|---|
| | | Round 1 | Round 2 | Round 3 | | Round 1 | Round 2 | Round 3 |
| GPT-4.1 | 43.6 | 54.5 | 63.6 | 63.6 | 68.7 | 84.7 | 93.0 | 93.9 |
| GPT-4.1-mini | 16.4 | 30.9 | 47.3 | 56.4 | 51.9 | 81.7 | 90.1 | 94.7 |
| ~ 3B Scale | | | | | | | | |
| DeepSeek-Coder-1.3B-Instruct | 30.9 | 32.7 | 32.7 | 32.7 | 63.4 | 76.3 | 77.9 | 78.6 |
| Qwen2.5-Coder-3B-Instruct | 3.6 | 5.5 | 5.5 | 5.5 | 74.1 | 76.3 | 76.3 | 76.3 |
| VisCoder-3B | 21.8 | 21.8 | 21.8 | 21.8 | 75.6 | 76.3 | 76.3 | 76.3 |
| **VisCoder2-3B** | 50.9 | 52.7 | 52.7 | 52.7 | 76.3 | 76.3 | 76.3 | 76.3 |
| ~ 7B Scale | | | | | | | | |
| DeepSeek-Coder-6.7B-Instruct | 7.3 | 9.1 | 10.9 | 10.9 | 91.6 | 93.9 | 94.7 | 94.7 |
| Qwen2.5-Coder-7B-Instruct | 5.5 | 5.5 | 5.5 | 5.5 | 77.9 | 79.4 | 79.4 | 79.4 |
| VisCoder-7B | 23.6 | 27.3 | 30.9 | 30.9 | 77.1 | 80.9 | 80.9 | 80.9 |
| **VisCoder2-7B** | 69.1 | 72.7 | 72.7 | 72.7 | 78.6 | 84.0 | 84.7 | 84.7 |
| ~ 14B Scale | | | | | | | | |
| DeepSeek-Coder-V2-Lite-Instruct | 49.1 | 52.7 | 52.7 | 52.7 | 69.5 | 69.5 | 69.5 | 71.0 |
| Qwen2.5-Coder-14B-Instruct | 50.0 | 65.3 | 72.5 | 76.0 | 83.0 | 86.8 | 86.8 | 86.8 |
| **VisCoder2-14B** | 54.6 | 63.6 | 63.6 | 63.6 | 81.7 | 86.3 | 86.3 | 86.3 |
| ~ 32B Scale | | | | | | | | |
| DeepSeek-Coder-33B-Instruct | 30.9 | 40.0 | 41.8 | 41.8 | 87.0 | 87.0 | 87.8 | 88.6 |
| Qwen2.5-Coder-32B-Instruct | 30.9 | 40.0 | 43.6 | 43.6 | 71.0 | 74.8 | 75.6 | 76.3 |
| **VisCoder2-32B** | 56.4 | 61.8 | 69.1 | 69.1 | 87.0 | 89.3 | 90.1 | 90.1 |

## G.3 SVG & LaTeX

Table 13: Execution pass rates (%) in SVG and LaTeX under the normal and self-debug settings. Models that fail initially are allowed up to three rounds of automatic correction. Left columns show SVG results, right columns show LaTeX results.
[Back to Appendix Contents]

| Model | Normal | SVG Self Debug | | | Normal | LaTeX Self Debug | | |
|---|---|---|---|---|---|---|---|---|
| | | Round 1 | Round 2 | Round 3 | | Round 1 | Round 2 | Round 3 |
| GPT-4.1 | 95.4 | 96.9 | 96.9 | 96.9 | 31.3 | 53.6 | 59.8 | 66.1 |
| GPT-4.1-mini | 95.4 | 96.9 | 96.9 | 96.9 | 29.5 | 50.9 | 55.4 | 58.9 |
| ~ 3B Scale | | | | | | | | |
| DeepSeek-Coder-1.3B-Instruct | 7.7 | 95.4 | 95.4 | 95.4 | 4.5 | 5.4 | 5.4 | 5.4 |
| Qwen2.5-Coder-3B-Instruct | 75.4 | 75.4 | 75.4 | 75.4 | 17.9 | 17.9 | 17.9 | 17.9 |
| VisCoder-3B | 76.9 | 76.9 | 76.9 | 76.9 | 23.2 | 25.9 | 25.9 | 25.9 |
| **VisCoder2-3B** | 87.7 | 87.7 | 87.7 | 87.7 | 36.6 | 38.4 | 38.4 | 38.4 |
| ~ 7B Scale | | | | | | | | |
| DeepSeek-Coder-6.7B-Instruct | 96.9 | 98.5 | 98.5 | 98.5 | 18.8 | 19.6 | 22.3 | 22.3 |
| Qwen2.5-Coder-7B-Instruct | 92.3 | 92.3 | 92.3 | 92.3 | 25.9 | 28.6 | 30.4 | 30.4 |
| VisCoder-7B | 93.9 | 93.9 | 93.9 | 93.9 | 25.9 | 38.4 | 42.0 | 43.8 |
| **VisCoder2-7B** | 96.9 | 96.9 | 96.9 | 96.9 | 39.3 | 42.9 | 42.9 | 42.9 |
| ~ 14B Scale | | | | | | | | |
| DeepSeek-Coder-V2-Lite-Instruct | 93.9 | 93.9 | 93.9 | 93.9 | 29.5 | 33.9 | 35.7 | 35.7 |
| Qwen2.5-Coder-14B-Instruct | 98.5 | 98.5 | 98.5 | 98.5 | 30.4 | 37.5 | 38.4 | 38.4 |
| **VisCoder2-14B** | 89.2 | 90.8 | 90.8 | 90.8 | 42.0 | 43.8 | 45.5 | 45.5 |
| ~ 32B Scale | | | | | | | | |
| DeepSeek-Coder-33B-Instruct | 92.3 | 92.3 | 92.3 | 92.3 | 24.1 | 28.6 | 31.3 | 31.3 |
| Qwen2.5-Coder-32B-Instruct | 93.9 | 93.9 | 93.9 | 93.9 | 29.5 | 42.9 | 50.0 | 51.8 |
| **VisCoder2-32B** | 81.5 | 84.6 | 86.2 | 86.2 | 42.9 | 55.4 | 59.8 | 61.6 |

## G.4 ASYMPTOTE & HTML

Table 14: Execution pass rates (%) in Asymptote and HTML under the normal and self-debug settings. Models that fail initially are allowed up to three rounds of automatic correction. Left columns show Asymptote results, right columns show HTML results.
[Back to Appendix Contents]

| Model | Normal | Asymptote Self Debug | | | Normal | HTML Self Debug | | |
|---|---|---|---|---|---|---|---|---|
| | | Round 1 | Round 2 | Round 3 | | Round 1 | Round 2 | Round 3 |
| GPT-4.1 | 21.7 | 35.9 | 43.5 | 46.7 | 89.8 | 96.3 | 97.2 | 97.2 |
| GPT-4.1-mini | 23.9 | 37.0 | 42.4 | 48.9 | 86.1 | 99.1 | 99.1 | 100 |
| ∼ 3B Scale | | | | | | | | |
| DeepSeek-Coder-1.3B-Instruct | 13.0 | 17.4 | 17.4 | 17.4 | 36.1 | 36.1 | 36.1 | 36.1 |
| Qwen2.5-Coder-3B-Instruct | 18.5 | 18.5 | 18.5 | 18.5 | 62.0 | 65.7 | 70.4 | 70.4 |
| VisCoder-3B | 30.4 | 31.5 | 32.6 | 32.6 | 79.6 | 83.3 | 83.3 | 83.3 |
| **VisCoder2-3B** | 62.0 | 63.0 | 63.0 | 63.0 | 93.5 | 94.4 | 94.4 | 94.4 |
| ∼ 7B Scale | | | | | | | | |
| DeepSeek-Coder-6.7B-Instruct | 0.0 | 1.1 | 2.2 | 2.2 | 22.2 | 25.0 | 25.0 | 25.0 |
| Qwen2.5-Coder-7B-Instruct | 13.0 | 16.3 | 20.7 | 20.7 | 64.8 | 75.9 | 76.9 | 76.9 |
| VisCoder-7B | 17.4 | 26.1 | 26.1 | 26.1 | 75.9 | 81.5 | 82.4 | 82.4 |
| **VisCoder2-7B** | 64.1 | 68.5 | 70.7 | 70.7 | 82.4 | 84.3 | 84.3 | 84.3 |
| ∼ 14B Scale | | | | | | | | |
| DeepSeek-Coder-V2-Lite-Instruct | 20.7 | 23.9 | 26.1 | 26.1 | 64.8 | 76.9 | 79.6 | 79.6 |
| Qwen2.5-Coder-14B-Instruct | 25.0 | 32.6 | 39.1 | 40.2 | 83.3 | 89.8 | 89.8 | 89.8 |
| **VisCoder2-14B** | 56.5 | 64.1 | 66.3 | 66.3 | 90.7 | 94.4 | 94.4 | 94.4 |
| ∼ 32B Scale | | | | | | | | |
| DeepSeek-Coder-33B-Instruct | 21.7 | 26.1 | 28.3 | 29.4 | 12.0 | 14.8 | 14.8 | 14.8 |
| Qwen2.5-Coder-32B-Instruct | 17.4 | 25.0 | 31.5 | 33.7 | 78.7 | 88.9 | 89.8 | 89.8 |
| **VisCoder2-32B** | 58.7 | 68.5 | 71.7 | 71.7 | 91.7 | 92.6 | 93.5 | 93.5 |

# H  BREAKDOWN ERROR TYPE RESULTS

In this section, we provide a breakdown error type results of execution errors for GPT-4.1 and VisCoder2-32B. For each language, we report error type across up to three rounds of automatic correction.

## H.1  PYTHON

Table 15: Distribution of execution errors for GPT-4.1 and VisCoder2-32B in Python. Each column shows error counts at different self-debugging rounds after initial failure.
[Back to Appendix Contents]

| Error Type | GPT-4.1 | | | | VisCoder2-32B | | | |
|---|---|---|---|---|---|---|---|---|
| | Normal | Round 1 | Round 2 | Round 3 | Normal | Round 1 | Round 2 | Round 3 |
| AttributeError | 17 | 12 | 9 | 8 | 15 | 12 | 12 | 10 |
| FileNotFoundError | - | - | - | - | 1 | 1 | 1 | 1 |
| ImportError | 2 | 2 | 1 | 0 | 2 | 1 | 1 | 1 |
| SchemaValidationError | 1 | 1 | 1 | 1 | - | - | - | - |
| KeyError | - | - | - | - | 3 | 2 | 1 | 0 |
| KeyboardInterrupt | 7 | 7 | 6 | 6 | 9 | 9 | 9 | 9 |
| CellSizeError | - | - | - | - | 1 | 1 | 1 | 1 |
| DataError | - | - | - | - | 2 | 2 | 2 | 2 |
| NameError | - | - | - | - | 2 | 1 | 0 | 0 |
| RuntimeError | 2 | 0 | 0 | 0 | - | - | - | - |
| SyntaxError | 1 | 1 | 1 | 1 | 1 | 1 | 1 | 1 |
| TypeError | 20 | 16 | 14 | 14 | 13 | 7 | 3 | 3 |
| ValueError | 20 | 10 | 4 | 1 | 19 | 10 | 8 | 8 |
| **Total Errors** | 70 | 49 | 36 | 31 | 68 | 47 | 39 | 36 |

## H.2  VEGA-LITE

Table 16: Distribution of execution errors for GPT-4.1 and VisCoder2-32B in Vega-Lite. Each column shows error counts at different self-debugging rounds after initial failure.
[Back to Appendix Contents]

| Error Type | GPT-4.1 | | | | VisCoder2-32B | | | |
|---|---|---|---|---|---|---|---|---|
| | Normal | Round 1 | Round 2 | Round 3 | Normal | Round 1 | Round 2 | Round 3 |
| JSONDecodeError | 1 | 0 | 0 | 0 | - | - | - | - |
| KeyboardInterrupt | 1 | 0 | 0 | 0 | 1 | 1 | 1 | 1 |
| ParseError | 8 | 2 | 2 | 2 | 2 | 1 | 1 | 1 |
| TypeError | 9 | 4 | 2 | 2 | 2 | 1 | 1 | 1 |
| RenderingError | 1 | 0 | 0 | 0 | 2 | 2 | 2 | 2 |
| **Total Errors** | 20 | 6 | 4 | 4 | 7 | 5 | 5 | 5 |

## H.3  LILYPOND

Table 17: Distribution of execution errors for GPT-4.1 and VisCoder2-32B in Lilypond. Each column shows error counts at different self-debugging rounds after initial failure.
[Back to Appendix Contents]

| Error Type | GPT-4.1 | | | | VisCoder2-32B | | | |
|---|---|---|---|---|---|---|---|---|
| | Normal | Round 1 | Round 2 | Round 3 | Normal | Round 1 | Round 2 | Round 3 |
| FileNotFoundError | 1 | 1 | 1 | 1 | 1 | 1 | 1 | 1 |
| MarkupError | 3 | 2 | 2 | 2 | 4 | 4 | 4 | 4 |
| SyntaxError | 25 | 17 | 12 | 12 | 14 | 12 | 10 | 10 |
| TypeError | 2 | 2 | 2 | 2 | 5 | 4 | 2 | 2 |
| **Total Errors** | 31 | 25 | 20 | 20 | 24 | 21 | 17 | 17 |

## H.4 MERMAID

Table 18: Distribution of execution errors for GPT-4.1 and VisCoder2-32B in Mermaid. Each column shows error counts at different self-debugging rounds after initial failure.
[Back to Appendix Contents]

| Error Type | GPT-4.1 | | | | VisCoder2-32B | | | |
|---|---|---|---|---|---|---|---|---|
| | Normal | Round 1 | Round 2 | Round 3 | Normal | Round 1 | Round 2 | Round 3 |
| StructureError | 2 | 2 | 0 | 0 | 1 | 0 | 0 | 0 |
| SyntaxError | 32 | 16 | 7 | 7 | 12 | 10 | 9 | 9 |
| TypeError | 2 | 1 | 1 | 1 | - | - | - | - |
| UnknownDiagramError | 4 | 1 | 0 | 0 | 1 | 1 | 1 | 1 |
| YAMLException | 1 | 0 | 0 | 0 | - | - | - | - |
| LogicError | - | - | - | - | 1 | 1 | 1 | 1 |
| DiagramLimitError | - | - | - | - | 1 | 1 | 1 | 1 |
| KeyboardInterrupt | - | - | - | - | 1 | 1 | 1 | 1 |
| **Total Errors** | 41 | 20 | 8 | 8 | 17 | 14 | 13 | 13 |

## H.5 SVG

Table 19: Distribution of execution errors for GPT-4.1 and VisCoder2-32B in SVG. Each column shows error counts at different self-debugging rounds after initial failure.
[Back to Appendix Contents]

| Error Type | GPT-4.1 | | | | VisCoder2-32B | | | |
|---|---|---|---|---|---|---|---|---|
| | Normal | Round 1 | Round 2 | Round 3 | Normal | Round 1 | Round 2 | Round 3 |
| SyntaxError | 1 | 1 | 1 | 1 | 4 | 2 | 2 | 2 |
| UnclosedError | 2 | 1 | 1 | 1 | 8 | 8 | 7 | 7 |
| **Total Errors** | 3 | 2 | 2 | 2 | 12 | 10 | 9 | 9 |

## H.6 LATEX

Table 20: Distribution of execution errors for GPT-4.1 and VisCoder2-32B in LaTeX. Each column shows error counts at different self-debugging rounds after initial failure.
[Back to Appendix Contents]

| Error Type | GPT-4.1 | | | | VisCoder2-32B | | | |
|---|---|---|---|---|---|---|---|---|
| | Normal | Round 1 | Round 2 | Round 3 | Normal | Round 1 | Round 2 | Round 3 |
| KeyboardInterrupt | 16 | 16 | 16 | 15 | 5 | 5 | 5 | 2 |
| PackageError | 2 | 2 | 1 | 1 | - | - | - | - |
| RuntimeError | 17 | 9 | 7 | 7 | 27 | 12 | 9 | 6 |
| StructureError | 3 | 2 | 2 | 1 | 6 | 3 | 3 | 3 |
| SyntaxError | 5 | 4 | 4 | 4 | 10 | 6 | 4 | 4 |
| UndefinedError | 21 | 17 | 15 | 15 | 28 | 26 | 24 | 23 |
| **Total Errors** | 64 | 50 | 45 | 43 | 77 | 52 | 45 | 38 |

## H.7 ASYMPTOTE

Table 21: Distribution of execution errors for GPT-4.1 and VisCoder2-32B in Asymptote. Each column shows error counts at different self-debugging rounds after initial failure.
[Back to Appendix Contents]

| Error Type | GPT-4.1 | | | | VisCoder2-32B | | | |
|---|---|---|---|---|---|---|---|---|
| | Normal | Round 1 | Round 2 | Round 3 | Normal | Round 1 | Round 2 | Round 3 |
| AmbiguousFunctionCall | - | - | - | - | 1 | 1 | 1 | 1 |
| AmbiguousUsageError | 1 | 1 | 1 | 1 | 1 | 1 | 1 | 1 |
| CastError | 2 | 1 | 1 | 1 | - | - | - | - |
| FunctionSignatureError | 28 | 20 | 18 | 16 | 9 | 4 | 3 | 3 |
| ModuleLoadError | 16 | 15 | 13 | 13 | 2 | 2 | 2 | 2 |
| RuntimeError | 1 | 1 | 1 | 1 | 8 | 7 | 6 | 6 |
| SyntaxError | 3 | 2 | 1 | 1 | - | - | - | - |
| VariableError | 21 | 19 | 17 | 16 | 15 | 12 | 11 | 11 |
| KeyboardInterrupt | - | - | - | - | 2 | 2 | 2 | 2 |
| **Total Errors** | 72 | 59 | 52 | 49 | 38 | 29 | 26 | 26 |

## H.8 HTML

Table 22: Distribution of execution errors for GPT-4.1 and VisCoder2-32B in HTML. Each column shows error counts at different self-debugging rounds after initial failure.
[Back to Appendix Contents]

| Error Type | GPT-4.1 | | | | VisCoder2-32B | | | |
|---|---|---|---|---|---|---|---|---|
| | Normal | Round 1 | Round 2 | Round 3 | Normal | Round 1 | Round 2 | Round 3 |
| ConsoleError | 1 | 1 | 1 | 1 | 3 | 2 | 2 | 2 |
| PageError | 9 | 2 | 1 | 1 | 3 | 3 | 2 | 2 |
| RequestFailed | 1 | 1 | 1 | 1 | 3 | 3 | 3 | 3 |
| **Total Errors** | 11 | 4 | 3 | 3 | 9 | 8 | 7 | 7 |

# I ADDITIONAL EXPERIMENTAL RESULTS & DISCUSSIONS

In this section, we present additional experimental results and discussions including evaluation settings, additional evaluation results and deep analysis of self-debug behavior & Task/Visual Score.

## I.1 EVALUATION SETTINGS

**Self-Debug Evaluation Protocol**  In VisPlotBench, we adopt the same self-debug evaluation mode used in VisCoder to simulate a realistic developer-style debugging workflow. In this setting, if the model's initial code generation fails to execute or does not produce a valid plot, the model is given up to K rounds to iteratively refine its output based on feedback from the previous attempt. In each round, only the tasks that remain unsolved from the previous iteration are reconsidered. The model receives a multi-turn conversational prompt consisting of (i) the original natural-language instruction, (ii) the previously generated code that failed, and (iii) the feedback derived from the execution error. Based on this dialogue history, the model produces a revised version of the code. If the revised code executes successfully and generates a valid plot, the task is marked as solved and excluded from further rounds; otherwise, the latest failed output is recorded and carried forward to the next iteration.

---

**Algorithm 1** Self-Debug Evaluation Protocol

---

1: Let $F_0$ be failed tasks from initial evaluation
2: **for** $i = 1$ to $K$ **do**
3:    **for** each task $x$ in $F_{i-1}$ not yet fixed **do**
4:       Fix $x$ via feedback-driven prompting
5:       Evaluate the result of the revised code
6:       **if** successful **then**
7:          Mark $x$ as fixed & record output
8:       **else**
9:          Record $x$'s latest failed output
10:       **end if**
11:    **end for**
12: **end for**
13: Evaluate all tasks with final recorded outputs

---

In all experiments, we set the maximum number of rounds to $K = 3$. After all rounds are completed, each task is evaluated using its latest recorded output (either the successfully corrected code from an earlier round or the final failed attempt) using the same evaluation pipeline as in the initial pass. This iterative mechanism mirrors the common "generate–execute–repair" workflow and provides a standardized way to evaluate how models recover from different error types across languages.

**Task and Visual Score Metrics**  In VisPlotBench, we follow the scoring procedure introduced in PandasPlotBench (Galimzyanov et al., 2024), and the judge prompts are provided in Appendix D.2. The core idea is to use a GPT model to compare the ground-truth image and the model-rendered image within the context of the task description. For the Task Score, the judge compares the generated plot against the task instruction; for the Visual Score, the judge compares the generated plot against the ground-truth reference image.

## I.2 ADDITIONAL EVALUATION RESULTS

**Results on PandasPlotBench**  We additionally evaluate Qwen2.5-Coder, VisCoder, VisCoder2, and proprietary models (GPT-4.1 / GPT-4.1-mini) on PandasPlotBench, which covers Python-based visualization libraries including `Matplotlib`, `Seaborn`, and `Plotly`. Table 23 reports Execution Pass Rate, Task Score, Visual Score, and the proportion of samples achieving a score≥75.

Across all three libraries, VisCoder2 consistently outperforms the base Qwen2.5-Coder models on execution success as well as both semantic (task) and perceptual (visual) metrics. The gains further increase under self-debug, where VisCoder2-14B achieves performance close to GPT-4.1. These results support the generalization capability of VisCoder2 on unseen visualization benchmarks.

Table 23: Results of VisCoder2 and Baselines on the PandasPlotBench. For each model, we report (1) execution pass rate (**Exec Pass**), (2) mean visual and task scores (**Mean**), and (3) the proportion of samples scoring at least 75 (**Good**).
[Back to Appendix Contents]

| Model | Matplotlib | | | | | Seaborn | | | | | Plotly | | | | |
| | Exec Pass | Mean | | Good($\geq$75) | | Exec Pass | Mean | | Good($\geq$75) | | Exec Pass | Mean | | Good($\geq$75) | |
| | | vis | task | vis | task | | vis | task | vis | task | | vis | task | vis | task |
|---|---|---|---|---|---|---|---|---|---|---|---|---|---|---|---|
| GPT-4.1 | 94.3 | 75 | 88 | 69% | 91% | 93.7 | 72 | 86 | 68% | 86% | 76.6 | 61 | 67 | 58% | 66% |
| GPT-4.1 + Self Debug | **100** | **77** | **90** | 70% | **94%** | 98.9 | **74** | **89** | 70% | **90%** | 97.7 | 74 | 85 | 69% | 85% |
| GPT-4.1-mini | 94.3 | 74 | 86 | 71% | 87% | 92 | 71 | 83 | 64% | 85% | 70.9 | 55 | 62 | 51% | 63% |
| GPT-4.1-mini + Self Debug | 98.9 | 76 | 89 | **73%** | 91% | **100** | **74** | 87 | 67% | **90%** | 97.1 | 72 | 84 | 65% | **86%** |
| $\sim$ 3B Scale | | | | | | | | | | | | | | | |
| Qwen2.5-Coder-3B-Instruct | 71.4 | 56 | **72** | 50% | 69% | 58.3 | 44 | 55 | 36% | 51% | 27.4 | 17 | 19 | 17% | 18% |
| VisCoder-3B | 81.7 | 60 | 69 | 53% | 69% | 73.7 | 48 | 65 | 38% | 61% | 60.6 | 38 | 45 | 32% | 44% |
| VisCoder-3B + Self Debug | 85.1 | 60 | 70 | 53% | 69% | **78.3** | 48 | **66** | 37% | 62% | 64.6 | 40 | 48 | 34% | 47% |
| **VisCoder2-3B** | 83.4 | 62 | 70 | 55% | 69% | 73.7 | 51 | 62 | 42% | 56% | 61.1 | 41 | 48 | 35% | 45% |
| **VisCoder2-3B + Self Debug** | **86.3** | **63** | 71 | **56%** | 69% | 77.7 | **53** | 64 | **43%** | 58% | 64 | **43** | **52** | **37%** | **49%** |
| $\sim$ 7B Scale | | | | | | | | | | | | | | | |
| Qwen2.5-Coder-7B-Instruct | 78.3 | 63 | 76 | 58% | 75% | 68.6 | 51 | 63 | 40% | 62% | 48 | 29 | 34 | 24% | 31% |
| VisCoder-7B | 87.4 | 66 | 78 | 60% | 80% | 76.6 | 57 | 70 | 50% | 68% | 74.3 | 48 | 60 | 41% | 61% |
| VisCoder-7B + Self Debug | 91.4 | 67 | **81** | 62% | **83%** | 90.3 | 62 | **77** | 51% | **75%** | 81.7 | 51 | 65 | 44% | 65% |
| **VisCoder2-7B** | 87.4 | 67 | 76 | 61% | 78% | 83.4 | 61 | 72 | 52% | 70% | 77.7 | 48 | 62 | 43% | 63% |
| **VisCoder2-7B + Self Debug** | **92** | **69** | 78 | **62%** | 80% | **93.7** | **64** | 76 | **53%** | 74% | **87.4** | **53** | **68** | **47%** | **67%** |
| $\sim$ 14B Scale | | | | | | | | | | | | | | | |
| Qwen2.5-Coder-14B-Instruct | 86.3 | 67 | 78 | 61% | 78% | 76.6 | 58 | 70 | 51% | 67% | 56 | 40 | 42 | 37% | 39% |
| VisCoder-14B | 86.3 | - | - | - | - | 78.9 | - | - | - | - | 74.3 | - | - | - | - |
| VisCoder-14B + Self Debug | 93.7 | - | - | - | - | 92.6 | - | - | - | - | 93.1 | - | - | - | - |
| **VisCoder2-14B** | 88 | 70 | 81 | 63% | 80% | 84 | 66 | 74 | 58% | 71% | 78.3 | 52 | 66 | 46% | 65% |
| **VisCoder2-14B + Self Debug** | **94.3** | **71** | **83** | **65%** | **83%** | **93.7** | **67** | **79** | **59%** | **78%** | **94.9** | **60** | **71** | **51%** | **70%** |

**Results on Human-Eval** We further evaluate Qwen2.5-Coder-Instruct and the fine-tuned Vis-Coder2 models on HumanEval and HumanEval+ (Chen et al., 2021).

Table 24: Performance on HumanEval and HumanEval+, Pass@1.
[Back to Appendix Contents]

| Model | HumanEval Pass@1 | HumanEval+ Pass@1 |
|---|---|---|
| GPT-4.1 | 97 | 91.5 |
| GPT-4.1-mini | 92.1 | 86.6 |
| Qwen2.5-Coder-3B-Instruct | 84.8 | 79.9 |
| VisCoder2-3B | 81.1 | 76.2 |
| Qwen2.5-Coder-7B-Instruct | 91.5 | 84.8 |
| VisCoder2-7B | 89 | 83.5 |
| Qwen2.5-Coder-14B-Instruct | 92.1 | 86.6 |
| VisCoder2-14B | 92.1 | 84.8 |
| Qwen2.5-Coder-32B-Instruct | 90.9 | 85.4 |
| VisCoder2-32B | 87.8 | 81.7 |

Table 24 shows that VisCoder2 exhibits only a modest 2–3 point decrease compared to the base Qwen2.5-Coder models on HumanEval/HumanEval+. This behavior aligns with the distributional differences between the two task families: VisCoder2 is trained heavily on multi-language visualization code, whereas HumanEval focuses on algorithmic and data-structure problems. Such minor fluctuations are therefore expected and do not indicate systematic capability degradation. Overall, VisCoder2 maintains stable general coding ability while achieving substantial gains on its target task of cross-language executable visualization code generation and multi-round self-debug.

I.3 DEEP ANALYSIS OF SELF-DEBUG BEHAVIOR AND TASK/VISUAL SCORE

**Self-Debug Analysis** In Appendix G, we report the complete self-debug results of all models across the eight visualization languages. To better understand model behavior during self-debug, we select four representative systems: GPT-4.1, GPT-4.1-mini, Qwen2.5-Coder-32B-Instruct, and our

fine-tuned VisCoder2-32B, and provide a deeper analysis that combines language-specific characteristics with model behaviors. Table 25, Table 26 and Table 27 present the corresponding results.

Table 25: Execution Pass Rate across Self-Debug Rounds (Python, Vega-Lite, LilyPond).
[Back to Appendix Contents]

| Model | Python | | | | Vega-Lite | | | | LilyPond | | | |
|---|---|---|---|---|---|---|---|---|---|---|---|---|
| | Normal | R1 | R2 | R3 | Normal | R1 | R2 | R3 | Normal | R1 | R2 | R3 |
| GPT-4.1 | 64.3 | 75.0 | 81.6 | 84.2 | 84.5 | 95.4 | 96.1 | 96.1 | 45.5 | 58.2 | 61.8 | 65.5 |
| GPT-4.1-mini | 64.8 | 73.5 | 79.1 | 80.6 | 84.5 | 95.4 | 96.9 | 96.9 | 22.2 | 37.0 | 50.0 | 57.4 |
| Qwen2.5-Coder-32B-Instruct | 50.5 | 70.9 | 78.1 | 79.1 | 83.0 | 87.6 | 89.9 | 89.9 | 30.9 | 40.0 | 43.6 | 43.6 |
| VisCoder2-32B | 65.3 | 76.0 | 80.1 | 81.6 | 94.6 | 96.1 | 96.1 | 96.1 | 56.4 | 61.8 | 69.1 | 69.1 |

Table 26: Execution Pass Rate across Self-Debug Rounds (Mermaid, SVG, LaTeX).
[Back to Appendix Contents]

| Model | Mermaid | | | | SVG | | | | LaTeX | | | |
|---|---|---|---|---|---|---|---|---|---|---|---|---|
| | Normal | R1 | R2 | R3 | Normal | R1 | R2 | R3 | Normal | R1 | R2 | R3 |
| GPT-4.1 | 68.7 | 84.7 | 93.9 | 93.9 | 92.3 | 93.9 | 95.4 | 95.4 | 31.3 | 53.6 | 59.8 | 66.1 |
| GPT-4.1-mini | 51.9 | 81.7 | 90.1 | 94.7 | 89.1 | 95.3 | 95.3 | 96.9 | 29.5 | 50.9 | 55.4 | 58.9 |
| Qwen2.5-Coder-32B-Instruct | 71.0 | 74.8 | 75.6 | 76.3 | 93.9 | 93.9 | 93.9 | 93.9 | 29.5 | 42.9 | 50.0 | 51.8 |
| VisCoder2-32B | 87.0 | 89.3 | 90.1 | 90.1 | 81.5 | 84.6 | 86.2 | 86.2 | 42.9 | 55.4 | 59.8 | 61.6 |

Table 27: Execution Pass Rate across Self-Debug Rounds (Asymptote, HTML).
[Back to Appendix Contents]

| Model | Asymptote | | | | HTML | | | |
|---|---|---|---|---|---|---|---|---|
| | Normal | R1 | R2 | R3 | Normal | R1 | R2 | R3 |
| GPT-4.1 | 21.7 | 35.9 | 43.5 | 46.7 | 89.8 | 96.3 | 97.2 | 97.2 |
| GPT-4.1-mini | 23.9 | 37.0 | 42.4 | 48.9 | 86.1 | 99.1 | 99.1 | 100.0 |
| Qwen2.5-Coder-32B-Instruct | 17.4 | 25.0 | 31.5 | 33.7 | 78.7 | 88.9 | 89.8 | 89.8 |
| VisCoder2-32B | 58.7 | 68.5 | 71.7 | 71.7 | 91.7 | 92.6 | 93.5 | 93.5 |

i) **Effect of self-debug:** Self-debugging consistently improves execution reliability across all models and most languages. GPT-4.1 and GPT-4.1-mini already exhibit the strongest cross-language performance at the initial generation stage, and self-debug further repairs the majority of syntax- and interface-related errors, allowing them to reach near-saturated execution rates in languages such as `Python`, `Vega-Lite`, and `Mermaid`. In contrast, Qwen2.5-Coder-32B-Instruct starts from a weaker baseline, but still benefits substantially from iterative correction. Our fine-tuned VisCoder2-32B achieves significantly higher initial and final execution success rates than the baseline in seven languages, with especially large gains in symbol-intensive languages such as `LilyPond`, `Asymptote`, and `LaTeX`, where self-debug brings it close to or even above the GPT models. Together, these results show that self-debug offers a robust, model-agnostic mechanism for correcting structural and shallow syntactic errors, and is a key driver of multi-language reliability.

ii) **Cross-language trends:** In declarative languages like `Vega-Lite` and `HTML`, clear structural rules and diagnostics allow most models to reach near-saturated execution within one or two rounds. In execution-driven languages such as Python and Mermaid, rich runtime diagnostics provide actionable signals that drive steady improvements across rounds. For symbolic or compiler-dependent languages (`LilyPond`, `Asymptote`, `LaTeX`), models fix shallow syntax early, but deeper semantic issues rarely surface in logs, so improvements taper off after the first round. For `SVG`, the rendering pipeline is sensitive to XML well-formedness but offers little semantic or layout feedback. Larger models generate more complex structures, making issues like unclosed tags or malformed attributes more common, and these are difficult to repair under weak feedback, limiting improvement.

iii) **Characteristics of self-debug:** Across all models and languages, the first round of self-debug consistently yields the largest improvement, correcting the majority of failures caused by shal-

low issues such as missing syntax, mismatched parameters, or incorrect references. Starting from the second round, the rate of improvement drops sharply, and by the third round performance typically plateaus. This pattern indicates that the current feedback mechanism effectively exposes structural and interface-related errors, but provides limited signals for deeper semantic inconsistencies, complex symbolic dependencies, or issues tied to specific rendering or parsing processes. As a result, self-debug follows a stable "large first-round gains followed by diminishing returns" trajectory across the entire evaluation.

iv) **Failures remain across models:** Building upon the error categorization in Section 5.3, we further examined the final unsuccessful cases of GPT-4.1 and VisCoder2-32B across eight visualization languages and found that the remaining failures fall into three representative patterns that are difficult for current self-debug mechanisms to resolve: (1) deep semantic inconsistencies, such as mismatched variable meanings, incorrect data relationships, or incoherent multi-step plotting logic, which rarely surface explicitly in execution logs and therefore prevent the model from identifying the true source of failure; (2) grammar- or compiler-dependent symbolic errors, including macro expansion, symbol binding, or scope-resolution failures in languages like `LilyPond`, Asymptote, and `LaTeX`, where parser messages tend to be generic and provide little actionable guidance; and (3) runtime-related behavioral errors, such as invoking rendering packages that were never loaded, calling APIs unsupported by the target language, or generating plotting logic that may trigger infinite loops or excessive resource consumption. Although these issues manifest as runtime failures, the execution feedback typically contains only coarse, non-localized error signals, making it difficult for the model to refine its output across self-debug rounds. Overall, the lack of explicit localization for semantic, symbolic, and runtime behavior-related errors constitutes the primary bottleneck behind the remaining unresolved cases.

**Task/Visual Score Analysis** In VisPlotBench, the execution pass rate and the Task/Visual Score are equally important evaluation dimensions. In Section 5.2, we analyze three representative languages that highlight different behaviors and discuss the phenomena of execution–semantic mismatch, symbolic grammar gains, and rendering library sensitivity. To provide additional insights into semantic correctness and visual alignment, we extend the analysis using the complete results in Appendix F, covering the remaining five languages and examining Task/Visual Score before and after self-debug as well as their relationship with execution pass rates.

i) **Supplementary Task/Visual Score analysis for the remaining languages:** (1) For `Python`, we observe that from Qwen2.5-Coder to VisCoder2, and further with self-debug enabled, the mean Task/Visual Score, the proportion of high-quality samples, and the execution pass rate improve in a largely synchronized manner. For example, at the 32B scale, VisCoder2-32B increases its Visual Score from 49 to 58 and its Task Score from 56 to 68, with the proportion of samples scoring visual $\geq 75$ rising from 42% to 46% and task $\geq 75$ from 54% to 62%. This shows that execution improvements are primarily driven by joint gains in semantic alignment and visual quality rather than by shallow syntax repairs. (2) In `Vega-Lite`, strong models such as GPT-4.1 and VisCoder2 already approach saturated performance after self-debug, yet Task/Visual Score still show mild upward trends; for example, for VisCoder2-32B the Visual Score increases from 60 to 62 and the Task Score from 70 to 72. This suggests that in declarative languages with explicit rules and rich diagnostics, models already produce mostly correct specifications, and self-debug mainly handles long-tail issues. (3) `Mermaid` shows notable gains across all metrics. For GPT-4.1: the Visual Score rises from 41 to 56, the Task Score from 57 to 77, and the proportion of high-scoring samples increases correspondingly. VisCoder2 consistently outperforms Qwen2.5-Coder at the same scale, and both visual and task performance improve steadily with self-debug, reflecting that in languages with explicit graph structures and detailed runtime diagnostics, models can use feedback to refine both relational semantics and diagram layout. (4) `Asymptote` remains one of the most challenging languages. Although most models show some improvement in Task/Visual Score, the overall level remains significantly lower than in Python or Vega-Lite. For example, VisCoder2-32B increases its Task Score from 46 to 53 after self-debug, while the Visual Score only rises from 27 to 31 with almost no change in high-score proportions, indicating that models can more easily align symbolic task semantics yet struggle to reliably control geometric details and rendering behavior, leaving a substantial gap between semantic correctness and visual consistency. (5) In `HTML`, strong models such as GPT-4.1 and VisCoder2 already exhibit high initial Task/Visual

Score, and self-debug leads only to moderate gains. For example, GPT-4.1 increases its Visual Score from 48 to 51 and its Task Score from 64 to 68, while VisCoder2-32B increases its Visual Score from 43 to 44 and its Task Score from 61 to 62. Overall, `HTML` specifications are relatively "model-friendly" in terms of semantic alignment, and remaining discrepancies are more reflective of minor visual deviations introduced by layout strategies and default rendering behaviors rather than true semantic errors.

ii) **Effects and limitations of self-debug on Task/Visual Score:** Across all eight languages, self-debug generally improves Task/Visual Score alongside execution reliability, with the clearest gains appearing in languages where models can leverage informative diagnostics. `Python` and Mermaid exemplify this trend: in `Python`, VisCoder2-32B improves from a Task Score of 56 to 68 and from a Visual Score of 49 to 58, while GPT-4.1 in Mermaid rises from 57 to 77 (task) and 41 to 56 (visual). These cases show that when feedback exposes meaningful semantic or structural signals, models can refine both correctness and rendered appearance rather than merely repairing syntax. In contrast, `LaTeX` and `Asymptote` show limited visual improvement despite higher execution success (e.g., VisCoder2-32B increases visual only from 27 to 31), reflecting that symbolic or compiler-dependent failures often surface only as coarse parser messages, offering insufficient guidance for deeper refinement. `SVG` represents the opposite extreme: execution and Task Score increase slightly, yet Visual Score remain almost unchanged (GPT-4.1 stays near 45; VisCoder2-32B increases only from 33 to 34), indicating that text-only feedback cannot reveal layout- or rendering-sensitive discrepancies. Taken together, these findings show that Task/Visual Score uncover a distinct layer of quality that execution alone cannot detect, and that the effectiveness of self-debug critically depends on how well feedback exposes semantically or visually meaningful error signals.

iii) **Cross-language relationship between Exec Pass and Task/Visual Score:** Examining all eight languages reveals two broad regimes. In `Python`, `Vega-Lite`, `Mermaid`, and `HTML`, execution success, task accuracy, and visual fidelity tend to rise together, indicating that most errors stem from structural or interface issues that, once resolved, directly translate into improved semantics and rendering; in these settings, execution pass rate is a reasonable proxy for overall quality. `LaTeX` and `SVG`, however, demonstrate clear decoupling: models often achieve much higher execution success without corresponding gains in semantic or visual alignment. For `LaTeX`, GPT-4.1 reaches a 66.1% execution pass rate after self-debug, yet Task/Visual Score remain in the mid-fifties and twenty-to-forty ranges due to macro expansion and compile-time fragility. For `SVG`, execution frequently saturates while Visual Score remain around forty to fifty across models, reflecting rendering-library sensitivity that feedback cannot capture. `Asymptote` lies between these extremes: all three metrics remain low but tend to improve synchronously, highlighting the intrinsic difficulty of symbolic geometric drawing and that performance on this language is still far from saturated. Together, these cross-language patterns show where execution is informative of downstream quality and where Task/Visual Score provide essential complementary signals.

## I.4 QUANTITATIVE ANALYSIS OF VISCODE-MULTI-679K

We provide more detailed descriptions of error rates, diversity, redundancy, and semantic alignment for the visualization portion of VisCode-Multi-679K.

For **Error Rates**, Table 28 and Table 29 shows the error proportions at each of the three stages in the visualization data pipeline.

Table 28: Error proportions across stages in the visualization data pipeline (Part I).
[Back to Appendix Contents]

| Stage/Language | Python | Vega-Lite | LilyPond | Mermaid | SVG | LaTeX |
|---|---|---|---|---|---|---|
| Initial Samples | 2,657,158 | 6,864 | 12,097 | 13,627 | 185,313 | 134,600 |
| Libs Filter ErrorRate | 91.90% | 0% | 0% | 0% | 57.16% | 0% |
| Code Extract ErrorRate | 0.10% | 0% | 0% | 0% | 0% | 0% |
| Runtime Validation ErrorRate | 13.03% | 1.08% | 0.03% | 1.81% | 41.27% | 7.85% |
| Final Valid Samples | 186,954 | 6,790 | 12,093 | 13,381 | 46,621 | 124,039 |

Table 29: Error proportions across stages in the visualization data pipeline (Part II).
[Back to Appendix Contents]

| Stage/Language | Asymptote | HTML | JavaScript | TypeScript | C++ | R |
|---|---|---|---|---|---|---|
| Initial Samples | 25,297 | 1,400,763 | 1,325,343 | 429,035 | 1,024,674 | 257,438 |
| Libs Filter ErrorRate | 0% | 87.12% | 93.03% | 96.75% | 92.30% | 87.77% |
| Code Extract ErrorRate | 0% | 0.40% | 21.90% | 5.61% | 3.18% | 20.75% |
| Runtime Validation ErrorRate | 10.90% | 24.73% | 60.07% | 52.08% | 78.05% | 46.15% |
| Final Valid Samples | 22,539 | 135,230 | 28,807 | 6,315 | 16,776 | 13,437 |

For **Diversity**, Table 30 reports the visualization-type distribution of VisCode-Multi-679K. The dataset covers 91 visualization types grouped into 15 categories, highlighting its broad coverage and diversity across multi-language visualization tasks.

Table 30: Distribution of visualization types in VisCode-Multi-679K.
[Back to Appendix Contents]

| VisType | Count | Category | VisType | Count | Category | VisType | Count | Category |
|---|---|---|---|---|---|---|---|---|
| area | 8,278 | Basic | surface-3d | 5,862 | Surface | network | 14,579 | Network |
| bar | 73,201 | Basic | flow-field | 382 | Surface | parallel-coordinates | 447 | Network |
| candlestick | 427 | Basic | surface | 5,994 | Surface | sankey | 1,036 | Network |
| donut | 980 | Basic | ternary-plot | 1,223 | Surface | annotation | 139 | Diagram |
| dotplot | 666 | Basic | vector-field | 6,194 | Surface | class-diagram | 542 | Diagram |
| grid | 13,944 | Basic | volume-render | 1,998 | Surface | er-diagram | 702 | Diagram |
| line | 53,992 | Basic | animation-frame | 1,744 | Temporal | flowchart | 4,855 | Diagram |
| pie | 19,064 | Basic | event | 230 | Temporal | generic-diagram | 12,055 | Diagram |
| polar | 3,951 | Basic | timeseries | 2,599 | Temporal | gitgraph | 662 | Diagram |
| radar | 790 | Basic | timeline | 3,663 | Temporal | other-diagram | 19,060 | Diagram |
| rectangle | 3,896 | Basic | calendar | 301 | Schedule | quadrant-chart | 696 | Diagram |
| rule | 1,538 | Basic | gantt | 2,304 | Schedule | requirement-diagram | 600 | Diagram |
| scatter | 29,628 | Basic | bubble | 22,040 | Relation | sequence-diagram | 606 | Diagram |
| spike-line | 199 | Basic | jointplot | 577 | Relation | state-diagram | 1,122 | Diagram |
| streamgraph | 228 | Basic | regression | 2,044 | Relation | dataflow-diagram | 1,295 | Diagram |
| density | 172 | Distribution | category-chart | 510 | Categorical | circle | 834 | Geometry |
| hexbin | 1,294 | Distribution | funnel | 1,314 | Categorical | circuit | 10,233 | Geometry |
| histogram | 11,354 | Distribution | gauge | 297 | Categorical | geometry | 23,727 | Geometry |
| kde | 9,855 | Distribution | indicator | 1,623 | Categorical | reference-shape | 4,579 | Geometry |
| rug | 367 | Distribution | waterfall | 916 | Categorical | choropleth | 8,244 | Map |
| swarm | 941 | Distribution | wordcloud | 281 | Categorical | dot-map | 5,053 | Map |
| box | 11,097 | Box | cluster | 609 | Hierarchy | symbol-map | 1,773 | Map |
| errorbar | 1,734 | Box | dendrogram | 630 | Hierarchy | geographic-line-map | 915 | Map |
| interval | 660 | Box | icicle | 647 | Hierarchy | geospatial-plot | 9,028 | Map |
| violin | 1,726 | Box | mindmap | 923 | Hierarchy | document-page | 10,432 | Document |
| correlationmatrix | 811 | Matrix | sunburst | 789 | Hierarchy | image | 13,140 | Document |
| contourmatrix | 2,424 | Matrix | tree | 2,062 | Hierarchy | math-box | 582 | Document |
| heatmap | 11,070 | Matrix | treemap | 1,780 | Hierarchy | problem-box | 10,924 | Document |
| adjacencymatrix | 1,075 | Matrix | arcdiagram | 427 | Network | table | 51,821 | Document |
| splom | 409 | Matrix | chord | 577 | Network | text | 8,302 | Document |
| | | | | | | textbox | 58,688 | Document |

For **Redundancy**, the upstream corpora used to construct VisCode-Multi-679K (Stack-Edu/Smol-IDs, CoSyn-400K, and SVG-diagrams) were released with deduplication applied. Building on these sources, we further quantify redundancy within VisCode-Multi-679K by running additional checks on its visualization-specific fields. Concretely, we normalize the instruction text and the code snippet separately, and apply SHA-1 fingerprinting to detect exact duplicates. The resulting redundancy statistics are summarized in Table 31 and Table 32.

Table 31: Global redundancy statistics for visualization samples.
[Back to Appendix Contents]

| Component | Total Samples | Unique Samples | Redundant Samples | Redundancy Rate |
|---|---|---|---|---|
| Instruction | 612,982 | 611,671 | 1,311 | 0.21% |
| Code | 612,982 | 594,382 | 18,600 | 3.03% |

Table 32: Per-language redundancy for visualization samples.
[Back to Appendix Contents]

| Language | Samples | Instr. Redundancy (%) | Code Redundancy (%) |
|---|---|---|---|
| Asymptote | 22,539 | 0.00 | 0.38 |
| C++ | 16,776 | 0.00 | 0.92 |
| HTML | 135,230 | 0.00 | 0.28 |
| JavaScript | 28,807 | 0.00 | 8.33 |
| LaTeX | 124,039 | 0.00 | 0.00 |
| LilyPond | 12,093 | 0.00 | 0.00 |
| Mermaid | 13,381 | 0.00 | 0.93 |
| Python | 186,954 | 0.70 | 8.24 |
| R | 13,437 | 0.00 | 0.15 |
| SVG | 46,621 | 0.00 | 0.00 |
| TypeScript | 6,315 | 0.00 | 0.55 |
| Vega-Lite | 6,790 | 0.00 | 0.00 |

Overall, instruction redundancy is 0.21% and code redundancy is 3.03%. Redundancy remains low across all visualization languages, indicating that VisCode-Multi-679K contains diverse natural-language instructions and executable visualization code rather than being dominated by duplicated samples.

## J    CASE STUDY

In this section, we present a set of representative examples from VisCoder2-32B to illustrate model behavior across the eight visualization languages.

### J.1    PYTHON: SUCCESSFUL GENERATION

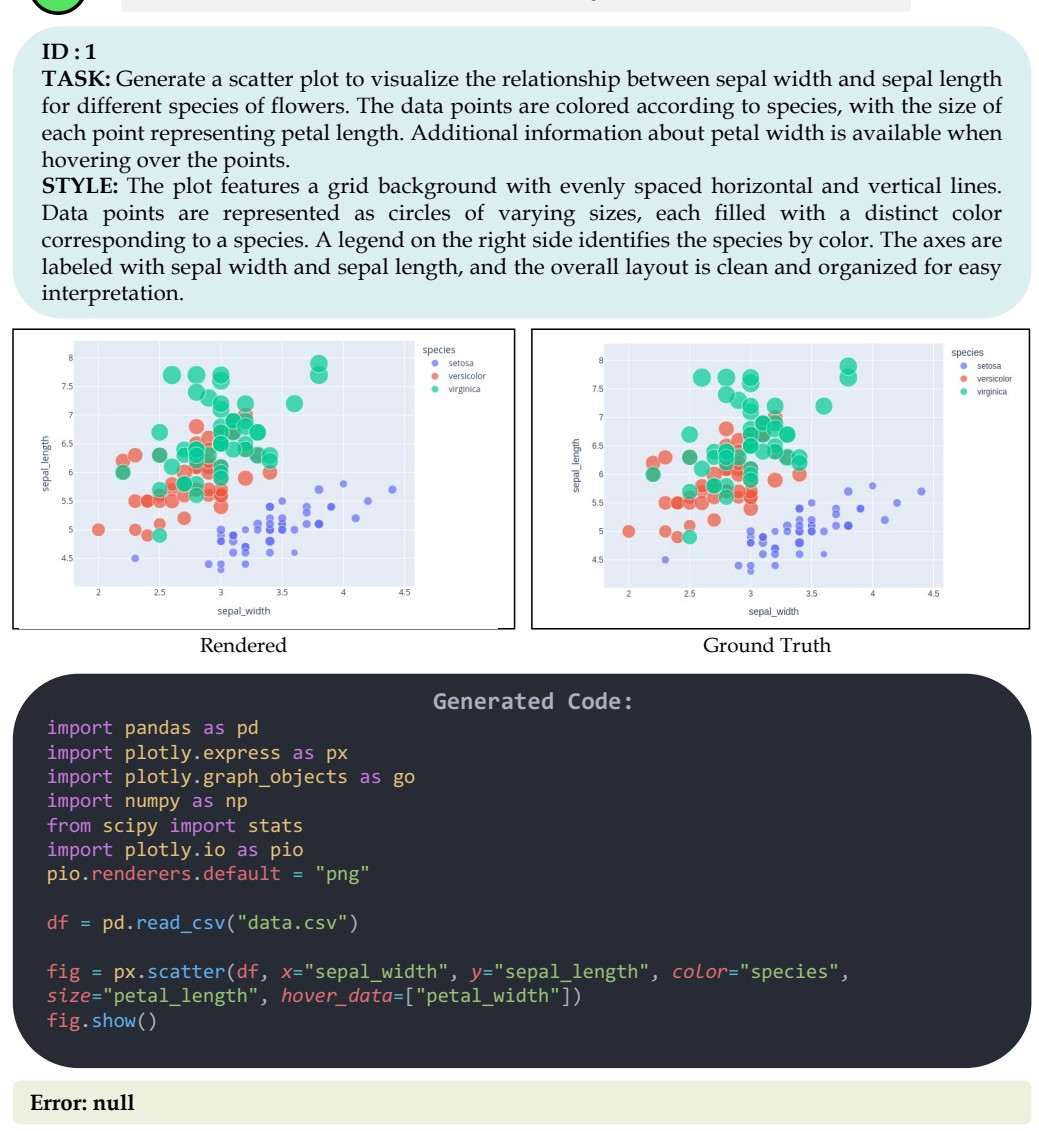

Figure 5: Example of a successful generation in **Python** (ID: 1). The model generates code that executes successfully and produces a plot consistent with the ground truth.

## J.2 PYTHON: SELF-DEBUG RECOVERY

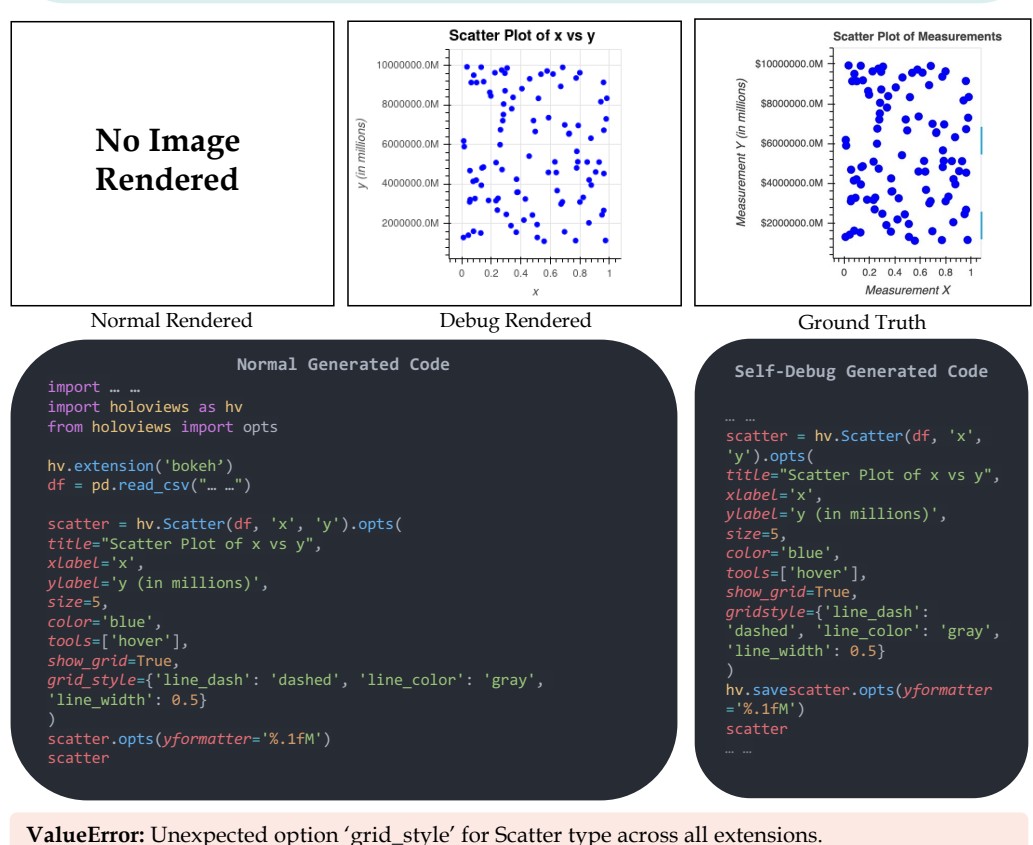

Figure 6: Example of a failed generation in **Python** (ID: 69), where the initial code raises a **ValueError** and is resolved in the **first** round of self-debug, resulting in a corrected plot that matches the intended semantics.
[Back to Appendix Contents]

## J.3 PYTHON: SELF-DEBUG FAILED

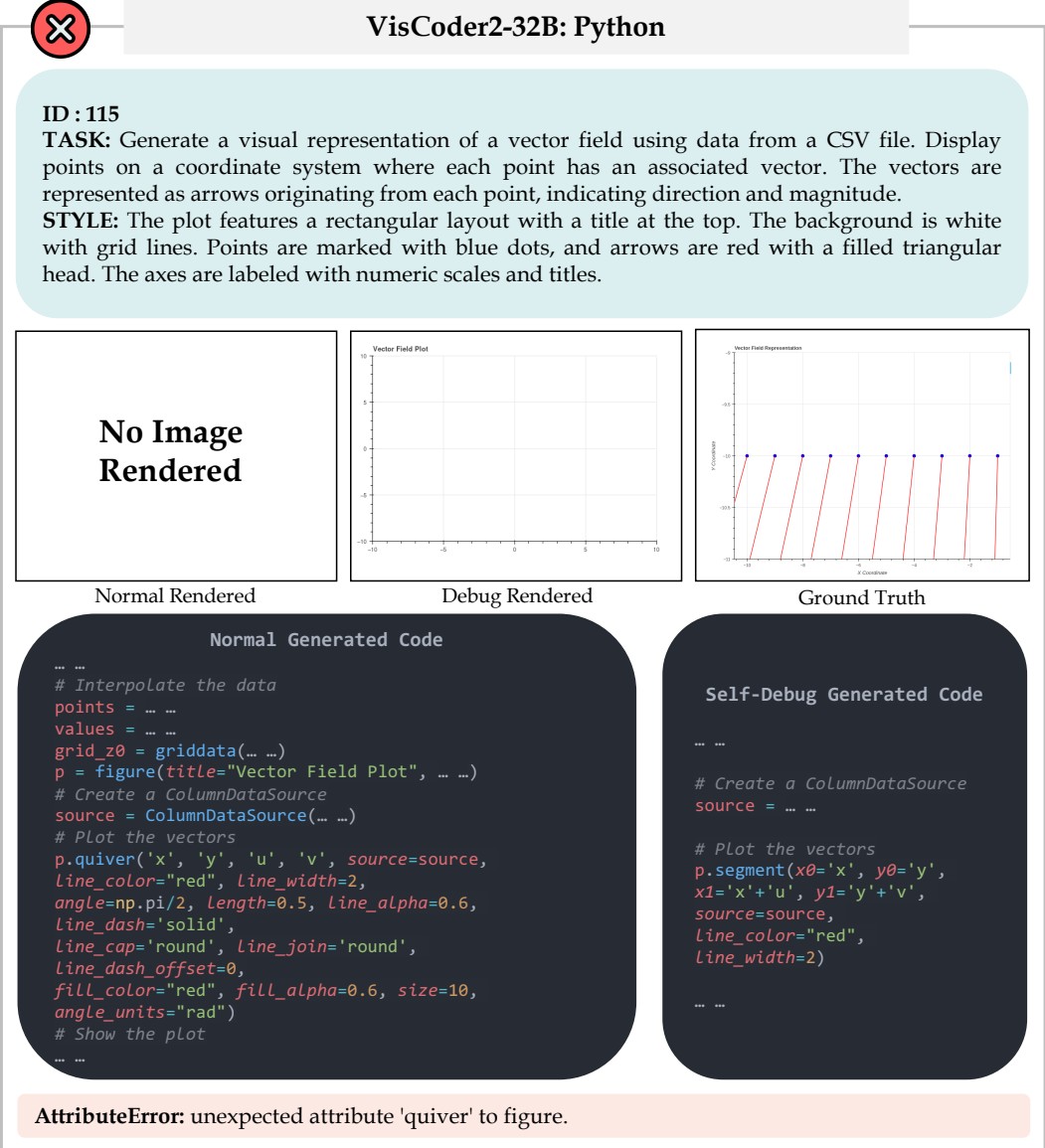

Figure 7: Example of a failed generation in **Python** (ID: 115), where the initial code raises a **AttributeError** and is still failed after three rounds self-debug.
[Back to Appendix Contents]

J.4 VEGA-LITE: SUCCESSFUL GENERATION

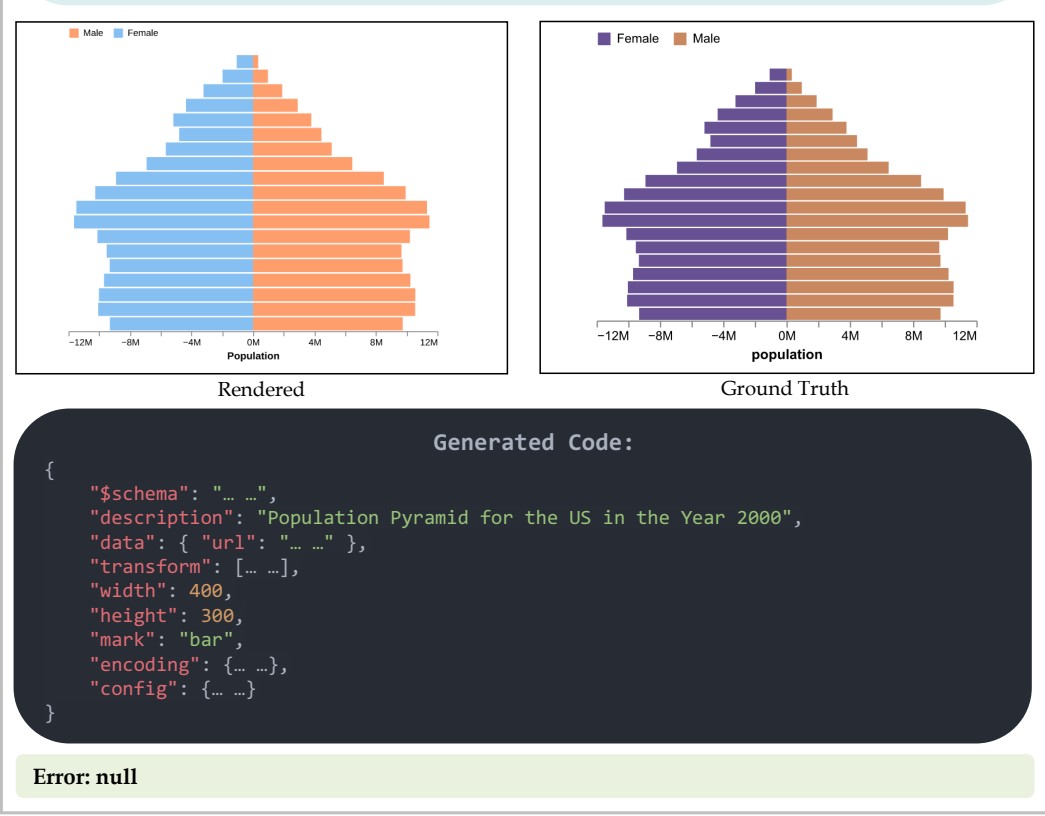

Figure 8: Example of a successful generation in **Vega-Lite** (ID: 18). The model generates code that executes successfully and produces a plot consistent with the ground truth.
[Back to Appendix Contents]

J.5 VEGA-LITE: SELF-DEBUG RECOVERY

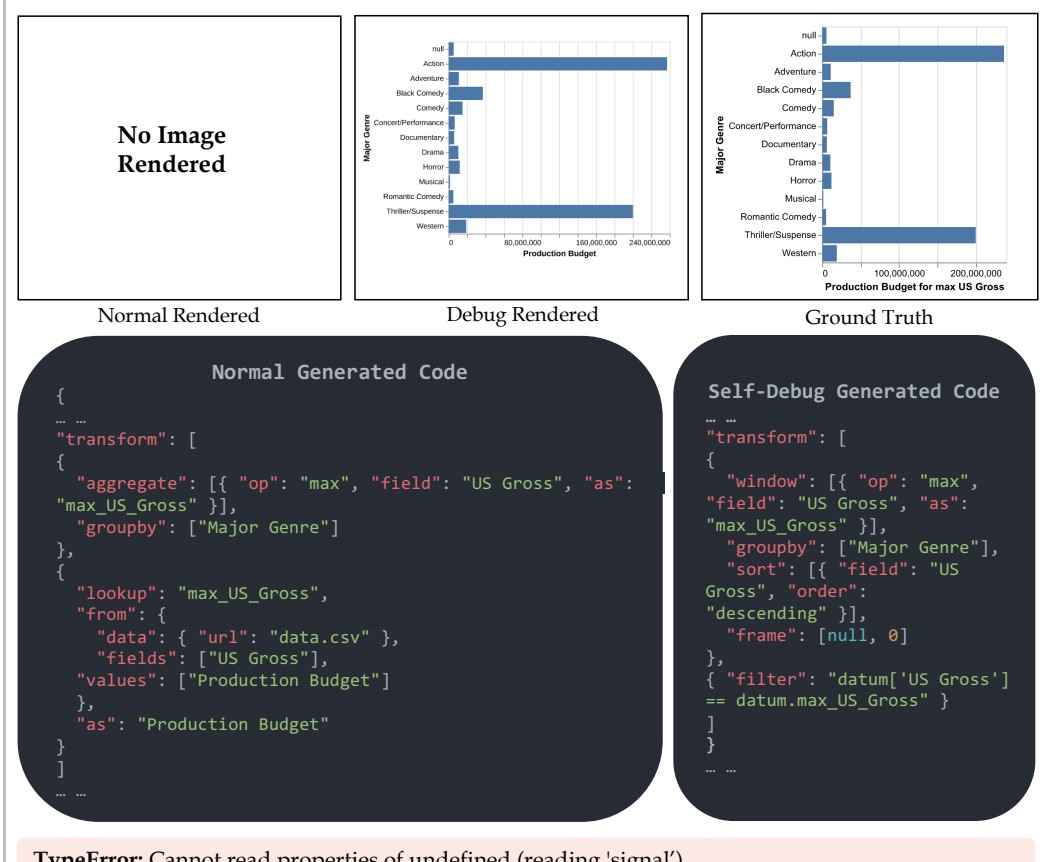

Figure 9: Example of a failed generation in **Vega-Lite** (ID: 14), where the initial code raises a **TypeError** and is resolved in the **second** round of self-debug, resulting in a corrected plot that matches the intended semantics.

[Back to Appendix Contents]

## J.6 VEGA-LITE: SELF-DEBUG FAILED

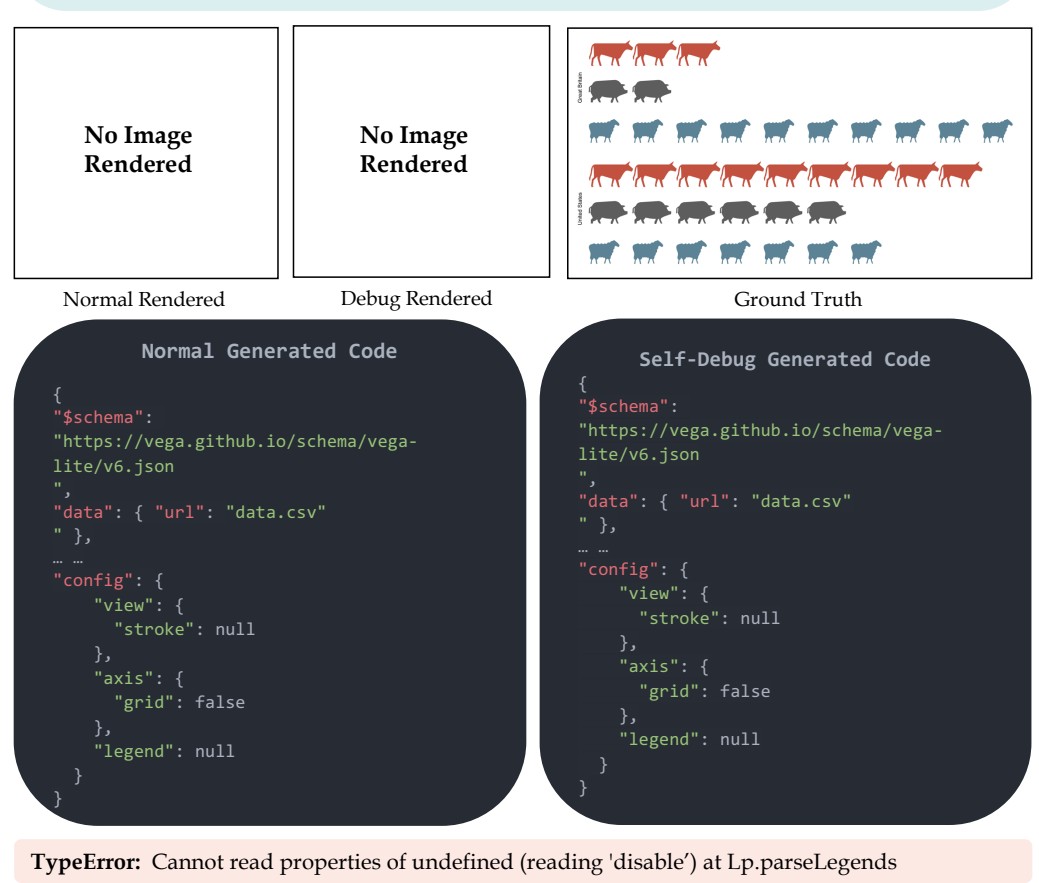

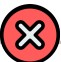

**VisCoder2-32B: Vega-Lite**

**ID : 50**
**TASK:** Create a grid of point plots that represents different animals categorized by country. Each row corresponds to a country, and within each row, different animals are represented by distinct shapes. The animals are visually encoded using both shape and color to differentiate between categories such as "cattle," "pigs," and "sheep." The plot does not display axes, focusing solely on the visual arrangement of shapes to convey the data.
**STYLE:** The visual consists of a series of rows, each labeled with a country name on the left. Within each row, animal shapes are arranged horizontally, with each shape filled with a specific color corresponding to the animal type. The shapes are evenly spaced and sized uniformly, creating a clean and organized appearance. The absence of axes and legends emphasizes the visual comparison of animal types across different countries.

| No Image Rendered | No Image Rendered | |
|---|---|---|
| Normal Rendered | Debug Rendered | Ground Truth |

**Normal Generated Code**

```
{
"$schema":
"https://vega.github.io/schema/vega-
lite/v6.json
",
"data": { "url": "data.csv
" },
… …
"config": {
    "view": {
      "stroke": null
    },
    "axis": {
      "grid": false
    },
    "legend": null
  }
}
```

**Self-Debug Generated Code**

```
{
"$schema":
"https://vega.github.io/schema/vega-
lite/v6.json
",
"data": { "url": "data.csv
" },
… …
"config": {
    "view": {
      "stroke": null
    },
    "axis": {
      "grid": false
    },
    "legend": null
  }
}
```

**TypeError:** Cannot read properties of undefined (reading 'disable') at Lp.parseLegends

Figure 10: Example of a failed generation in **Vega-Lite** (ID: 50), where the initial code raises a **TypeError** and is still failed after three rounds self-debug.
[Back to Appendix Contents]

## J.7 LILYPOND: SUCCESSFUL GENERATION

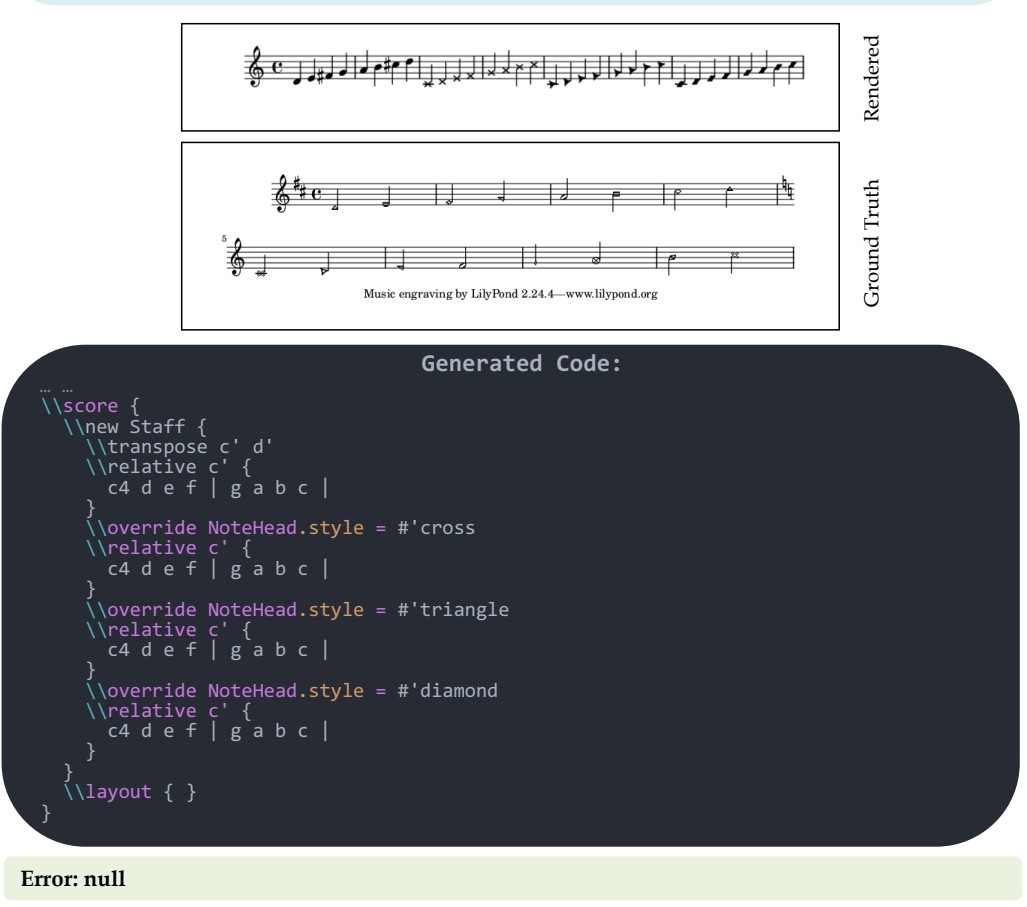

Figure 11: Example of a successful generation in **Lilypond** (ID: 15). The model generates code that executes successfully and produces a plot consistent with the ground truth.

## J.8 LILYPOND: SELF-DEBUG RECOVERY

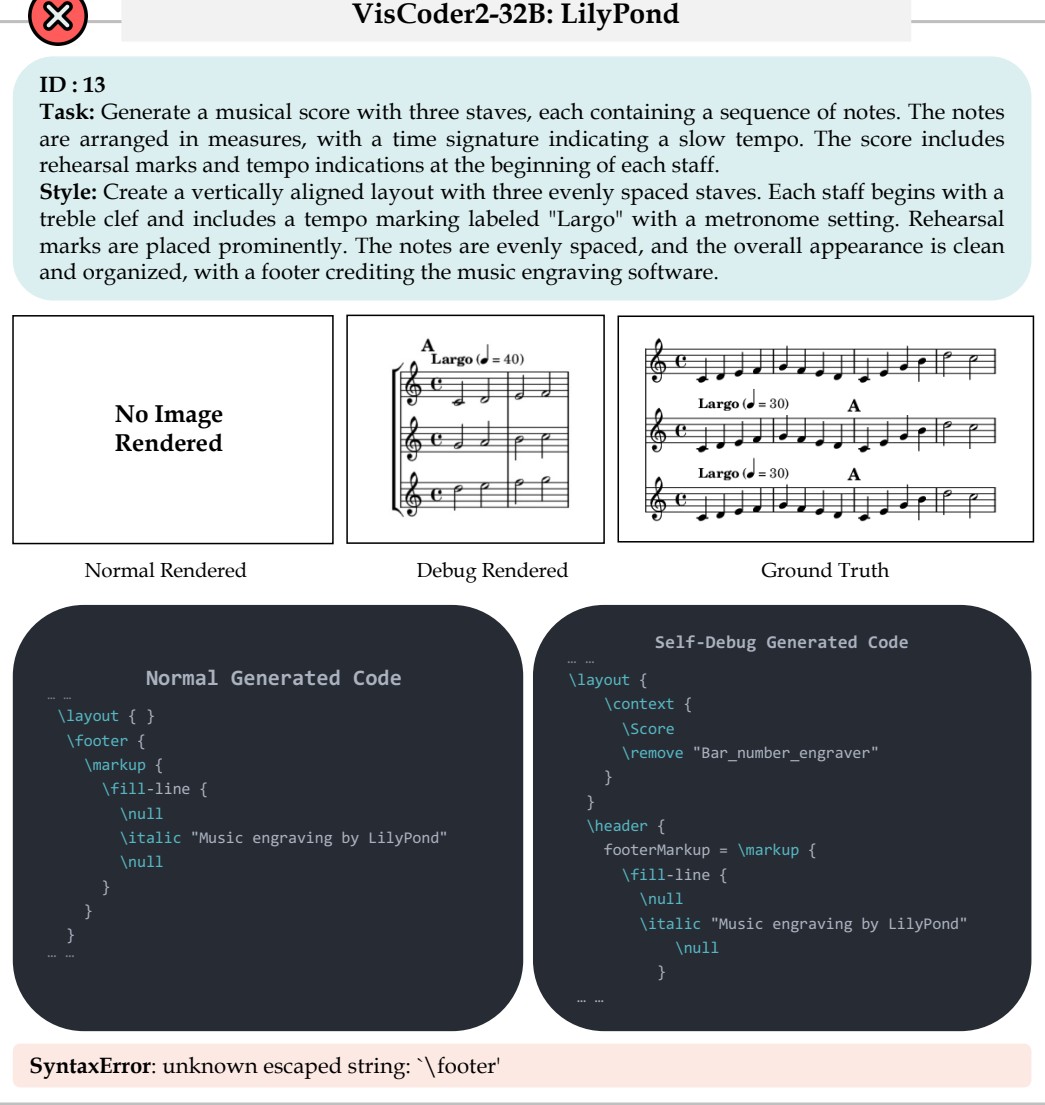

Figure 12: Example of a failed generation in **Lilypond** (ID: 13), where the initial code raises a **SyntaxError** and is resolved in the **first** round of self-debug, resulting in a corrected plot that matches the intended semantics.

[Back to Appendix Contents]

## J.9 LILYPOND: SELF-DEBUG FAILED

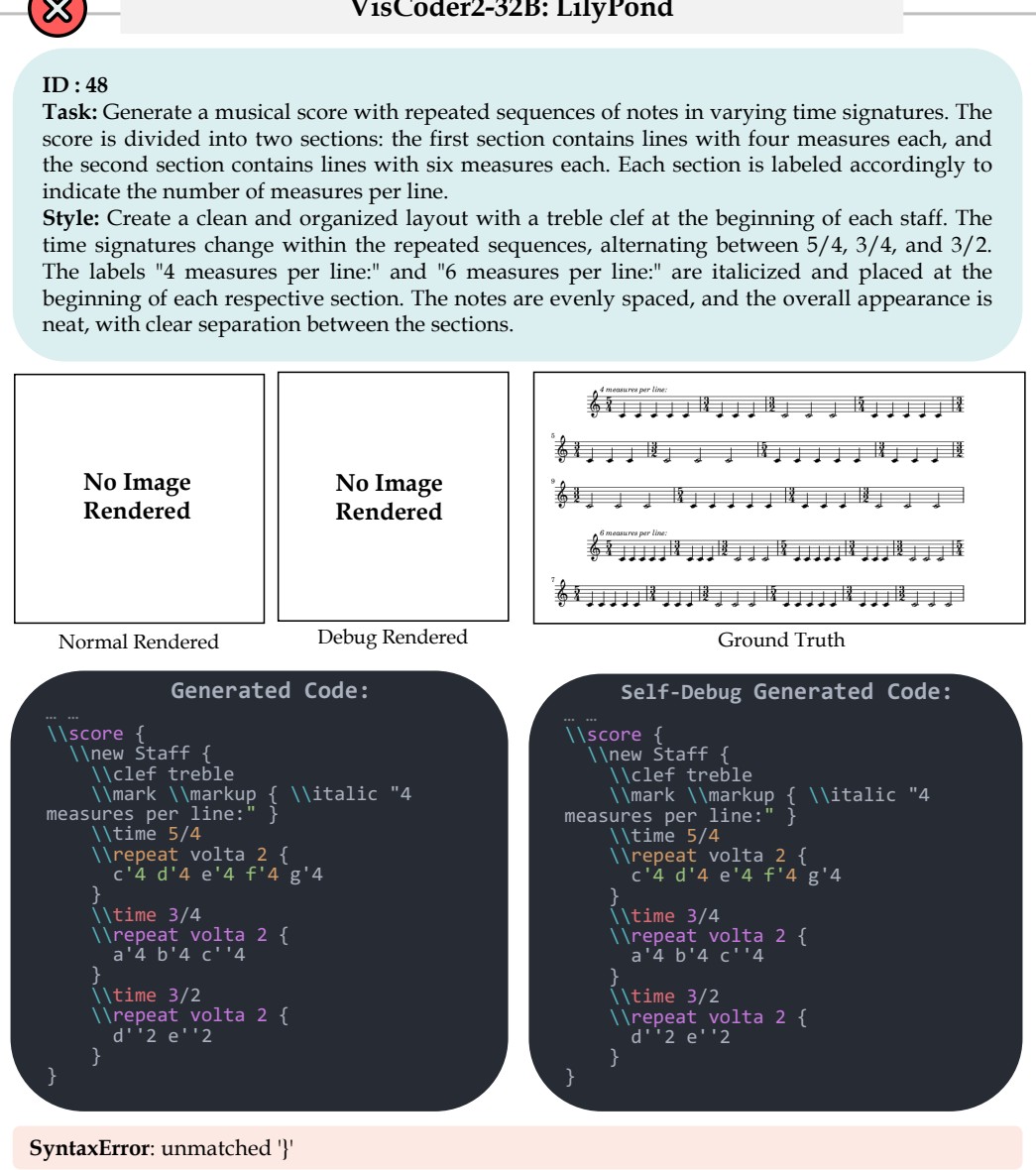

Figure 13: Example of a failed generation in **Lilypond** (ID: 48), where the initial code raises a **TypeError** and is still failed after three rounds self-debug.
[Back to Appendix Contents]

## J.10 MERMAID: SUCCESSFUL GENERATION

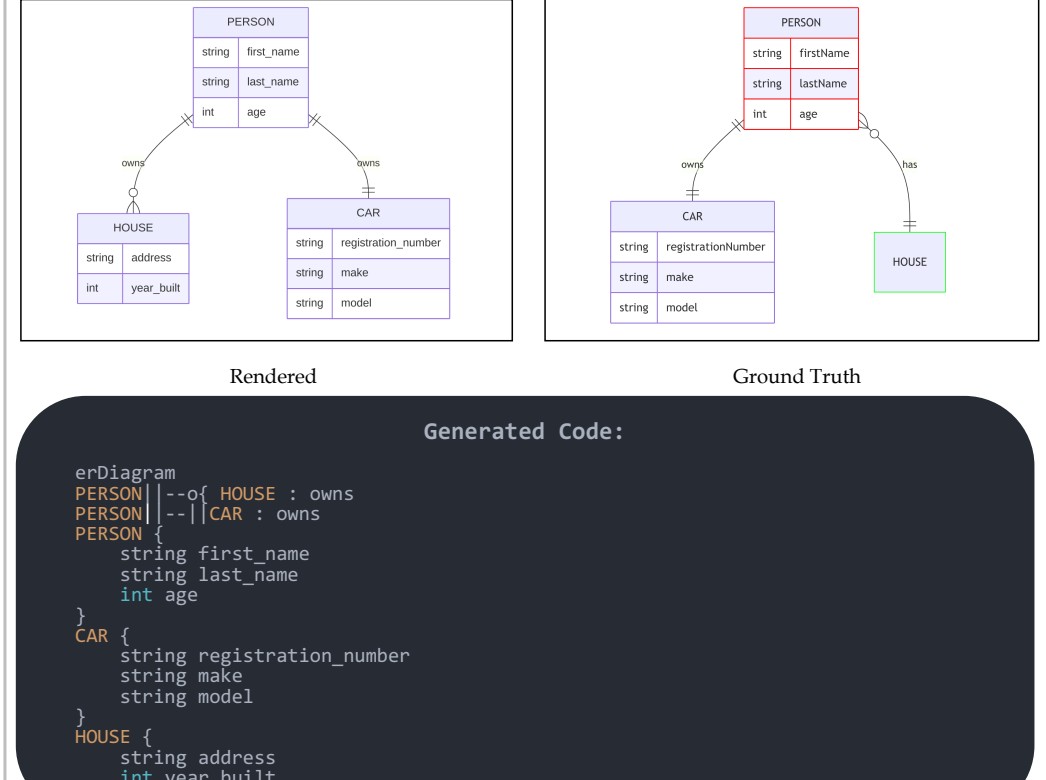

**ID : 19**

**Task:** Generate an entity-relationship diagram illustrating the relationships between three entities: PERSON, CAR, and HOUSE. The PERSON entity includes attributes for first name, last name, and age. The CAR entity includes attributes for registration number, make, and model. The diagram shows a one-to-one relationship between PERSON and CAR, indicating that a person owns a car. Additionally, there is a one-to-many relationship between PERSON and HOUSE, indicating that a person can have multiple houses.

**Style:** The diagram features rectangular boxes for each entity, with the PERSON entity outlined in red, the CAR entity in purple, and the HOUSE entity in green. Each box contains rows listing the attributes, with data types specified. Lines connect the entities, with labels indicating the nature of the relationships. The lines have different end markers to represent the type of relationship, and the overall layout is clean and organized with a clear visual hierarchy.

Rendered                           Ground Truth

**Generated Code:**

```
erDiagram
PERSON||--o{ HOUSE : owns
PERSON||--||CAR : owns
PERSON {
    string first_name
    string last_name
    int age
}
CAR {
    string registration_number
    string make
    string model
}
HOUSE {
    string address
    int year_built
}
```

**Error: null**

Figure 14: Example of a successful generation in **Mermaid** (ID: 19). The model generates code that executes successfully and produces a plot consistent with the ground truth.
[Back to Appendix Contents]

## J.11 MERMAID: SELF-DEBUG RECOVERY

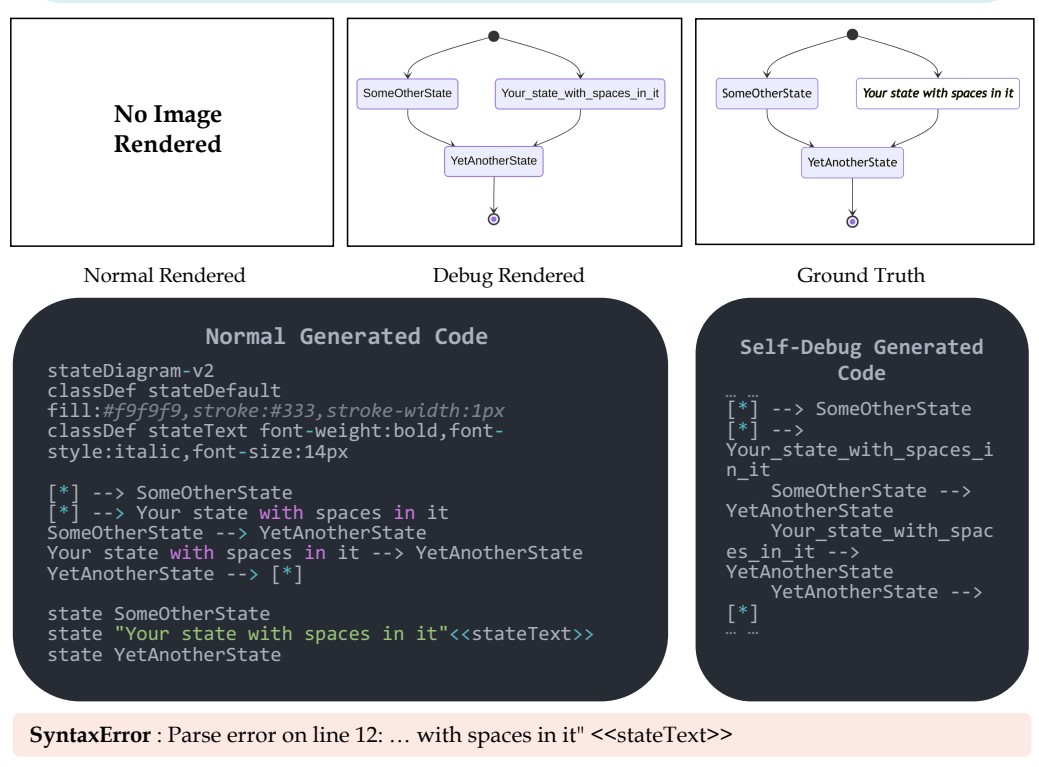

Figure 15: Example of a failed generation in **Mermaid** (ID: 88), where the initial code raises a **SyntaxError** and is resolved in the **second** round of self-debug, resulting in a corrected plot that matches the intended semantics.

[Back to Appendix Contents]

## J.12 MERMAID: SELF-DEBUG FAILED

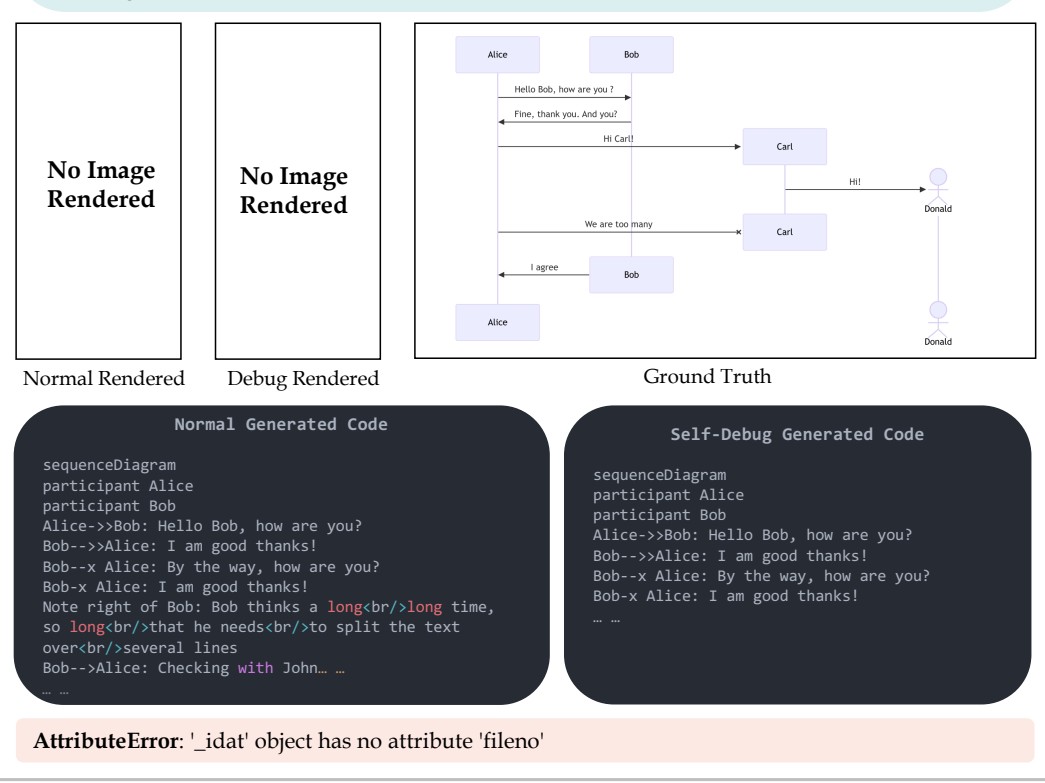

**ID : 80**
**Task:** Generate a sequence diagram illustrating interactions between participants named Alice, Bob, Carl, and an actor named Donald. The diagram depicts a conversation where Alice greets Bob, and they exchange pleasantries. Alice then introduces Carl, who subsequently greets Donald. The diagram also shows the destruction of Carl and Bob, with Alice commenting on the number of participants and Bob agreeing.
**Style:** The visual consists of horizontal arrows connecting labeled rectangular boxes representing participants, with a stick figure symbolizing the actor. The boxes and arrows are aligned vertically, showing the flow of communication. Messages are displayed as text along the arrows, and the destruction of participants is marked by an 'x' on the arrows. The layout is clean, with a light color scheme and clear separation between interactions.

Normal Rendered    Debug Rendered    Ground Truth

**Normal Generated Code**
```
sequenceDiagram
participant Alice
participant Bob
Alice->>Bob: Hello Bob, how are you?
Bob-->>Alice: I am good thanks!
Bob--x Alice: By the way, how are you?
Bob-x Alice: I am good thanks!
Note right of Bob: Bob thinks a long
long time,
so long
that he needs
to split the text
over
several lines
Bob-->Alice: Checking with John… …
… …
```

**Self-Debug Generated Code**
```
sequenceDiagram
participant Alice
participant Bob
Alice->>Bob: Hello Bob, how are you?
Bob-->>Alice: I am good thanks!
Bob--x Alice: By the way, how are you?
Bob-x Alice: I am good thanks!
… …
```

**AttributeError**: '_idat' object has no attribute 'fileno'

Figure 16: Example of a failed generation in **Mermaid** (ID: 80), where the initial code raises a **AttributeError** and is still failed after three rounds self-debug.
[Back to Appendix Contents]

## J.13 SVG: SUCCESSFUL GENERATION

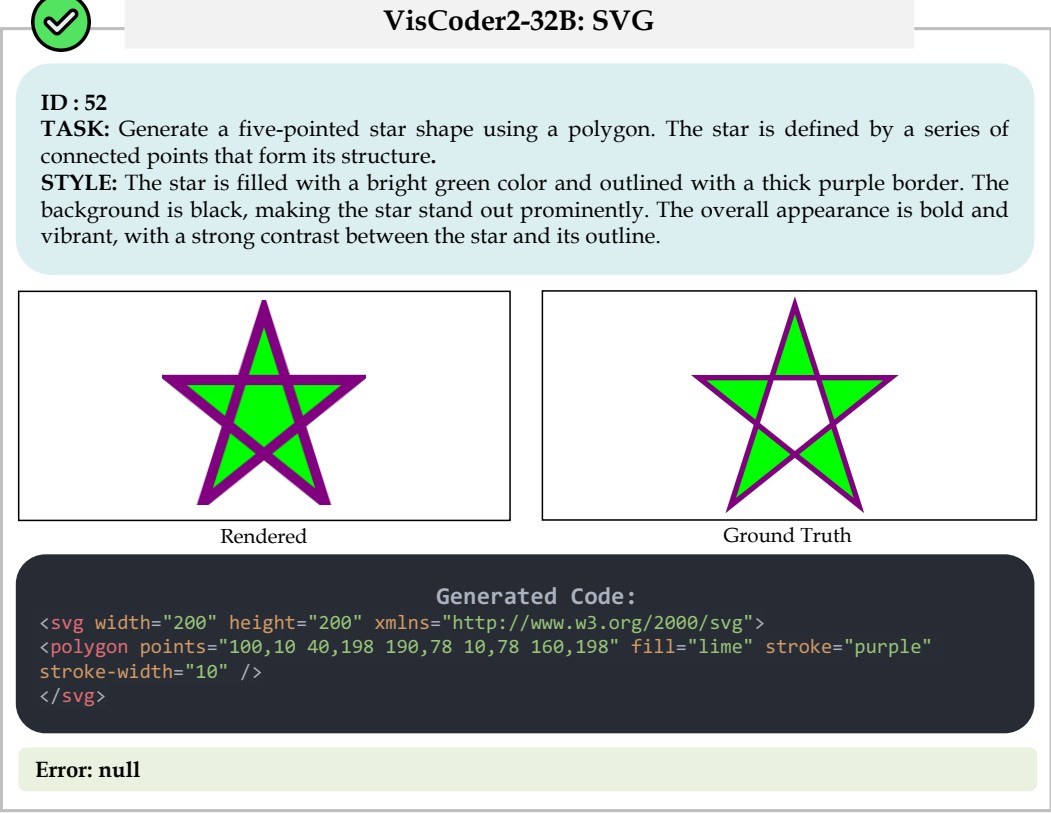

Figure 17: Example of a successful generation in **SVG** (ID: 52). The model generates code that executes successfully and produces a plot consistent with the ground truth.
[Back to Appendix Contents]

## J.14 SVG: SELF-DEBUG RECOVERY

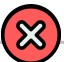

**VisCoder2-32B: SVG**

**ID : 42**

**TASK:** Generate a flowchart illustrating the process for producing a W3C Recommendation. The flowchart includes stages such as Working Draft (WD), Candidate Recommendation (CR), Proposed Recommendation (PR), and W3C Recommendation (REC). It also includes decision points and possible transitions between these stages, such as advancing to the next stage, returning to a previous stage, or publishing as a note. Each stage and decision point is labeled with descriptive text.

**STYLE:** Create a visual with elliptical nodes representing different stages, connected by arrows indicating the flow of the process. The nodes are labeled with abbreviations like WD, CR, PR, and REC, and are filled with white. Text annotations provide additional context for each transition, using a small font size. Arrows are solid or dashed, with varying colors to indicate different types of transitions. Some elements are partially transparent to indicate optional paths. The layout is horizontal, with nodes aligned in a sequence from left to right.

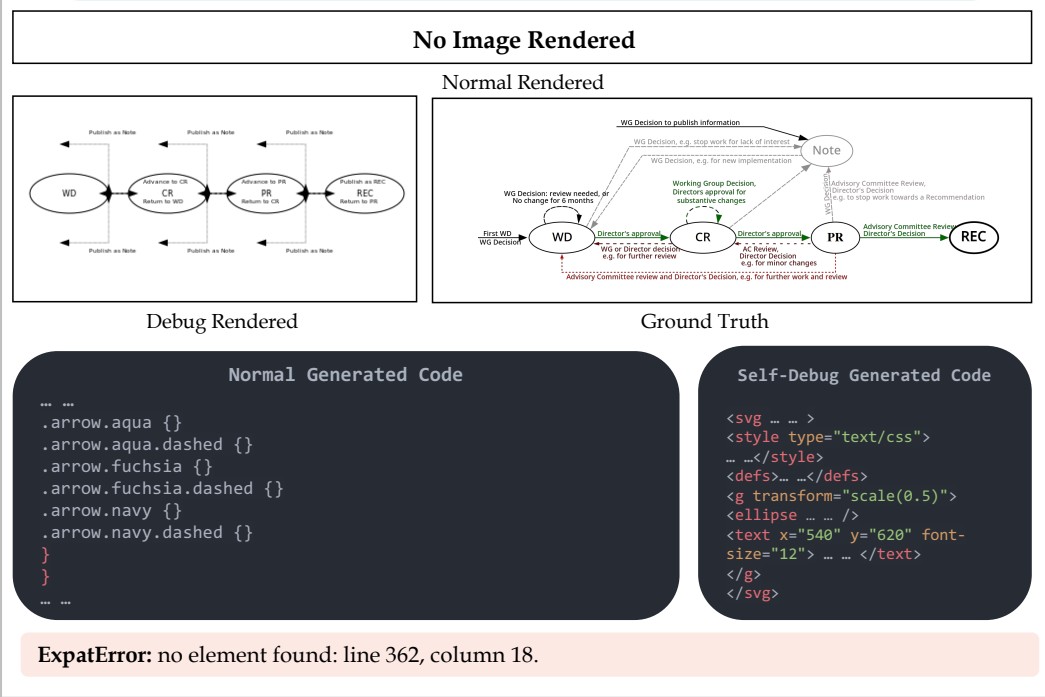

**ExpatError:** no element found: line 362, column 18.

Figure 18: Example of a failed generation in **SVG** (ID: 42), where the initial code raises a **ExPatError** and is resolved in the **first** round of self-debug, resulting in a corrected plot that matches the intended semantics.

[Back to Appendix Contents]

## J.15  SVG: SELF-DEBUG FAILED

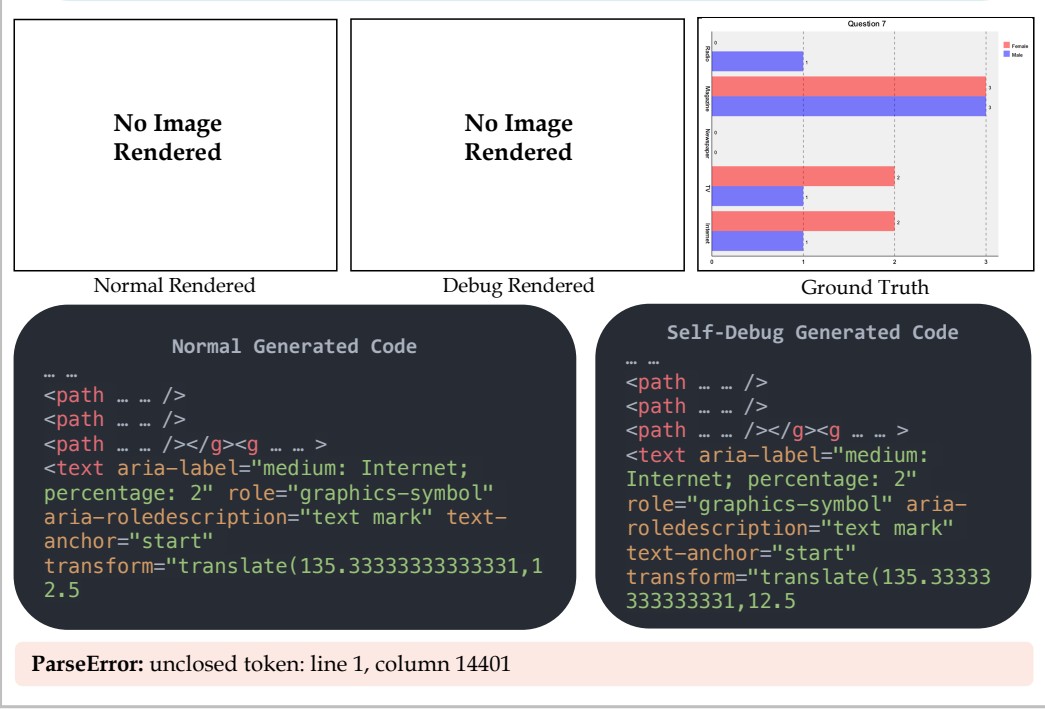

Figure 19: Example of a failed generation in **SVG** (ID: 12), where the initial code raises a
**ParseError** and is still failed after three rounds self-debug.
[Back to Appendix Contents]

## J.16 LATEX: SUCCESSFUL GENERATION

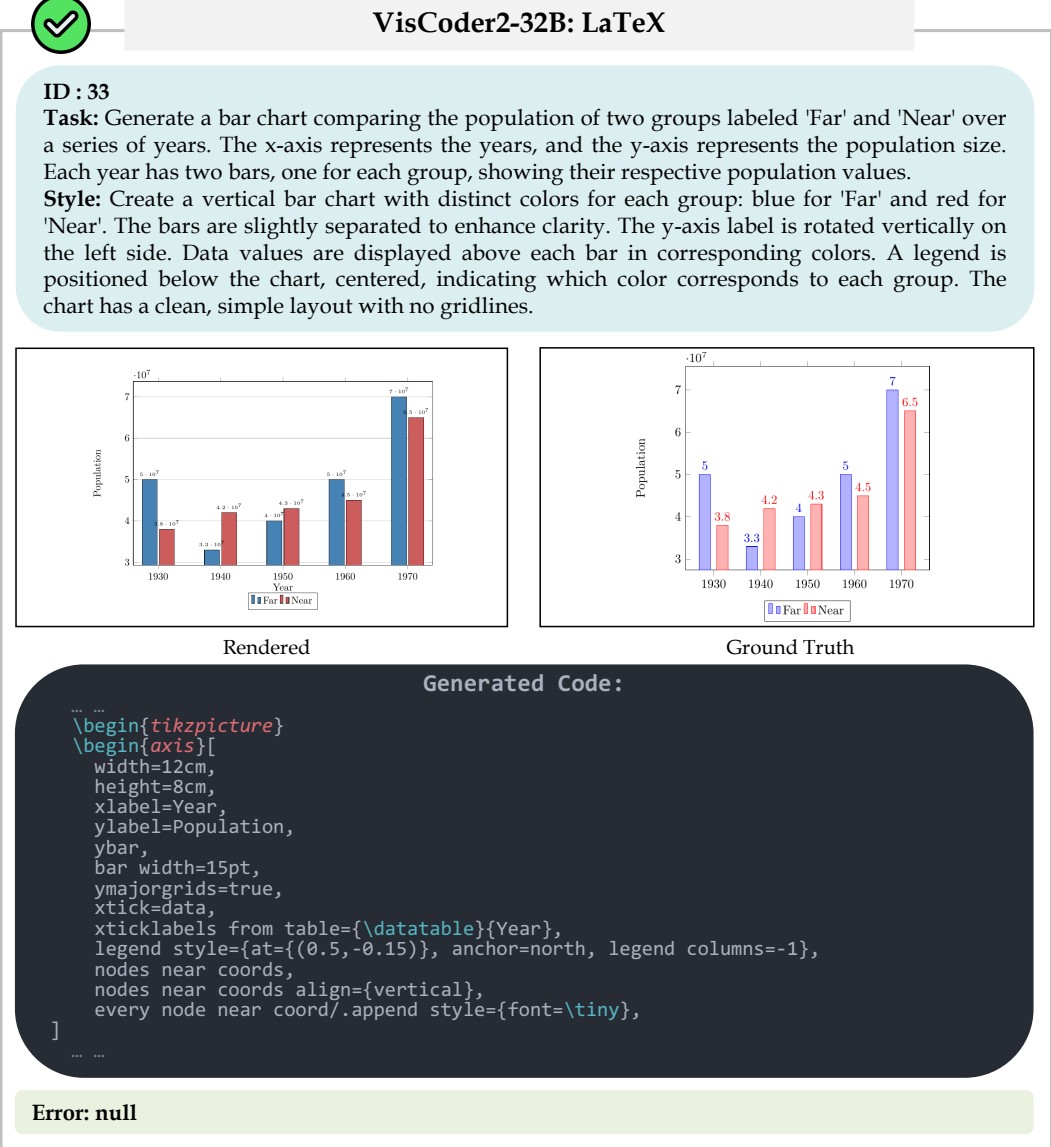

Figure 20: Example of a successful generation in **LaTeX** (ID: 33). The model generates code that executes successfully and produces a plot consistent with the ground truth.

## J.17 LaTeX: Self-Debug Recovery

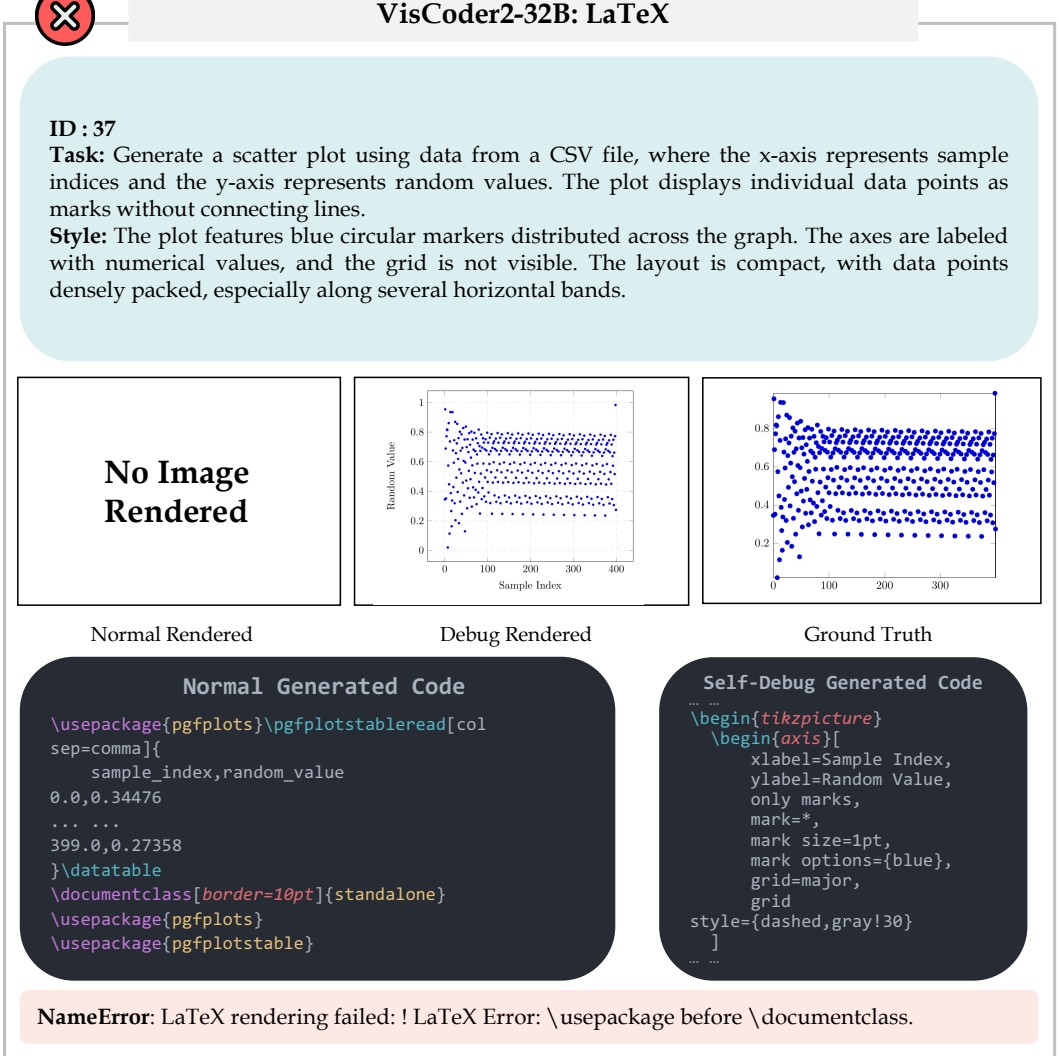

Figure 21: Example of a failed generation in **LaTeX** (ID: 37), where the initial code raises a **NameError** and is resolved in the **second** round of self-debug, resulting in a corrected plot that matches the intended semantics.

[Back to Appendix Contents]

## J.18 LATEX: SELF-DEBUG FAILED

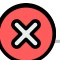

### VisCoder2-32B: LaTeX

**ID : 97**
**Task:** Generate a schematic diagram of an electrical circuit featuring a voltage source labeled V_s, an inductor labeled L_x, a capacitor labeled C_1, and another inductor labeled L_m on the primary side. The circuit includes a transformer with a turns ratio N_1:N_2 , a diode, a capacitor labeled C_2 , and a resistor labeled R  on the secondary side. The output voltage is denoted as v_o(t) .
**Style:** Create a clean and organized layout with components aligned horizontally. The circuit elements are connected with straight lines, and labels are placed near each component for clarity. The transformer is depicted with coils and a turns ratio label. The diagram uses a simple, monochrome style with a consistent line thickness and small font size for labels.

| No Image Rendered | No Image Rendered | Ground Truth circuit diagram |
|---|---|---|
| Normal Rendered | Debug Rendered | Ground Truth |

**Normal Generated Code**

```
… …
\begin{circuitikz}[scale=0.8]
    \draw (0,0)
    to[vsource, l=$V_s$] (0,4)
    to[L, l=$L_x$] (4,4)
    to[C, l=$C_1$] (4,0) -- (0,0);

    \draw (4,4) -- (6,4)
    to[cute inductor, l=$L_m$] (6,0) -- (4,0);

    \draw (6,4) -- (8,4)
    to[transformer core] (8,0) -- (6,0);

    \draw (9,3.5) to[diode] (11,3.5)
    to[C, l=$C_2$] (11,0.5) -- (9,0.5)
    to[R, l=$R$] (9,3.5);

    \draw (11,3.5) -- (12,3.5)
    node[right] {$v_o(t)$};

    \node[above] at (7,4) {$N_1:N_2$};
\end{circuitikz}
\end{document}
… …
```

**Self-Debug Generated Code**

```
… …
\begin{circuitikz}[scale=0.8]
    \draw (0,0)
    to[vsource, l=$V_s$] (0,4)
    to[L, l=$L_x$] (4,4)
    to[C, l=$C_1$] (4,0) -- (0,0);

    \draw (4,4) -- (6,4)
    to[cute inductor, l=$L_m$] (6,0) -- (4,0);

    \draw (6,4) -- (8,4)
    to[transformer core] (8,0) -- (6,0);

    \draw (9,3.5) to[diode] (11,3.5)
    to[C, l=$C_2$] (11,0.5) -- (9,0.5)
    to[R, l=$R$] (9,3.5);

    \draw (11,3.5) -- (12,3.5)
    node[right] {$v_o(t)$};

    \node[above] at (7,4) {$N_1:N_2$};
\end{circuitikz}
\end{document}
… …
```

**NameError**: ! Package circuitikz Error: Unknown circuitikz element `vsource'.

Figure 22: Example of a failed generation in **LaTeX** (ID: 97), where the initial code raises a **NameError** and is still failed after three rounds self-debug.
[Back to Appendix Contents]

## J.19 ASYMPTOTE: SUCCESSFUL GENERATION

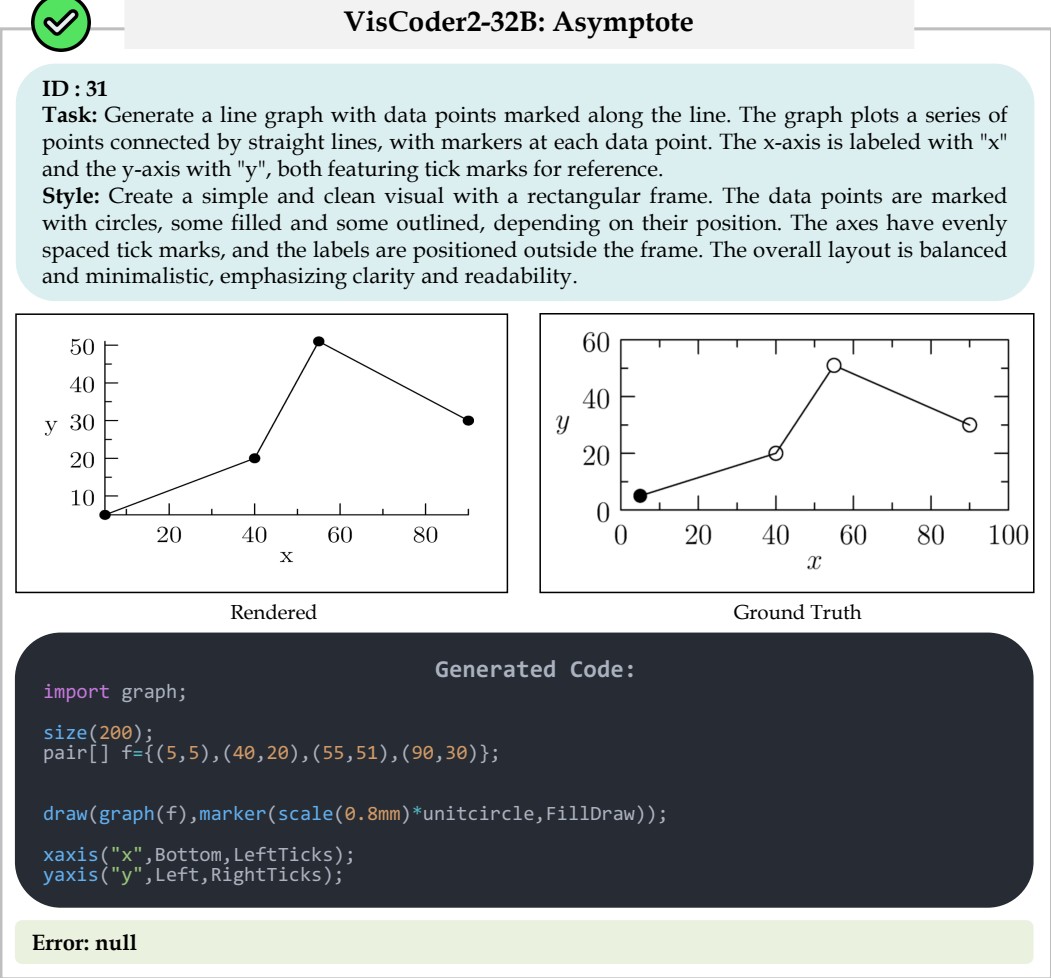

Figure 23: Example of a successful generation in **Asymptote** (ID: 31). The model generates code that executes successfully and produces a plot consistent with the ground truth.
[Back to Appendix Contents]

## J.20 ASYMPTOTE: SELF-DEBUG RECOVERY

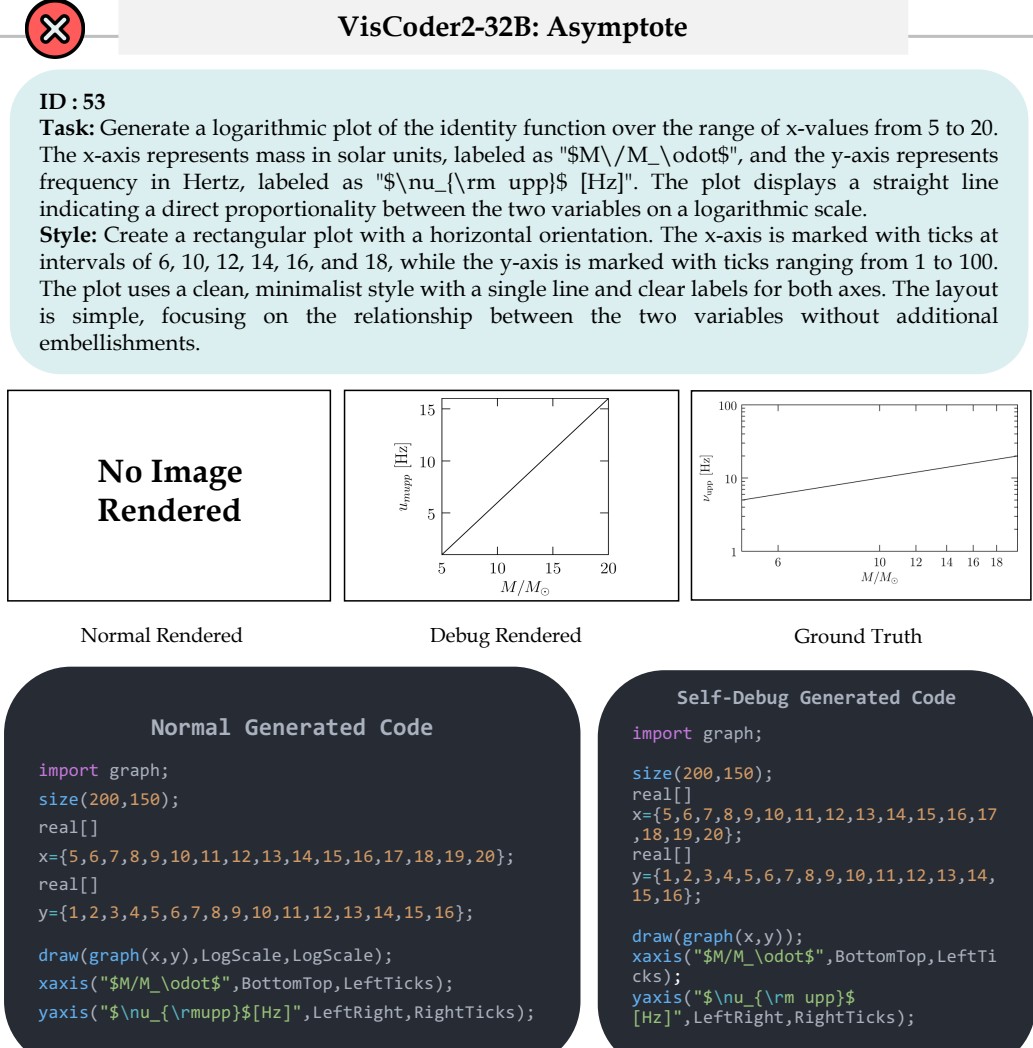

Figure 24: Example of a failed generation in **Asymptote** (ID: 53), where the initial code raises a **NameError** and is resolved in the **third** round of self-debug, resulting in a corrected plot that matches the intended semantics.
[Back to Appendix Contents]

## J.21 ASYMPTOTE: SELF-DEBUG FAILED

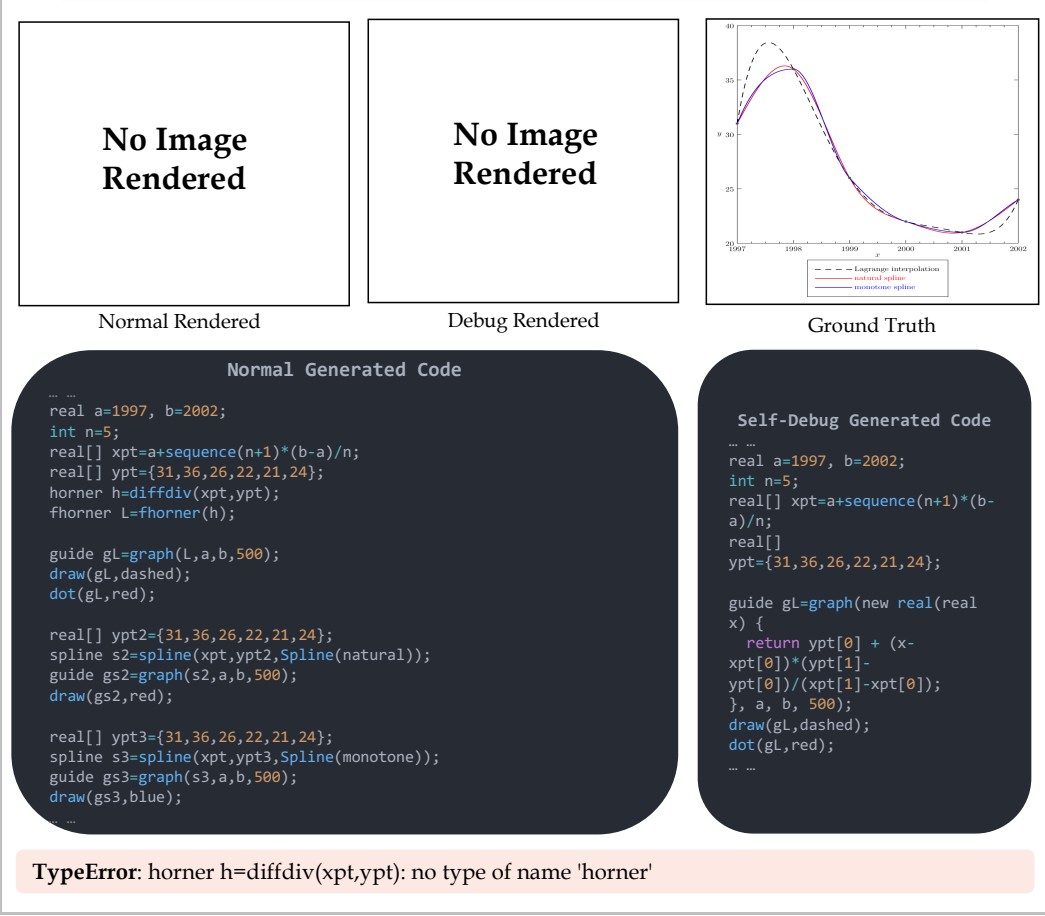

Figure 25: Example of a failed generation in **Asymptote** (ID: 79), where the initial code raises a **TypeError** and is still failed after three rounds self-debug.
[Back to Appendix Contents]

## J.22 HTML: SUCCESSFUL GENERATION

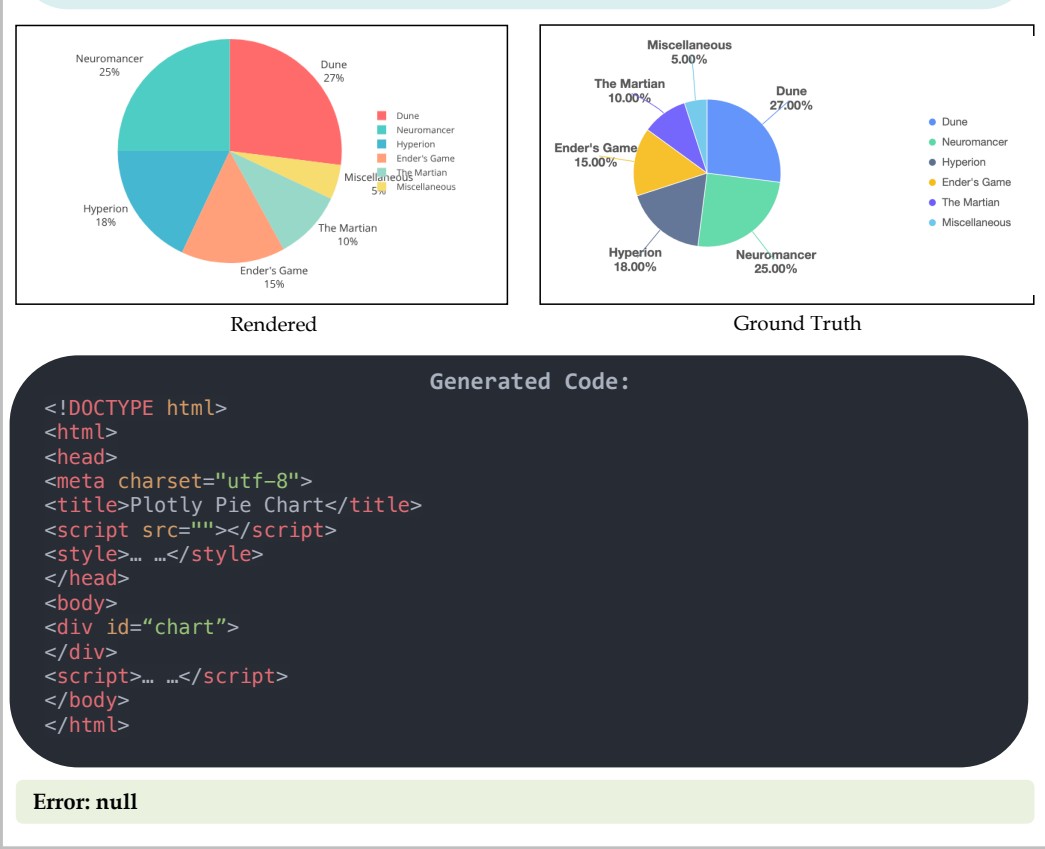

**VisCoder2-32B: HTML**

**ID : 6**
**Task:** Generate a pie chart to display the distribution of various science fiction categories. The chart segments represent different categories, such as "Dune," "Neuromancer," "Hyperion," "Ender's Game," "The Martian," and "Miscellaneous," with each segment sized according to its value. The chart includes labels showing the category name and its percentage of the total.
**STYLE:** Create a centered layout with a light gray background. The pie chart is placed within a white, rounded rectangle with a subtle shadow, providing a clean and modern look. The title, "Science Fiction Categories Distribution," is centered above the chart in a bold, dark gray font. The legend is positioned to the right of the pie chart, with category names in a smaller font size, each colored to match its corresponding pie segment. Labels on the pie chart are bold and positioned outside the segments for clarity.

Rendered

Ground Truth

**Generated Code:**

```html
<!DOCTYPE html>
<html>
<head>
<meta charset="utf-8">
<title>Plotly Pie Chart</title>
<script src=""></script>
<style>… …</style>
</head>
<body>
<div id="chart">
</div>
<script>… …</script>
</body>
</html>
```

**Error: null**

Figure 26: Example of a successful generation in **HTML** (ID: 6). The model generates code that executes successfully and produces a plot consistent with the ground truth.
[Back to Appendix Contents]

## J.23 HTML: SELF-DEBUG RECOVERY

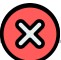 **VisCoder2-32B: HTML**

**ID : 9**

**Task:** Generate a radial bar chart to visualize the number of GitHub stars for various AntV projects. Each bar represents a project, with the length corresponding to the star count. The bars are color-coded based on the number of stars, with different colors indicating different ranges of star counts. The chart includes a central annotation displaying the total number of stars across all projects.

**Style:** Create a circular chart with a white background and a subtle shadow effect. The bars are arranged in a radial pattern, with rounded ends and varying colors. The chart is centered on the page, with a title above it. Inside the chart, there is a central text area with a GitHub icon, the word "AntV," and the total star count in blue. The overall layout is clean and balanced, with a modern aesthetic.

|  |  |  |
|---|---|---|
| **No Image Rendered** | 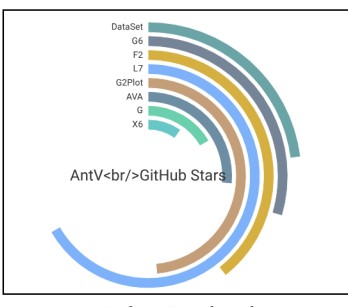 | 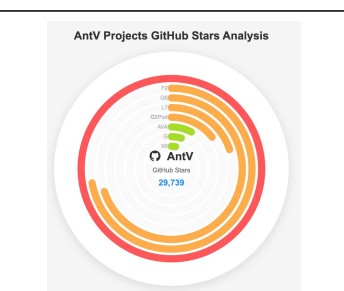 |
| Normal Rendered | Debug Rendered | Ground Truth |

**Normal Generated Code**

```
… …
<script>
… …
annotations: [
{… …},
{
  type: 'image',
  start: ['50%', '50%'],
  src:
'https://gw.alipayobjects.com/zos/basement_prod/6f38e5a2-
0a1a-4a0e-8b07-6e3a5a7b7b7a.svg',
  width: 64,
  height: 64,
  offsetY: -120
}
]
});
plot.render();
</script>
</body>
</html>
```

**Self-Debug Generated Code**

```
… …
< annotations: [
{
  type: 'text',
  position: ['50%', '50%'],
  content: 'AntV
GitHub
Stars',
  style: {
  fontSize: 24,
  fill: '#333',
  textAlign: 'center',
  textBaseline: 'middle'
  }
}
]
… …
</script>
</body>
</html>
```

**ImportError**: HTML Request failed: https://......7a.svg (404 Not Found).

Figure 27: Example of a failed generation in **HTML** (ID: 9), where the initial code raises a **ImportError** and is resolved in the **first** round of self-debug, resulting in a corrected plot that matches the intended semantics.

[Back to Appendix Contents]

## J.24 HTML: SELF-DEBUG FAILED

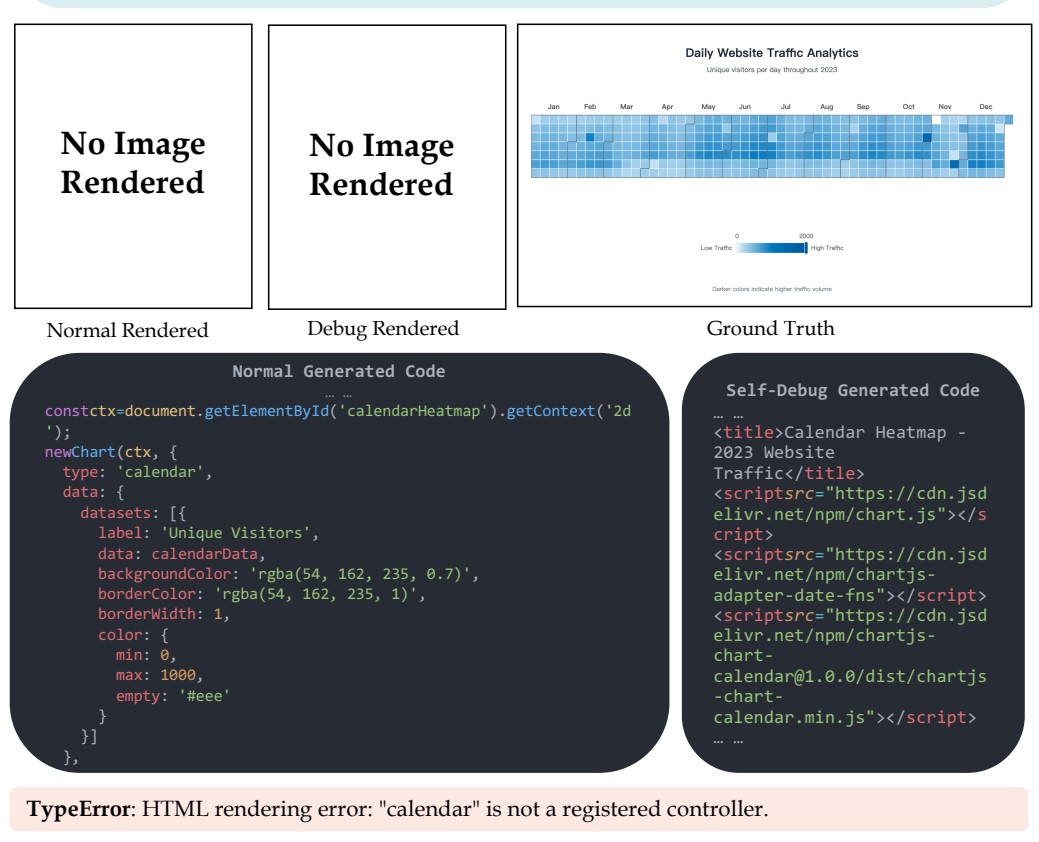

Figure 28: Example of a failed generation in **HTML** (ID: 85), where the initial code raises a **TypeError** and is still failed after three rounds self-debug.

## K    LLM USAGE

Large Language Models (LLMs) were used to assist with language editing and polishing of the manuscript. Specifically, we used an LLM to refine wording, improve readability, and enhance clarity in several sections.

The LLM was not involved in research ideation, methodology, data analysis, or experimental design. All technical ideas, analyses, and conclusions were developed and validated by the authors; the tool was limited to editorial support.

The authors take full responsibility for the content of the manuscript, including any text edited with LLM assistance. We verified suggested edits for accuracy and originality, ensured compliance with ethical guidelines, and noted that LLMs are not listed as authors. We disclose this usage in accordance with ICLR's policies on LLM assistance and attribution.

