# OpenReview forum: "VisCoder2: Building Multi-Language Visualization Coding Agents"
_ICLR.cc/2026/Conference — ICLR 2026 Poster_

### Official Review · Reviewer_wBjo · 2025-10-27

**Soundness:** 2
**Presentation:** 3
**Contribution:** 2
**Rating:** 2
**Confidence:** 4

**Summary:**

This paper studies the task of generating visualization code from natural language instructions. It introduces VisCode-Multi-679K, a large multi-language dataset containing executable code-image pairs and multi-turn correction dialogues; VisPlotBench, a benchmark for evaluation; and VisCoder2, a fine-tuned model based on Qwen2.5-Coder. Experiments show that VisCoder2 outperforms open-source baselines and approaches the performance of proprietary models like GPT-4.1, with further gains from iterative self-debug.

**Strengths:**

- S1: A large multi-language dataset and a benchmark are introduced; both are useful to the community.
- S2: The overall task setting aligns with real-world needs and is worth studying.
- S3: The fine-tuned models VisCoder2 show good performance in terms of execution pass rate.

**Weaknesses:**

- W1: The dataset is curated by combining existing datasets with filtering and cleaning, so the dataset contribution feels limited.
- W2: The paper evaluates different coding models (mostly scaling within the same model family). Including a wider range of models (e.g., GPT-5 or reasoning-oriented models) would make the comparisons more informative; scaling size alone is not very insightful, as larger models usually perform better.
- W3: More discussion is needed on the evaluation protocol (Section 3.4). Evaluating visual outputs semantically is inherently challenging, as it requires judging whether the generated image aligns with the textual task. Therefore, the key metrics -- Task Score  and Visual Score -- are important. However, both are judged using an LLM-as-a-judge, whose reliability is neither validated nor justified in this benchmark. Can the LLM accurately evaluate these metrics? If yes, to what extent? A more detailed analysis is needed.
- W4: The paper’s results rely heavily on the execution pass rate, which is not highly informative. Execution success only indicates that the code runs without errors, but one could “cheat” by outputting syntactically valid yet semantically incorrect code. This makes the analysis less convincing. More discussion and results based on the Task Score and Visual Score would make the evaluation more meaningful.
- W5: It would be valuable to analyze why other models (e.g., GPT-4.1) fail and to explain how fine-tuning helps reduce specific types of errors. Which errors does fine-tuning effectively mitigate, and which remain challenging? What insights can be drawn for practitioners to improve visualization-coding models?
- W6: The author claims that they are building "Coding Agents". But there is no real autonomous loop, planning, or tool integration -- only self-debugging within a single model. So the use of "agent" feels somehow overclaimed.
- W7: The paper lacks sufficient task statistics for the benchmark dataset (e.g., category distribution, code length distribution, difficulty distribution, etc.). Some statistics are provided, but more detailed and fine-grained statistics would strengthen the paper’s contribution as a benchmark paper.

**Questions:**

- Line 427: "Execution success is high across most models" --> the execution success of which models?
- Why does Qwen2.5-Coder-32B-Instruct perform worse than the 14B model in Table 2?
- How are the training and testing data split? How do you guarantee the quality and independence of the training and testing sets?
- Lines 396–398: What does "LilyPond shows the largest gains on symbolic grammars" mean? And what does “SVG exposes model-library sensitivity where semantic and perceptual signals diverge" mean?
- Regarding the fine-tuned models: how do they perform on other benchmarks? Does the model lose its capabilities on other types of tasks (e.g., MMMU, HumanEval)?
- How does performance scale with the number of self-debugging rounds? Do more rounds lead to better performance? Is there a saturation point? More analysis would be helpful.

---

> ### Author Response · Authors · 2025-11-18
> **Response [1/3]**
>
> ### For Weakness
>
> 1. We would like to clarify that VisCode-Multi-679K is not a simple combination of existing datasets. For the 613K multi-language visualization samples, we construct a complete multi-stage pipeline consisting of multi-source code filtering, code block extraction and reconstruction, Jupyter-based runtime validation, and instruction generation. None of the original sources contain ready-made instruction and visualization code pairs. Meanwhile, although the 66K Code-Feedback multi-turn dialogues are incorporated directly into the training corpus, their role is to provide supervision for iterative code repair based on execution feedback, complementing single-turn visualization samples that cannot teach self-debug behavior. Therefore, the core contribution of our dataset lies in providing a supervised instruction tuning corpus for visualization code generation and feedback-driven correction across twelve programming languages, rather than merely filtering or cleaning existing resources.
>
> 2. Thank you for the suggestion. We have added the GPT-5 results on VisPlotBench:
>
> |Model|Python|||||Vega-Lite|||||LilyPond|||||Mermaid|||||
> |:---:|:---:|:---:|:---:|:---:|:---:|:---:|:---:|:---:|:---:|:---:|:---:|:---:|:---:|:---:|:---:|:---:|:---:|:---:|:---:|:---:|
> ||**ExecPass**|**Mean**||**Good(>=75)**||**ExecPass**|**Mean**||**Good(>=75)**||**ExecPass**|**Mean**||**Good(>=75)**||**ExecPass**|**Mean**||**Good(>=75)**||
> |||**visual**|**task**|**visual**|**task**||**visual**|**task**|**visual**|**task**||**visual**|**task**|**visual**|**task**||**visual**|**task**|**visual**|**task**|
> |**GPT5**|68.9|54|62|52%|60%|83.6|60|68|54%|67%|56.4|12|50|2%|49%|55.0|30|45|14%|45%|
> |**GPT5+SelfDebug**|85.7|67|76|65%|76%|96.1|65|77|57%|75%|89.1|22|77|5%|71%|89.3|53|76|27%|76%|
>
> |Model|SVG|||||LaTeX|||||Asymptote|||||HTML|||||
> |:---:|:---:|:---:|:---:|:---:|:---:|:---:|:---:|:---:|:---:|:---:|:---:|:---:|:---:|:---:|:---:|:---:|:---:|:---:|:---:|:---:|
> ||**ExecPass**|**Mean**||**Good(>=75)**||**ExecPass**|**Mean**||**Good(>=75)**||**ExecPass**|**Mean**||**Good(>=75)**||**ExecPass**|**Mean**||**Good(>=75)**||
> |||**visual**|**task**|**visual**|**task**||**visual**|**task**|**visual**|**task**||**visual**|**task**|**visual**|**task**||**visual**|**task**|**visual**|**task**|
> |**GPT5**|93.9|43|81|17%|82%|28.6|20|24|17%|23%|17.4|10|15|7%|13%|91.7|48|63|21%|51%|
> |**GPT5+SelfDebug**|95.4|46|91|18%|92%|67.8|43|58|29%|55%|42.4|22|37|8%|34%|98.1|53|68|24%|55%|
>
>
> Regarding the concern that “scaling size alone is not very insightful,” our goal is not to study model scale itself, but to provide a visualization-code benchmark that enables comparison across different model types and sizes. Including multiple scales within the same model family ensures that the benchmark remains informative across capacities and enables us to verify the consistent gains brought by VisCode-Multi-679K at different scales.
>
> 3. Please refer to **General Response 1.2**.
>
> 4. Please refer to **General Response 3.2**.

---

> ### Author Response · Authors · 2025-11-18
> **Response [2/3]**
>
> 5. Thank you for the suggestion. Building on the analysis in **Section 4.3** for *VisCoder2-32B*, we additionally examine *GPT-4.1* using the breakdown error type results in **Appendix J**. The results are shown below:
>
> |ErrorCategory|Python|Vega-Lite|LilyPond|Mermaid|SVG|LaTeX|Asymptote|HTML|
> |--------------------------|---------|-----------|----------|---------|-------|--------|-----------|--------|
> |StructuralErrors|2→2|9→2|31→17|35→7|3→2|8→5|3→1|0→0|
> |Type&Interface|37→22|9→2|2→2|6→1|0→0|0→0|31→18|0→0|
> |Semantic/Data|20→1|0→0|0→0|0→0|0→0|21→15|21→16|0→0|
> |Runtime/Env.|11→6|2→0|1→1|0→0|0→0|35→23|17→14|11→3|
>
> *GPT-4.1* shows a relatively stable reduction of structural and type/interface errors across languages during multi-round self-debugging. In Python, Type & Interface errors decrease from 37 to 22, while structural errors remain within two cases; in Vega-Lite, type/parse errors drop from 9 to 2 and structural errors from 9 to 2; Mermaid’s syntax/structure errors fall substantially from 35 to 7; and Asymptote’s function-signature-related interface errors decrease from 31 to 18. Overall, these shallow errors consistently shrink across rounds.
>
> In contrast, *GPT-4.1* provides only limited improvement on semantic and runtime-environment errors. In LaTeX, Semantic/Data errors decrease only from 21 to 15, and Runtime/Env. errors from 35 to 23, remaining the dominant error types; in Asymptote, semantic errors fall from 21 to 16 and runtime-environment errors from 17 to 14; LilyPond’s markup/grammar errors drop from 31 to 17 but remain substantial. These deep errors exhibit much higher residual counts after self-debug compared with structural and interface errors.
>
> Regarding the question of “which errors fine-tuning mitigates and which remain challenging,” **Section 4.3** of the paper already provides this discussion. Fine-tuning, together with self-debug, reliably reduces structural and type/interface errors that expose explicit diagnostic signals (e.g., missing syntax, invalid arguments, or mismatched signatures). In contrast, semantic/data and runtime/environment errors remain difficult to resolve, particularly in symbolic or renderer-dependent languages such as LaTeX and Asymptote.
>
> Based on **Section 4.3** and the supplementary analysis of *GPT-4.1* above, we believe practitioners aiming to improve visualization-coding models may consider two directions: (1) For structural and interface errors, which exhibit clear diagnostic cues in multi-language settings, expanding library-call coverage, adding training examples with representative syntax patterns, and retaining execution logs with lightweight repair loops during inference can yield stable improvements. (2) For semantic/data and runtime/environment errors, which are most prominent in symbolic or renderer-dependent languages and stem from cross-statement dependencies, variable consistency, and implicit rendering rules, constructing more grammar-aware training samples, incorporating supervision that enforces cross-statement consistency, and strengthening coverage of symbolic structures can better address these deeper failure modes.
>
> 6. Regarding the concern that the use of “coding agents” may be overstated, our paper uses the term visualization coding agents to describe an iterative system built around a real multi-language execution environment. In each round, the model interacts with an execute → render → self-debug loop, where code is run, visual outputs are rendered, and runtime feedback is used for automatic correction. This process is not merely self-debugging within a single model, but relies on the VisPlotBench framework, which integrates execution engines, rendering backends, and error-log capture across eight languages. Therefore, our use of the term “agent” refers specifically to this execution-feedback-driven iterative mechanism rather than a fully autonomous planning or tool-orchestration agent.

---

> ### Author Response · Authors · 2025-11-18
> **Response [3/3]**
>
> 7. Thank you for raising the concern regarding missing task statistics. For category distribution, we explicitly report the distribution in **Section 3.1**, and **Appendix G** provides the full statistics of Visual Categories and Subtypes. For code length distribution, we do not plan to release ground truth code or provide code-length statistics, since exposing the code would lead to data leakage. For difficulty distribution, we assign Easy / Medium / Hard labels based on the library usage complexity, statement structure depth, and visual complexity of the rendered figures. The results are as follows:
>
> |Difficulty|Overall|Python|Vega-Lite|LilyPond|Mermaid|SVG|LaTeX|Asymptote|HTML|
> |:----------:|:-------:|:------:|:---------:|:--------:|:-------:|:-----:|:-----:|:---------:|:-----:|
> |Easy|32.1%|30.1%|34.9%|30.9%|39.7%|35.4%|25.0%|25.0%|35.2%|
> |Medium|48.9%|44.9%|50.4%|54.5%|50.4%|44.6%|50.0%|55.4%|45.4%|
> |High|19.0%|25.0%|14.7%|14.5%|9.9%|20.0%|25.0%|19.6%|19.4%|
>
> Finally, we would like to clarify that the contribution of this paper goes far beyond the benchmark alone, not only "contribution as a benchmark paper". We present a complete multi-language visualization-coding framework, including the large-scale training dataset VisCode-Multi-679K, the cross-language execution-and-rendering benchmark VisPlotBench, and the *VisCoder2* model family. The overall system-level contribution is summarized in the Abstract, Introduction, and Conclusion.
>
> ### For Question
>
> 1. Thank you for the question. The phrase “execution success is high across most models” refers specifically to the SVG results in Table 2, as indicated by the paragraph heading. In the SVG column, *GPT-4.1*, *GPT-4.1-mini*, *Qwen2.5-Coder*, and several *VisCoder2* variants all achieve high execution pass rate.
>
> 2. This behavior is due to the baseline model’s own cross-language generalization differences. We re-evaluated both *Qwen2.5-Coder-14B-Instruct* and *Qwen2.5-Coder-32B-Instruct* on VisPlotBench and obtained consistent results. It is also worth noting that the 32B model does not underperform the 14B model across all languages, breakdown results can be found in **Appendix H**.
>
> 3. Sorry we do not perform a train–test split because the training dataset (VisCode-Multi-679K) and the benchmark dataset(VisPlotBench) are constructed independently, so no split is required. The detailed construction procedures are described in **Section 2** and **Section 3.2** of the paper.
>
> 4. “LilyPond shows the largest gains on symbolic grammars” means that LilyPond is a strongly symbolic, grammar-driven language, and the targeted symbolic-grammar coverage in VisCode-Multi-679K leads to the largest improvement of *VisCoder2* over baselines in both execution and semantic correctness. “SVG exposes model–library sensitivity where semantic and perceptual signals diverge” indicates that most models can execute SVG code, but the visual rendering depends heavily on library-specific behaviors, causing high task scores but comparatively low visual scores due to rendering-level rather than semantic differences.
>
> 5. Since our work focuses on multi-language visualization code generation, models were not trained with multimodal inputs and cannot be evaluated on multimodal benchmarks such as MMMU. For general code capabilities, following your suggestion, we provide HumanEval results and analysis in **General Response 2.2**.
>
> 6. Please refer to **General Response 3.1**. For complete experimental details, please refer to **Appendices I and J**.

---

### Official Review · Reviewer_ipL2 · 2025-10-31

**Soundness:** 2
**Presentation:** 4
**Contribution:** 4
**Rating:** 8
**Confidence:** 3

**Summary:**

The authors propose VisCoder2, a multi-language visualization coding agent designed to generate, execute, and iteratively correct visualization code. To support this, they introduce VisCode-Multi-679K, a large-scale dataset of executable visualization code and correction dialogues across twelve programming languages, and VisPlotBench, a diverse benchmark for systematic evaluation. Experiments show that VisCoder2 outperforms open-source baselines and matches the reliability of proprietary models like GPT-4.1, especially with self-debugging. These resources advance the development of practical, robust visualization coding agents.

**Strengths:**

* **Focus on an Important Problem:**

  The paper targets automatic visualization code generation and correction, a task that is increasingly important for data analysis and reporting with LLMs but still lacks robust, multi-language solutions.

* **High-Quality, Multi-Language Dataset:**

  VisCode-Multi-679K covers twelve programming languages, including both popular and symbolic ones. The dataset is large-scale, contains only executable code, and includes multi-turn correction dialogues. These features make it valuable for developing and benchmarking robust visualization coding agents.

* **Effective Application of Iterative Correction:**

    The paper proposes an iterative correction approach that, while not highly novel (being similar to established program repair methods), is effective in practice. The results show significant improvements.

**Weaknesses:**

* **Lack of Quantitative Dataset Analysis:**

  The paper does not provide sufficient quantitative metrics about the dataset (such as error rates, diversity, redundancy, or semantic alignment). Without these, it is difficult to fully evaluate the dataset’s quality and practical value.

**Questions:**

Please address my concern in the Quantitative Analysis. Thanks

---

> ### Author Response · Authors · 2025-11-18
> **Response [1/2]**
>
> ### For Weakness & Question
>
> Thank you for pointing out the lack of quantitative analysis in our dataset. Following your suggestion, we have added more detailed descriptions of error rates, diversity, redundancy, and semantic alignment for the visualization portion of the data.
>
> 1. For the **error rates**, we report the error proportions at each of the three stages in the visualization data pipeline, as shown below:
>
> |Stage/Language|Python|Vega-Lite|LilyPond|Mermaid|SVG|LaTeX|Asymptote|HTML|JavaScript|TypeScript|C++|R|
> |--------------------------------------|---------:|----------:|---------:|--------:|--------:|--------:|----------:|---------:|-----------:|-----------:|--------:|--------:|
> |Initial Samples|2,657,158|6,864|12,097|13,627|185,313|134,600|25,297|1,400,763|1,325,343|429,035|1,024,674|257,438|
> |Libs Filter ErrorRate|91.90%|0%|0%|0%|57.16%|0%|0%|87.12%|93.03%|96.75%|92.30%|87.77%|
> |Code Extract ErrorRate|0.10%|0%|0%|0%|0%|0%|0%|0.40%|21.90%|5.61%|3.18%|20.75%|
> |Runtime Validation ErrorRate|13.03%|1.08%|0.03%|1.81%|41.27%|7.85%|10.90%|24.73%|60.07%|52.08%|78.05%|46.15%|
> |Final Valid Samples|186,954|6,790|12,093|13,381|46,621|124,039|22,539|135,230|28,807|6,315|16,776|13,437|
>
>
> 2. For diversity，We analyze the visualization type distribution in VisCode-Multi-679K as below:
>
> |**VisType**|**Count**|**Category**|**VisType**|**Count**|**Category**|**VisType**|**Count**|**Category**|
> |----------------------|-------|---------------|------------------------|-------|------------|------------------------|-------|-----------|
> |area|8278|Basic|surface-3d|5862|Surface|network|14579|Network|
> |bar|73201|Basic|flow-field|382|Surface|parallel-coordinates|447|Network|
> |candlestick|427|Basic|surface|5994|Surface|sankey|1036|Network|
> |donut|980|Basic|ternary-plot|1223|Surface|annotation|139|Diagram|
> |dotplot|666|Basic|vector-field|6194|Surface|class-diagram|542|Diagram|
> |grid|13944|Basic|volume-render|1998|Surface|er-diagram|702|Diagram|
> |line|53992|Basic|animation-frame|1744|Temporal|flowchart|4855|Diagram|
> |pie|19064|Basic|event|230|Temporal|generic-diagram|12055|Diagram|
> |polar|3951|Basic|timeseries|2599|Temporal|gitgraph|662|Diagram|
> |radar|790|Basic|timeline|3663|Temporal|other-diagram|19060|Diagram|
> |rectangle|3896|Basic|calendar|301|Schedule|quadrant-chart|696|Diagram|
> |rule|1538|Basic|gantt|2304|Schedule|requirement-diagram|600|Diagram|
> |scatter|29628|Basic|bubble|22040|Relation|sequence-diagram|606|Diagram|
> |spike-line|199|Basic|jointplot|577|Relation|state-diagram|1122|Diagram|
> |streamgraph|228|Basic|regression|2044|Relation|dataflow-diagram|1295|Diagram|
> |density|172|Distribution|category-chart|510|Categorical|circle|834|Geometry|
> |hexbin|1294|Distribution|funnel|1314|Categorical|circuit|10233|Geometry|
> |histogram|11354|Distribution|gauge|297|Categorical|geometry|23727|Geometry|
> |kde|9855|Distribution|indicator|1623|Categorical|reference-shape|4579|Geometry|
> |rug|367|Distribution|waterfall|916|Categorical|choropleth|8244|Map|
> |swarm|941|Distribution|wordcloud|281|Categorical|dot-map|5053|Map|
> |box|11097|Box|cluster|609|Hierarchy|symbol-map|1773|Map|
> |errorbar|1734|Box|dendrogram|630|Hierarchy|geographic-line-map|915|Map|
> |interval|660|Box|icicle|647|Hierarchy|geospatial-plot|9028|Map|
> |violin|1726|Box|mindmap|923|Hierarchy|document-page|10432|Document|
> |correlationmatrix|811|Matrix|sunburst|789|Hierarchy|image|13140|Document|
> |contourmatrix|2424|Matrix|tree|2062|Hierarchy|math-box|582|Document|
> |heatmap|11070|Matrix|treemap|1780|Hierarchy|problem-box|10924|Document|
> |adjacencymatrix|1075|Matrix|arcdiagram|427|Network|table|51821|Document|
> |splom|409|Matrix|chord|577|Network|text|8302|Document|
> |||||||textbox|58688|Document|
>
> We categorize a total of 91 visualization types across 15 categories, showcasing the diversity and coverage of VisCode-Multi-679K in multi-language visualization.

---

> ### Author Response · Authors · 2025-11-18
> **Response [2/2]**
>
> 3. Regarding the **redundancy** concern: the upstream sources used in our dataset construction (*Stack-Edu/Smol-IDs*, *CoSyn-400K*, and *SVG-diagrams*) have already undergone deduplication before release. Building on that, we further performed additional redundancy checks on the **visualization code parts** of VisCode-Multi-679K. Specifically, we separately normalized the *instruction* and *code* components and applied SHA-1 fingerprinting for **exact-duplicate** detection. The redundancy statistics for all visualization samples as shown below:
>
> ### Global Redundancy
> |Component|TotalSamples|UniqueSamples|RedundantSamples|RedundancyRate|
> |---------------|---------------|----------------|--------------------|------------------|
> |Instruction|612,982|611,671|1,311|**0.21%**|
> |Code|612,982|594,382|18,600|**3.03%**|
>
> ### Per-language Redundancy
> |Language|Samples|InstrRedund(%)|CodeRedund(%)|
> |-----------|----------:|----------------:|----------------:|
> |Asymptote|22,539|0.00|0.38|
> |C++|16,776|0.00|0.92|
> |HTML|135,230|0.00|0.28|
> |JavaScript|28,807|0.00|**8.33**|
> |LaTex|124,039|0.00|0.00|
> |LilyPond|12,093|0.00|0.00|
> |Mermaid|13,381|0.00|0.93|
> |Python|186,954|**0.70**|**8.24**|
> |R|13,437|0.00|0.15|
> |SVG|46,621|0.00|0.00|
> |TypeScript|6,315|0.00|0.55|
> |Vega-Lite|6,790|0.00|0.00|
>
> Overall, the instruction redundancy is only **0.21%**, and the code redundancy is **3.03%**, with consistently low redundancy across all languages. These results demonstrate that VisCode-Multi-679K exhibits substantial diversity in both natural-language instructions and executable visualization code, and is not dominated by duplicated samples.
>
> 4. For semantic alignment, we explicitly addressed this issue during instruction generation. Since the semantics of visualization tasks come jointly from the code and the rendered image, and different data sources vary in structure and language characteristics, we designed separate instruction-generation templates for different sources and languages (**Appendix F.1**), rather than using a single universal prompt. These templates enforce a fixed “segmented + structured” format that constrains the content of each instruction, ensuring consistency between the data semantics, visual structure, and rendered elements. During construction, we also randomly sampled 300 examples per language for manual inspection to prevent semantic drift introduced by prompt design.

---

### Official Review · Reviewer_4iTA · 2025-11-01

**Soundness:** 3
**Presentation:** 3
**Contribution:** 3
**Rating:** 4
**Confidence:** 4

**Summary:**

The paper introduces VisCoder2, a large-scale dataset and benchmark for multi-domain, multilingual visual code generation. It integrates code, visual plots, and textual prompts across 12 programming languages and multiple application domains (data visualization, UI generation, etc.). The benchmark aims to evaluate both code correctness and visual fidelity in multi-turn coding tasks. The dataset fills an important gap in current research, providing a broader and more diverse benchmark than existing ones like PandasPlotBench. Experiment results demonstrate the effectiveness of the proposed training data.

**Strengths:**

- Comprehensive dataset coverage: The dataset spans multiple languages and visual domains, filling a real void in existing benchmarks which are typically language- or task-limited.

- Practical relevance: Targets the emerging need for models capable of generating visual outputs (plots, figures, GUIs) from natural-language instructions.

- Readable and well-structured writing: The paper is easy to follow, with clear organization and accessible examples.

**Weaknesses:**

- Weak multi-turn construction: The “multi-turn” setup seems artificial — simply mixing in existing conversational code data rather than building domain-specific, multimodal dialogues. This undermines the claimed contribution on multi-turn reasoning.

- Incomplete evaluation coverage: Although 12 languages are used for training, only 8 are evaluated. The absence of results for the remaining languages leaves the dataset’s multilingual utility unverified.

- Limited baseline comparison: The model is only evaluated on the proposed benchmark. There’s no comparison against other visual code generation benchmarks (e.g., PandasPlotBench), making it hard to assess generalization.

- Unclear metrics: The paper introduces “visual” and “task” scores but fails to define how visual quality is assessed. For visual generation, perceptual and structural accuracy are as important as code execution success.

- Missing analysis on debug capability: Given the inclusion of multi-turn data, it’s surprising that no explicit debugging evaluation is performed, nor any comparison to instruction-tuned models like Qwen2.5 Coder Instruct. This weakens claims of enhanced reasoning or debugging ability.

**Questions:**

- How were the “visual” and “task” metrics computed? Are they human-judged, automatic, or hybrid?

- What criteria guided the choice of 8 evaluation languages — were they the largest subsets, or randomly sampled?

- How does the model perform on existing benchmarks like PandasPlotBench? Any transfer results?

- Does the multi-turn data actually improve multimodal debugging? Or just the general debugging ability shared by regular code LLMs?

---

> ### Author Response · Authors · 2025-11-18
>
> ### For Weakness
> 1. We provide the complete self-debug protocol in **General Response 1.1** and show its alignment with the real-world “generate–execute–repair” workflow. To clarify, the paper does not claim multi-turn reasoning as a contribution, nor do we aim to build domain-specific or multimodal dialogue systems. The multi-turn Code-Feedback data included in VisCode-Multi-679K is used solely to supervise iterative code repair based on execution feedback, rather than to construct multi-turn semantic reasoning for visualization tasks. The experiments and analyses in VisPlotBench focus on multi-language visualization performance and self-debug behavior, not on dialogue-based reasoning ability. While multimodal self-debug is beyond the scope of this work, it remains an interesting direction. In future work, we plan to explore evaluation settings that also provide the rendered image alongside the code, enabling multimodal models to debug visualization outputs across multi-language.
>
> 2. Thank you for raising the concern regarding incomplete evaluation coverage. Compared with the training data, VisPlotBench does not include C++, R, JavaScript, or TypeScript because, during construction of the evaluation framework, the Jupyter-based kernels for these languages pose substantial engineering challenges for multi-round self-debug and unified framework integration. We plan to extend support in future work. As emphasized in **Section 3.1** of the paper, existing benchmarks all focus on a single programming language, whereas VisPlotBench expands the evaluation scope to eight commonly used visualization languages, providing a more comprehensive verification of multi-language utility. Meanwhile, our work targets multi-programming-language settings rather than multilingual evaluation.
>
> 3. Please refer to **General Response 2.1**, where we include the additional evaluation on PandasPlotBench, demonstrating the generalization ability of the *VisCoder2* series models.
>
> 4. Thank you for the suggestion. We clarify the metric definitions in **General Response 1.2**, and further provide in-depth analysis of the Task/Visual Score results in **General Response 3.2**.
>
> 5. We provide explicit debugging evaluations in **Sections 4.1 and 4.2** of the paper, including direct comparisons with the instruction-tuned *Qwen2.5-Coder-Instruct* model. Detailed breakdown results are also provided in **Appendices H and I**. Furthermore, as discussed in General Response 3, we offer an in-depth analysis of self-debug behavior, covering both execution pass rate and Task/Visual Scores.
>
> ### For Question
>
> 1. Please refer to **General Response 1.2**.
>
> 2. We selected the 12 languages in our training data because they cover the most commonly used and widely adopted visualization programming languages in both academia and industry. The 8 evaluation languages in VisPlotBench are neither the largest subsets nor randomly sampled; rather, they represent the subset that can stably execute, render images, and support multi-round self-debug within our Jupyter-based evaluation framework. C++, R, JavaScript, and TypeScript were not included due to substantial engineering challenges during integration, and we plan to extend support in future work.
>
> 3. Please refer to the PandasPlotBench evaluation results provided in **General Response 2.1**, where the model demonstrates effective transfer performance on this existing benchmark.
>
> Does the multi-turn data actually improve multimodal debugging? Or just the general debugging ability shared by regular code LLMs?
>
> 4. We do not aim to address “multimodal debugging” in this paper. *VisCoder2* is a code-only model that does not take image inputs, and the training data contain no multimodal components; therefore, the multi-turn data cannot improve any form of multimodal debugging. The multi-turn Code-Feedback data are used solely to  supervise iterative code repair based on execution feedback, enhancing general debugging ability rather than multimodal debugging.

---

### Official Review · Reviewer_R3P7 · 2025-11-02

**Soundness:** 3
**Presentation:** 1
**Contribution:** 3
**Rating:** 8
**Confidence:** 4

**Summary:**

This paper introduces VisCoder2, a framework for building multi-language visualization coding agents. Specifically, the work contain three key components: VisCode-Multi-679K, a large-scale dataset of 679K executable visualization code–image pairs with multi-turn correction dialogues across 12 programming languages; VisPlotBench, a benchmark spanning 8 languages and 13 visualization categories, supporting both single-round and multi-round self-debug evaluation; VisCoder2, a family of open-source multi-language models trained on the above dataset.

**Strengths:**

1. The author provides a comprehensive dataset construction. They detail a well-structured pipeline including language filtering, runtime validation, and instruction synthesis. Each visualization sample is paired with rendered outputs and multi-turn feedback, ensuring executable, realistic supervision. Compared to the previous method, VisCode-Multi-679K supports 12 languages, which makes its coverage cover most languages.
2. The author provides extensive experiments; the proposed VisCoder surpasses open-source baselines by 10–15 points in execution pass rate and achieves parity with GPT-4.1 on several languages. The analysis of self-debug gains and error types (syntax vs. runtime vs. semantic) is particularly insightful.
3. The paper is well-organized and easy to follow.

**Weaknesses:**

1. The detail of self-debugging is missed. The paper demonstrates that iterative correction improves performance, but it does not deeply analyze how models leverage feedback logs. It is better to describe how the self-debug works. Moreover, the author shows that there are persistent failures in semantic and runtime errors. It is also better to provide an analysis of why self-debugging does not work on these cases rather than just saying it doesn't work.
2.  The current evaluation is imbalanced. Execution pass rate dominates the quantitative results, whereas semantic accuracy and visual similarity (Task/Visual Scores) are only partially reported. I understand the space is limited, and the author provides full results in the appendix. However, compared to the execution pass rate, whether the generated results are correct is more important. If possible, it is better to manually check the results and report the correctness for every model.
3. Both the training corpus (VisCoder) and the evaluation benchmark (VisPlotBench) are constructed by the authors and share the same generation–execution–debug protocol. The paper does not evaluate zero-shot transfer to unseen visualization libraries, natural user prompts, or noisy real-world code, leaving generalization ability uncertain, which limits the usage of the model. It is better to provide natural user test results.

**Questions:**

1. The current LLMs like GPT-5 can also support image input. Is it possible to extend the current dataset? e.g., provide the image input and request code output.
2.  It is better to provide the process of self-debug and better evaluation results.

---

> ### Author Response · Authors · 2025-11-18
>
> ### For Weakness
> 1. Thank you for the suggestion. As addressed in **General Response 1.1 and 3.1**, we provide a detailed description of the multi-round self-debug workflow, as well as an analysis of the failure patterns that explain why self-debug is less effective in certain cases across different languages.
>
> 2. We agree that correctness is more important than execution success alone. Task/Visual Scores are equally important metrics in our evaluation. As discussed in **General Response 3.2**, we have added analyses for the remaining five languages that were omitted from the main paper due to space limits, and based on additional manual verification, we further analyze how self-debug affects Task/Visual Scores across languages and how these scores relate to execution pass rate.
>
> 3. Thank you for raising this concern. Our training corpus and VisPlotBench do not share the same “generation–execution–debug” protocol. The training data consist of (i) multi-language visualization samples containing only single-turn instructions and executable code, and (ii) a general feedback dataset that is not focused on visualization debugging. As discussed in **Appendix D** (Ablation Study), this separation is also reflected in the empirical results. Therefore, the self-debug ability evaluated in VisPlotBench is unseen for *VisCoder2* and represents a genuine zero-shot transfer of the iterative correction mechanism. Regarding natural user prompts, we plan to provide an interactive demo after acceptance, allowing users to test the model under real, unconstrained inputs.
>
> ### For Question
> 1. Thank you for the valuable suggestion. Multi-language image-to-code generation is still underexplored, and we will consider this direction in future work.
>
> 2. Please refer to **General Response 1 and 3**.

---

### Author Response · Authors · 2025-11-18
**General Response [5/5]**

(3) Mermaid shows notable gains across all metrics. For *GPT-4.1*: the Visual Score rises from 41 to 56, the Task Score from 57 to 77, and the proportion of high-scoring samples increases correspondingly. *VisCoder2* consistently outperforms *Qwen2.5-Coder* at the same scale, and both visual and task performance improve steadily with self-debug, reflecting that in languages with explicit graph structures and detailed runtime diagnostics, models can use feedback to refine both relational semantics and diagram layout.

(4) Asymptote remains one of the most challenging languages. Although most models show some improvement in Task/Visual Score, the overall level remains significantly lower than in Python or Vega-Lite. For example, *VisCoder2-32B* increases its Task Score from 46 to 53 after self-debug, while the Visual Score only rises from 27 to 31 with almost no change in high-score proportions, indicating that models can more easily align symbolic task semantics yet struggle to reliably control geometric details and rendering behavior, leaving a substantial gap between semantic correctness and visual consistency.

(5) In HTML, strong models such as *GPT-4.1* and *VisCoder2* already exhibit high initial Task/Visual Score, and self-debug leads only to moderate gains. For example, *GPT-4.1* increases its Visual Score from 48 to 51 and its Task Score from 64 to 68, while *VisCoder2-32B* increases its Visual Score from 43 to 44 and its Task Score from 61 to 62. Overall, HTML specifications are relatively “model-friendly” in terms of semantic alignment, and remaining discrepancies are more reflective of minor visual deviations introduced by layout strategies and default rendering behaviors rather than true semantic errors.

**2. Effects and limitations of self-debug on Task/Visual Score.**

Across all eight languages, self-debug generally improves Task/Visual Score alongside execution reliability, with the clearest gains appearing in languages where models can leverage informative diagnostics. Python and Mermaid exemplify this trend: in Python, *VisCoder2-32B* improves from a Task Score of 56 to 68 and from a Visual Score of 49 to 58, while *GPT-4.1* in Mermaid rises from 57 to 77 (task) and 41 to 56 (visual). These cases show that when feedback exposes meaningful semantic or structural signals, models can refine both correctness and rendered appearance rather than merely repairing syntax. In contrast, LaTeX and Asymptote show limited visual improvement despite higher execution success (e.g., *VisCoder2-32B* increases visual only from 27 to 31), reflecting that symbolic or compiler-dependent failures often surface only as coarse parser messages, offering insufficient guidance for deeper refinement. SVG represents the opposite extreme: execution and Task Score increase slightly, yet Visual Score remain almost unchanged (*GPT-4.1* stays near 45; *VisCoder2-32B* increases only from 33 to 34), indicating that text-only feedback cannot reveal layout- or rendering-sensitive discrepancies. Taken together, these findings show that Task/Visual Score uncover a distinct layer of quality that execution alone cannot detect, and that the effectiveness of self-debug critically depends on how well feedback exposes semantically or visually meaningful error signals.

**3. Cross-language relationship between Exec Pass and Task/Visual Score.**

Examining all eight languages reveals two broad regimes. In Python, Vega-Lite, Mermaid, and HTML, execution success, task accuracy, and visual fidelity tend to rise together, indicating that most errors stem from structural or interface issues that, once resolved, directly translate into improved semantics and rendering; in these settings, execution pass rate is a reasonable proxy for overall quality. LaTeX and SVG, however, demonstrate clear decoupling: models often achieve much higher execution success without corresponding gains in semantic or visual alignment. For LaTeX, *GPT-4.1* reaches a 66.1% execution pass rate after self-debug, yet Task/Visual Score remain in the mid-fifties and twenty-to-forty ranges due to macro expansion and compile-time fragility. For SVG, execution frequently saturates while Visual Score remain around forty to fifty across models, reflecting rendering-library sensitivity that feedback cannot capture. Asymptote lies between these extremes: all three metrics remain low but tend to improve synchronously, highlighting the intrinsic difficulty of symbolic geometric drawing and that performance on this language is still far from saturated. Together, these cross-language patterns show where execution is informative of downstream quality and where Task/Visual Score provide essential complementary signals.

---

### Author Response · Authors · 2025-11-18
**General Response [4/5]**

3. **Characteristics of self-debug:** Across all models and languages, the first round of self-debug consistently yields the largest improvement, correcting the majority of failures caused by shallow issues such as missing syntax, mismatched parameters, or incorrect references. Starting from the second round, the rate of improvement drops sharply, and by the third round performance typically plateaus. This pattern indicates that the current feedback mechanism effectively exposes structural and interface-related errors, but provides limited signals for deeper semantic inconsistencies, complex symbolic dependencies, or issues tied to specific rendering or parsing processes. As a result, self-debug follows a stable “large first-round gains followed by diminishing returns” trajectory across the entire evaluation.
4. **Failures remain across models:** Building upon the error categorization in **Section 4.3**, we further examined the final unsuccessful cases of *GPT-4.1* and *VisCoder2-32B* across eight visualization languages and found that the remaining failures fall into three representative patterns that are difficult for current self-debug mechanisms to resolve: (1) deep semantic inconsistencies, such as mismatched variable meanings, incorrect data relationships, or incoherent multi-step plotting logic, which rarely surface explicitly in execution logs and therefore prevent the model from identifying the true source of failure; (2) grammar- or compiler-dependent symbolic errors, including macro expansion, symbol binding, or scope-resolution failures in languages like LilyPond, Asymptote, and LaTeX, where parser messages tend to be generic and provide little actionable guidance; and (3) runtime-related behavioral errors, such as invoking rendering packages that were never loaded, calling APIs unsupported by the target language, or generating plotting logic that may trigger infinite loops or excessive resource consumption. Although these issues manifest as runtime failures, the execution feedback typically contains only coarse, non-localized error signals, making it difficult for the model to refine its output across self-debug rounds. Overall, the lack of explicit localization for semantic, symbolic, and runtime behavior-related errors constitutes the primary bottleneck behind the remaining unresolved cases.
### 3.2 Task/Visual Score Analysis
In VisPlotBench, the execution pass rate and the Task/Visual Score are equally important evaluation dimensions. In **Section 4.2** of the main paper, we analyze three representative languages that highlight different behaviors and discuss the phenomena of execution–semantic mismatch, symbolic grammar gains, and rendering library sensitivity. To further address the reviewers’ questions regarding semantic correctness and visual alignment, we extend the analysis using the complete results provided in **Appendix H**, covering the remaining five languages and examining the changes in Task/Visual Score before and after self-debug as well as their overall relationship with execution pass rates.

**1. Supplementary Task/Visual Score analysis for the remaining languages.**

(1) For Python, we observe that from *Qwen2.5-Coder* to *VisCoder2*, and further with self-debug enabled, the mean Task/Visual Score, the proportion of high-quality samples, and the execution pass rate improve in a largely synchronized manner. For example, at the 32B scale, *VisCoder2-32B* increases its Visual Score from 49 to 58 and its Task Score from 56 to 68, with the proportion of samples scoring visual ≥ 75 rising from 42% to 46% and task ≥ 75 from 54% to 62%. This shows that execution improvements are primarily driven by joint gains in semantic alignment and visual quality rather than by shallow syntax repairs.

(2) In Vega-Lite, strong models such as *GPT-4.1* and *VisCoder2* already approach saturated performance after self-debug, yet Task/Visual Score still show mild upward trends; for example, for *VisCoder2-32B* the Visual Score increases from 60 to 62 and the Task Score from 70 to 72. This suggests that in declarative languages with explicit rules and rich diagnostics, models already produce mostly correct specifications, and self-debug mainly handles long-tail issues.

---

### Author Response · Authors · 2025-11-18
**General Response [3/5]**

## 3. Deep Analysis of Self-Debug Behavior and Task/Visual Score
### 3.1 Self-Debug Analysis
In **Appendix I**, we report the complete self-debug results of all models across the eight visualization languages. To address the reviewer’s concerns regarding how models behave during self-debug, we select four representative systems: *GPT-4.1*, *GPT-4.1-mini*, *Qwen2.5-Coder-32B-Instruct*, and our fine-tuned *VisCoder2-32B*, and provide a deeper analysis that combines language-specific characteristics with model behaviors. Tables 3, 4, and 5 present the corresponding results.

### Table 3. Execution Pass Rate across Self-Debug Rounds (Python, Vega-Lite, LilyPond)
|Model|**Python**||||**Vega-Lite**||||**LilyPond**||||
|--------------------------|:---------:|:----:|:----:|:----:|:-------------:|:----:|:----:|:----:|:------------:|:----:|:----:|:----:|
||Normal|R1|R2|R3|Normal|R1|R2|R3|Normal|R1|R2|R3|
|GPT-4.1|64.3|75.0|81.6|84.2|84.5|95.4|96.1|96.1|45.5|58.2|61.8|65.5|
|GPT-4.1-mini|64.8|73.5|79.1|80.6|84.5|95.4|96.9|96.9|22.2|37.0|50.0|57.4|
||||||||||||||
|Qwen2.5-Coder-32B-Ins.|50.5|70.9|78.1|79.1|83.0|87.6|89.9|89.9|30.9|40.0|43.6|43.6|
|VisCoder2-32B|65.3|76.0|80.1|81.6|94.6|96.1|96.1|96.1|56.4|61.8|69.1|69.1|

### Table 4. Execution Pass Rate across Self-Debug Rounds (Mermaid, SVG, LaTeX)
|Model|**Mermaid**||||**SVG**||||**LaTeX**||||
|--------------------------|:-----------:|:----:|:----:|:----:|:-------:|:----:|:----:|:----:|:---------:|:----:|:----:|:----:|
||Normal|R1|R2|R3|Normal|R1|R2|R3|Normal|R1|R2|R3|
|GPT-4.1|68.7|84.7|93.9|93.9|92.3|93.9|95.4|95.4|31.3|53.6|59.8|66.1|
|GPT-4.1-mini|51.9|81.7|90.1|94.7|89.1|95.3|95.3|96.9|29.5|50.9|55.4|58.9|
||||||||||||||
|Qwen2.5-Coder-32B-Ins.|71.0|74.8|75.6|76.3|93.9|93.9|93.9|93.9|29.5|42.9|50.0|51.8|
|VisCoder2-32B|87.0|89.3|90.1|90.1|81.5|84.6|86.2|86.2|42.9|55.4|59.8|61.6|

### Table 5. Execution Pass Rate across Self-Debug Rounds (Asymptote, HTML)
|Model|**Asymptote**||||**HTML**||||
|--------------------------|:-------------:|:----:|:----:|:----:|:--------:|:----:|:----:|:----:|
||Normal|R1|R2|R3|Normal|R1|R2|R3|
|GPT-4.1|21.7|35.9|43.5|46.7|89.8|96.3|97.2|97.2|
|GPT-4.1-mini|23.9|37.0|42.4|48.9|86.1|99.1|99.1|100.0|
||||||||||||||
|Qwen2.5-Coder-32B-Ins.|17.4|25.0|31.5|33.7|78.7|88.9|89.8|89.8|
|VisCoder2-32B|58.7|68.5|71.7|71.7|91.7|92.6|93.5|93.5|

1. **Effect of self-debug:** Self-debugging consistently improves execution reliability across all models and most languages. *GPT-4.1* and *GPT-4.1-mini* already exhibit the strongest cross-language performance at the initial generation stage, and self-debug further repairs the majority of syntax- and interface-related errors, allowing them to reach near-saturated execution rates in languages such as Python, Vega-Lite, and Mermaid. In contrast, *Qwen2.5-Coder-32B-Instruct* starts from a weaker baseline, but still benefits substantially from iterative correction. Our fine-tuned *VisCoder2-32B* achieves significantly higher initial and final execution success rates than the baseline in seven languages, with especially large gains in symbol-intensive languages such as LilyPond, Asymptote, and LaTeX, where self-debug brings it close to or even above the GPT models. Together, these results show that self-debug offers a robust, model-agnostic mechanism for correcting structural and shallow syntactic errors, and is a key driver of multi-language reliability.
2. **Cross-language trends:** In declarative languages like Vega-Lite and HTML, clear structural rules and diagnostics allow most models to reach near-saturated execution within one or two rounds. In execution-driven languages such as Python and Mermaid, rich runtime diagnostics provide actionable signals that drive steady improvements across rounds. For symbolic or compiler-dependent languages (LilyPond, Asymptote, LaTeX), models fix shallow syntax early, but deeper semantic issues rarely surface in logs, so improvements taper off after the first round. For SVG, the rendering pipeline is sensitive to XML well-formedness but offers little semantic or layout feedback. Larger models generate more complex structures, making issues like unclosed tags or malformed attributes more common, and these are difficult to repair under weak feedback, limiting improvement.

---

### Author Response · Authors · 2025-11-18
**General Response [2/5]**

## 2. Additional Evaluation Results
### 2.1 Results on PandasPlotBench
To address reviewers’ concerns regarding generalization beyond our proposed benchmark, we additionally evaluate *Qwen2.5-Coder*, *VisCoder*, *VisCoder2*, and proprietary models (*GPT-4.1* / *GPT-4.1-mini*) on PandasPlotBench, which covers Python-based visualization libraries including Matplotlib, Seaborn, and Plotly. Table 1 reports execution pass rate, Task Score, Visual Score, and the proportion of samples achieving a score ≥75.
### Table 1: Results of VisCoder2 and Baselines on the PandasPlotBench Benchmark
|Model|Matplotlib|||||Seaborn|||||Plotly|||||
|:---|:---:|:---:|:---:|:---:|:---:|:---:|:---:|:---:|:---:|:---:|:---:|:---:|:---:|:---:|:---:|
||**ExecPass**|**Mean**||**Good(>=75)**||**ExecPass**|**Mean**||**Good(>=75)**||**ExecPass**|**Mean**||**Good(>=75)**||
|||**visual**|**task**|**visual**|**task**||**visual**|**task**|**visual**|**task**||**visual**|**task**|**visual**|**task**|
|GPT-4.1|94.3|75|88|69%|91%|93.7|72|86|68%|86%|76.6|61|67|58%|66%|
|GPT-4.1+SelfDebug|**100**|**77**|**90**|70%|**94%**|98.9|**74**|**89**|**70%**|**90%**|**97.7**|**74**|**85**|**69%**|85%|
|GPT-4.1-mini|94.3|74|86|71%|87%|92|71|83|64%|85%|70.9|55|62|51%|63%|
|GPT-4.1-mini+SelfDebug|98.9|76|89|**73%**|91%|**100**|**74**|87|67%|**90%**|97.1|72|84|65%|**86%**|
|||||||||||||||||
|Qwen2.5-Coder-3B-Ins.|71.4|56|72|50%|**69%**|58.3|44|55|36%|51%|27.4|17|19|17%|18%|
|VisCoder-3B|81.7|60|69|53%|**69%**|73.7|48|65|38%|61%|60.6|38|45|32%|44%|
|VisCoder-3B+SelfDebug|85.1|60|70|53%|**69%**|**78.3**|48|**66**|37%|**62%**|**64.6**|40|48|34%|47%|
|VisCoder2-3B|83.4|62|70|55%|**69%**|73.7|51|62|42%|56%|61.1|41|48|35%|45%|
|VisCoder2-3B+SelfDebug|**86.3**|**63**|**71**|**56%**|**69%**|77.7|**53**|64|**43%**|58%|64|**43**|**52**|**37%**|**49%**|
|||||||||||||||||
|Qwen2.5-Coder-7B-Ins.|78.3|63|76|58%|75%|68.6|51|63|40%|62%|48|29|34|24%|31%|
|VisCoder-7B|87.4|66|78|60%|80%|76.6|57|70|50%|68%|74.3|48|60|41%|61%|
|VisCoder-7B+SelfDebug|91.4|67|**81**|**62%**|**83%**|90.3|62|**77**|51%|**75%**|81.7|51|65|44%|65%|
|VisCoder2-7B|87.4|67|76|61%|78%|83.4|61|72|52%|70%|77.7|48|62|43%|63%|
|VisCoder2-7B+SelfDebug|**92**|**69**|78|**62%**|80%|**93.7**|**64**|76|**53%**|**74%**|**87.4**|**53**|**68**|**47%**|**67%**|
|||||||||||||||||
|Qwen2.5-Coder-14B-Ins.|86.3|67|78|61%|78%|76.6|58|70|51%|67%|56|40|42|37%|39%|
|VisCoder-14B|86.3|-|-|-|-|78.9|-|-|-|-|74.3|-|-|-|-|
|VisCoder-14B+SelfDebug|93.7|-|-|-|-|92.6|-|-|-|-|93.1|-|-|-|-|
|VisCoder2-14B|88|70|81|63%|80%|84|66|74|58%|71%|78.3|52|66|46%|65%|
|VisCoder2-14B+SelfDebug|**94.3**|**71**|**83**|**65%**|**83**|**93.7**|**67**|**79**|**59%**|**78%**|**94.9**|**60**|**71**|**51%**|**70%**|

Across all three libraries, *VisCoder2* consistently outperforms the base *Qwen2.5-Coder* models on execution success as well as both semantic (task) and perceptual (visual) metrics. The gains further increase under the self-debug setting, where *VisCoder2-14B* achieves performance close to *GPT-4.1*. These results support the generalization capability of *VisCoder2* on unseen visualization benchmarks.
## 2.2 Human-Eval
We further evaluate *Qwen2.5-Coder-Instruct* and the fine-tuned *VisCoder2* models on HumanEval and HumanEval+. Table 2 reports the Pass@1 results.
### Table 2: Performance on HumanEval and HumanEval+
|Model|Human-EvalPass@1|Human-Eval-PlusPass@1|
|:---|:---:|:---:|
|GPT-4.1|97|91.5|
|GPT-4.1-mini|92.1|86.6|
||||
|Qwen2.5-Coder-3B-Ins.|84.8|79.9|
|VisCoder2-3B|81.1|76.2|
||||
|Qwen2.5-Coder-7B-Ins.|91.5|84.8|
|VisCoder2-7B|89|83.5|
||||
|Qwen2.5-Coder-14B-Ins.|92.1|86.6|
|VisCoder2-14B|92.1|84.8|
||||
|Qwen2.5-Coder-32B-Ins.|90.9|85.4|
|VisCoder2-32B|87.8|81.7|

The results show that *VisCoder2* exhibits only a modest 2–3 point decrease compared to the base *Qwen2.5-Coder* models on HumanEval/HumanEval+. This behavior aligns with the distributional differences between the two task families: *VisCoder2* is trained heavily on multi-language visualization code, whereas HumanEval focuses on algorithmic and data-structure problems. Such minor fluctuations are therefore expected and do not indicate systematic capability degradation. Overall, *VisCoder2* maintains stable general coding ability while achieving substantial gains on its target task of cross-language executable visualization code generation and multi-round self-debug.

---

### Author Response · Authors · 2025-11-18
**General Response [1/5]**

# General Response
We thank all reviewers for their constructive feedback. Below, we provide a consolidated General Response addressing the shared concerns across reviews, including clarifications of the VisPlotBench evaluation protocol, additional evaluations such as PandasPlotBench and HumanEval, and deeper analysis and insights from our experimental results.
## 1. VisPlotBench Evaluation Setting
### 1.1 Self-Debug Evaluation Protocol
In VisPlotBench, we adopt the same self-debug evaluation mode used in VisCoder to simulate a realistic developer-style debugging workflow. In this setting, if the model’s initial code generation fails to execute or does not produce a valid plot, the model is given up to K rounds to iteratively refine its output based on feedback from the previous attempt.
In each round, only the tasks that remain unsolved from the previous iteration are reconsidered. The model receives a multi-turn conversational prompt consisting of (i) the original natural-language instruction, (ii) the previously generated code that failed, and (iii) the feedback derived from the execution error. Based on this dialogue history, the model produces a revised version of the code. If the revised code executes successfully and generates a valid plot, the task is marked as solved and excluded from further rounds; otherwise, the latest failed output is recorded and carried forward to the next iteration.
```
Let F₀ be the set of failed tasks from the initial evaluation.
For i = 1 to K:
  For each task x in Fᵢ₋₁ that is not yet fixed:
    1. Generate a revised version of x using feedback-driven prompting.
    2. Execute the revised code.
    3. If execution succeeds:
         - Mark x as fixed and record the successful output.
       Else:
         - Record the latest failed output for x.
Evaluate every task using its final recorded output.
```
In all experiments, we set the maximum number of rounds to K = 3. After all rounds are completed, each task is evaluated using its latest recorded output (either the successfully corrected code from an earlier round or the final failed attempt) using the same evaluation pipeline as in the initial pass.
This iterative mechanism mirrors the common “generate–execute–repair” workflow and provides a standardized way to evaluate how models recover from different error types across programming languages.
### 1.2 Task and Visual Score Metrics
In VisPlotBench, we follow the scoring procedure introduced in PandasPlotBench, and the judge prompts are provided in **Appendix F.2**. The core idea is to use a GPT model to compare the ground-truth image and the model-rendered image within the context of the task description. For the Task Score, the judge compares the generated plot against the task instruction; for the Visual Score, the judge compares the generated plot against the ground-truth reference image. The Task and Visual Score metrics are described in detail in the PandasPlotBench paper.

---

### Meta-Review · Area_Chair_P8gJ · 2025-12-17

**Summary:**

This paper presents VisCoder2, a framework for building multi-language visualization coding agents. The contributions of the work consist of three main components: (1) VisCode-Multi-679K, a large-scale dataset containing 679K executable visualization code–image pairs with multi-turn correction dialogues across 12 programming languages; (2) VisPlotBench, a benchmark covering 8 languages and 13 visualization categories, supporting both single-round and multi-round self-debugging evaluation; and (3) VisCoder2, a family of open-source multi-language models trained on the proposed dataset. Reviewers have raised some concerns:
1. Some experimental settings are unclear, such as self-debug.
2. The details of curation process and constructed data are not very clear. I think more clarification indeed needs to avoid unnecessary misunderstanding.
3. More deep analyses should be added.
4. The details of Exec Pass and Task/Visual Score should be added.

During rebuttal process, the authors have conducted sufficient explanations and experiments to address these problems. These changes should be added in the next version.

**Reviewer Concerns:**

I believe that almost all of the reviewers’ concerns have been addressed.
Reviewers wBjo and 4iTA mainly raised questions regarding the experimental settings and the novelty of the paper. I think some of these concerns stem from unclear details in the manuscript, which the authors’ responses have largely clarified.

**Reviewer Scores:**

The scores of Review wBjo and 4iTA may be raised.

---

### Decision · Program_Chairs · 2026-01-26

Accept (Poster)